# Multi-ancestry genome-wide association study of major depression aids locus discovery, fine mapping, gene prioritization and causal inference

Most genome-wide association studies (GWAS) of major depression (MD) have been conducted in samples of European ancestry. Here we report a multi-ancestry GWAS of MD, adding data from 21 cohorts with 88,316 MD cases and 902,757 controls to previously reported data. This analysis used a range of measures to define MD and included samples of African (36% of effective sample size), East Asian (26%) and South Asian (6%) ancestry and Hispanic/Latin American participants (32%). The multi-ancestry GWAS identified 53 significantly associated novel loci. For loci from GWAS in European ancestry samples, fewer than expected were transferable to other ancestry groups. Fine mapping benefited from additional sample diversity. A transcriptome-wide association study identified 205 significantly associated novel genes. These findings suggest that, for MD, increasing ancestral and global diversity in genetic studies may be particularly important to ensure discovery of core genes and inform about transferability of findings.

Major depression (MD) is one of the most pressing global health challenges[1]. While genome-wide association studies (GWAS) have shown promise of uncovering biological mechanisms underlying the development of MD[2,3], they have revealed a highly polygenic genetic architecture, characterized by variants that individually confer small risk increases[4], probably due to the heterogeneity of MD symptoms and etiology[5]. Previous genetic research explored the impact of different outcome definitions[2,6], sex[7–9] and trauma exposure[10–12] on heterogeneity. However, the role of ancestry and ethnicity in the genetics of MD has not yet been systematically evaluated.

So far, GWAS of MD were mostly conducted in individuals of European ancestry[2,13–15]. The largest MD GWAS combined data from several studies and identified 223 independent significant single-nucleotide polymorphisms (SNPs)[15]. That study also included data from 59,600 African Americans from the Million Veteran Program (MVP) cohort. In their bi-ancestral meta-analysis, the number of significant SNPs increased to 233. Other MD GWASs were conducted in African American and Hispanic/Latin American participants with limited sample sizes, and did not find variants with statistically significant associations with MD[16,17].

With 10,640 female Chinese participants, the CONVERGE study is the largest MD GWAS conducted outside 'Western' countries so far[3]. The study identified two genome-wide significant associations linked to mitochondrial biology and reported a genetic correlation of 0.33 with MD in European ancestry samples[18]. In line with this, our recent work demonstrated that some of the previously identified loci from GWAS conducted in samples of European ancestry are not transferable to samples of East Asian ancestry[19].

Heterogeneity in genetic effects could impact on findings when evaluating causal effects of risk factors for MD. Previous studies in samples of European ancestry reported genetic correlations and causal relationships between MD and cardiometabolic outcomes[2,13,15,20]. Notably, our previous study indicated a contradicting direction for associations between MD and body mass index (BMI) in East Asian individuals and European ancestry individuals (positive causal effect of BMI)[19,21]. Thus, investigating causal relationships using Mendelian randomization (MR) in diverse ancestry groups and in different disease subtypes is important to ensure generalizability and to distinguish between biological and societal mechanisms underlying the relationship between a risk factor and the disease.

✉e-mail: k.kuchenbaecker@ucl.ac.uk

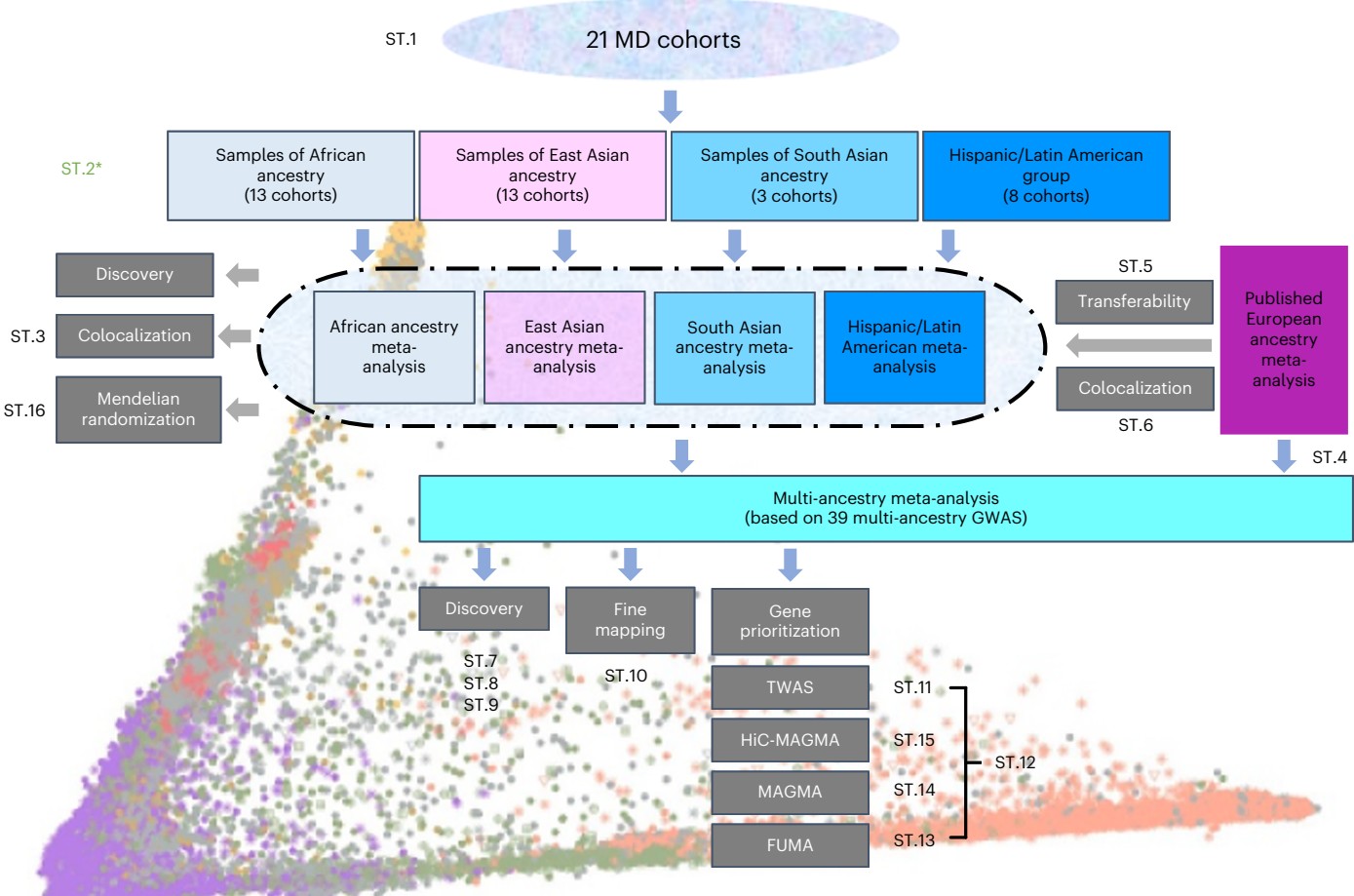

**Fig. 1 | Schematic diagram of the analyses in this study.** We included data from 21 cohorts with diverse ancestry. We assigned individuals into ancestry/ethnic groups and carried out association analyses with MD for each. Subsequently, we meta-analyzed the results by ancestry/ethnic group. We tested whether previously reported MD loci from European ancestry studies are transferable to these groups. We also used the results for discovery of novel depression associations and MR to assess the causal effects of cardiometabolic traits by ancestry. We subsequently merged all ancestry/ethnicity-specific results in a multi-ancestry meta-analysis that also included samples with European ancestry. The multi-ancestry meta-analysis results formed the basis for locus discovery, fine mapping to identify causal variants and several gene prioritization approaches to identify genes linked to MD risk. ST.(n) refers to the corresponding Supplementary Table. ST.2* (in green) refers to Supplementary Table 2, showing genomic inflation estimates of multiple analyses.

Increasing diversity in genetic research is also important to ensure equitable health benefits[22]. In the United States, differences in presentation of MD across ethnic groups can impact on the likelihood of diagnosis[23]. Genetics optimized for European ancestry participants would primarily benefit that group of patients and could therefore further widen the disparities in diagnosis and treatment between groups.

In this Article, we used data from samples with diverse ancestries and carried out genome-wide association meta-analyses, followed by fine mapping and prioritization of target genes (Fig. 1). We assessed the transferability of genetic loci across ancestry groups. Finally, we explored bi-directional causal links between MD and cardiometabolic traits.

## Results

### GWAS in African, East Asian and South Asian ancestry and Hispanic/Latin American samples

We first conducted GWAS meta-analyses stratified by ancestry/ethnic group. Individuals were assigned to ancestry groups (African, South Asian, East Asian or European) using principal component analyses based on genetic relatedness matrices. Assignment to the Hispanic/Latin American group was based on self-report or on recruitment in a Latin American country (Supplementary Figs. 1–7)[24–26]. We acknowledge the arbitrary nature of this approach and of choosing reference groups

and cut-offs to assign participants. However, creating such groups enabled us to look for associations that are specific to groups and to assess the transferability of previously identified loci. The studies included in the meta-analyses used the following measures to define MD: structured clinical interviews, medical healthcare records, symptoms questionnaires and self-completed surveys (Supplementary Table 1 and Supplementary Note).

The analyses included 36,818 MD cases and 161,679 controls of African ancestry, 21,980 cases and 360,956 controls of East Asian ancestry, 4,505 cases and 27,176 controls of South Asian ancestry, and 25,013 cases and 352,946 controls in the Hispanic/Latin American group (Extended Data Fig. 1 and Supplementary Figs. 8–11). To account for the minor inflation found in the Hispanic/Latin American samples ($\lambda_{1,000}$ = 1.002 and linkage disequilibrium score regression (LDSC) intercept 1.051; Supplementary Table 2), we corrected test statistics for this analysis based on the LDSC intercept.

In the Hispanic/Latin American group, the G-allele of rs78349146 at 2q24.2 was associated with increased risk of MD (effect allele frequency (EAF) 0.04, $\beta$ (regression coefficient) = 0.15, s.e.m. 0.03, $P$ = 9.3 × 10⁻⁹) (Supplementary Fig. 12). To test the role of these loci in molecular profiles, we performed colocalization for depression and multi-ancestry brain expression quantitative trait loci (eQTLs)[27]. Loci with posterior probability (PP) >90% for both traits being associated and sharing two

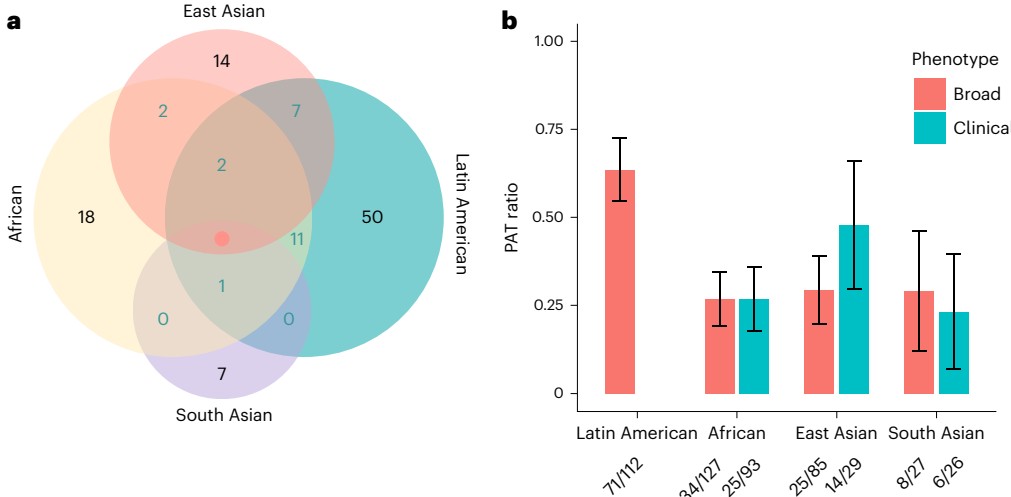

**Fig. 2 | Transferability of previously reported loci from European ancestry discovery GWAS of MD to other ancestry groups. a**, A Venn diagram showing the numbers of previously identified loci from European ancestry studies with evidence of transferability to the other ancestry/ethnic groups: African, Hispanic/Latin American, South Asian and East Asian (in black) and their intersections (in cyan). Only the 112 loci with evidence of transferability to at least one ancestry group are shown here. **b**, A plot showing power-adjusted transferability (PAT) ratios. We first calculated the observed number of transferable loci out of the 195, 196, 179 and 180 loci that were present in the

African, Hispanic/Latin American, South Asian and East Asian ancestries, respectively. These were divided by the expected number of transferable loci (numbers displayed underneath the figure), taking effect estimates from previous European ancestry studies, and allele frequency and sample size information from our African, Hispanic/Latin American, South Asian and East Asian ancestry cohorts. The ratios are presented separately for broadly defined MD and clinically ascertained MD. The error bars indicate 95% CIs for PAT ratios. We were unable to compute results for clinical MD in the Hispanic/Latin American group because of insufficient numbers of cases.

different but linked variants (hypothesis (H)3) or a single causal variant (H4) were considered as colocalized. We observed significant colocalization for *DPP4*, *RBMS1* and *TANK*. We tested ancestry-specific eQTLs from blood and observed *RBMS1* (H3: PP (Hispanic/Latin American) 99.12%) and *TANK* (H3: PP (European) 97.85%; H3: PP (Hispanic/Latin American) 99.61%) at the 2q24.2 locus. For the protein quantitative trait loci (pQTLs) from blood, we either did not find the genes in the cohort or the number of SNPs within the gene was too low (<20) to test for colocalization (Supplementary Table 3).

No variants were associated at genome-wide significance in the GWAS in samples of African, East and South Asian ancestry (Extended Data Fig. 1a,b,d). One locus was suggestively associated in the African ancestry GWAS (Extended Data Fig. 1a and Supplementary Fig. 12). The lead variant, rs6902879 (effect allele: A, EAF 0.16, $\beta = -0.08$, s.e.m. 0.01, $P = 5.3 \times 10^{-8}$) at 6q16 is located upstream of the melanin-concentrating hormone receptor 2 gene (*MCHR2*) and associated with increased expression of *MCHR2* in cortex based on genotype–tissue expression (GTEx v8) ($P = 6.0 \times 10^{-6}$). Testing the multi-ancestry brain eQTLs[27], we observed significant colocalization for *GRIK2* and *ASCC3*, with significant ancestry differences for *ASCC3* (H3: PP (European) 99.97%). *MCHR2* was not present in the RNA data.

Although the lead variants at 2q24.2 and at 6q16 did not display strong evidence of association in a large published GWAS in participants of European ancestry[14] ($P > 0.01$), in each case there was an uncorrelated variant within 500 kb of the lead variants associated at $P < 10^{-6}$ (Supplementary Fig. 12). Hence, although the evidence does not support a shared causal variant, we cannot rule out that there is an association at the same locus, but possibly with a different causal variant in European ancestry participants.

As a sensitivity analysis, we conducted a meta-analysis for each ancestry/ethnic group for clinical depression, comprising studies in which MD was ascertained by structured clinical interviews or medical healthcare records following the International Classification of Diseases (ICD9)/10 or the Diagnostic and Statistical Manual of Mental Disorders (DSM)-IV/5 criteria for major depressive disorder (Supplementary Table 1). There were 29,389 cases and 49,999 controls of

African ancestry, 7,886 cases and 14,412 controls of East Asian ancestry, 848 cases and 13,908 controls in the Hispanic/Latin American group, and 4,252 cases and 26,738 controls of South Asian ancestry (Extended Data Fig. 2 and Supplementary Figs. 13–16). In the South Asian ancestry GWAS, the A allele of rs7749931 at 6q15 was associated with decreased risk of MD (effect allele: A, EAF 0.49, $\beta = -0.15$, s.e.m. 0.03, $P = 4.3 \times 10^{-8}$) (Extended Data Fig. 2d and Supplementary Fig. 15). The variant is located downstream of *STX7* (syntaxin 7). We did not observe genome-wide significant loci associated with clinical depression in any other ancestry group.

**Transferability of MD associations across ancestry groups**

Previous GWAS in samples of European ancestry have identified 206 loci associated with MD (Supplementary Table 4)[13–15]. The results for 196 of these loci were available in at least one of the ancestry/ethnic groups. We assessed whether these genetic associations are shared across different ancestry groups. Individual loci may be underpowered to demonstrate an association; therefore, we followed an approach we recently developed[28] and first estimated the number of loci we expect to see an association for when accounting for sample size ($n$), linkage disequilibrium (LD) and minor allele frequency (MAF). This estimate varied widely between ancestry groups, for example, we expected to detect significant associations for 65% of MD loci in the GWAS with samples of African ancestry, but only for 15% of MD loci in samples of South Asian ancestry (Fig. 2). We report the power-adjusted transferability (PAT) ratio, that is, the observed number divided by the expected number of loci. Transferability was low, with PAT ratios of 0.27 (95% confidence interval (CI) 0.19 to 0.35) in African ancestry samples, and 0.29 in both East Asian (95% CI 0.20 to 0.39) and South Asian (95% CI 0.12 to 0.46) ancestry samples. In the Hispanic/Latin American group, the PAT ratio was 0.63 (95% CI 0.55 to 0.72), notably higher than in the other groups. PAT estimates for clinical MD were close to those for broad MD, with overlapping CIs in each case (Fig. 2). We were unable to estimate PAT ratios for clinical MD in the Hispanic/Latin American group because of insufficient numbers of cases based on this definition. We also assessed the transferability of 102 loci identified in the Psychiatric Genomics

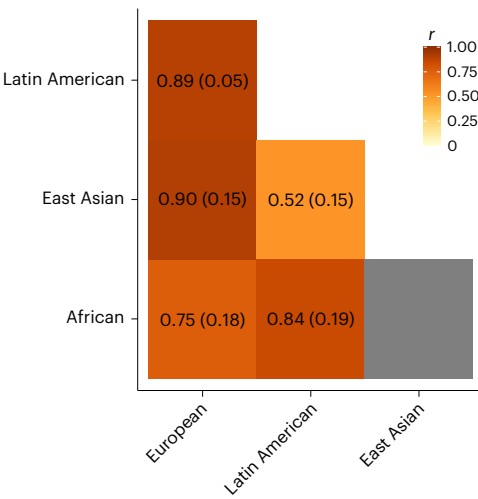

**Fig. 3 | Genetic correlations for MD between different ancestry groups.** A plot showing the genome-wide genetic correlations between the African, European, East Asian and Hispanic/Latin American groups. The intensity of the coloring reflects the strength of the correlation. The estimated coefficients and standard errors are also shown in each cell. We only present estimates where the s.e.m. was smaller than 0.3; otherwise, the field is colored in gray.

Consortium–Major Depressive Disorder Working Group's (PGC-MDD) GWAS[13] and in an independent study in samples of European ancestry, the Australian Genetics of Depression Study (AGDS)[14]. The PAT ratio was 1.48, considerably higher than the cross-ancestry PAT estimates. We report evidence of transferability of individual loci (Supplementary Table 5) as well as their ancestry-specific eQTL and pQTL colocalization (Supplementary Table 6 and Supplementary Fig. 17).

In addition, we estimated trans-ancestry genetic correlations using POPCORN version 1.0 (ref. 29). We only present genetic correlation estimates where the s.e.m. was less than 0.3. The sample size for the South Asian ancestry group was too small to conduct this analysis. The genetic correlations for MD between the European and the Hispanic/ Latin American, African and East Asian ancestry groups were ≥0.75. The lowest estimate was observed between the East Asian ancestry, and the Hispanic/Latin American group ($r_g$ = 0.52) (Fig. 3).

### Multi-ancestry meta-analysis

We carried out a multi-ancestry meta-analysis using data from studies conducted in participants of African, East Asian and South Asian ancestry and Hispanic/Latin American samples (Supplementary Note), and combined them with previously published data for 258,364 cases and 571,252 controls of European ancestry[13,14], yielding a total sample size of 345,389 cases and 1,469,702 controls. These analyses provided results for 22,941,580 SNPs after quality control. There was no evidence of residual population stratification ($\lambda_{1,000}$ = 1.001, LDSC intercept 1.019; Supplementary Table 2). We identified 190 independent genome-wide significant SNPs mapping to 169 loci that were separated from each other by at least 500 kb (Extended Data Fig. 3, Supplementary Table 7 and Supplementary Fig. 18). Fifty-three of the SNPs represent novel associations ($r^2$ < 0.1 and located more than ±250 kb from previously reported variants). Most of the 196 previously reported loci were associated at genome-wide significance in the multi-ancestry meta-analysis, which incorporates the discovery data for these loci (Supplementary Table 4).

As a sensitivity analysis, we also conducted a multi-ancestry meta-analysis for clinical depression. There were 57,714 cases and 110,358 controls of European ancestry under the clinical definition of MD, which were subsequently combined with the aforementioned non-European clinically ascertained studies by meta-analysis (100,089

cases and 214,415 controls in total) (Extended Data Fig. 4 and Supplementary Figs. 19 and 20). This analysis identified seven genome-wide significant loci, two of which were novel (rs2085224 at 3p22.3 and rs78676209 at 5p12) (Supplementary Table 8).

We then excluded cohorts that had an extreme case–control ratio ($n_{cases}/n_{controls}$ <0.25) and did not adjust for this analytically, as well as cohorts with adolescent participants. This sensitivity analysis also yielded consistent results for the 190 lead SNPs (Supplementary Fig. 21).

Finally, we re-analyzed the data using a multi-ancestry meta-analysis approach implemented in MR-MEGA, which resulted in 44 independent regions associated with MD after lambda GC correction, some of which had been missed in the main analyses due to their between-ancestry heterogeneity (Supplementary Table 9).

### Multi-ancestry fine mapping

We used a multi-ancestry Bayesian fine-mapping method[30] to derive 99% credible sets for 155 loci that were associated at genome-wide significance and did not show evidence of multiple independent signals. For comparison, we also implemented single ancestry fine mapping of the same loci based on GWAS conducted in participants of European ancestry, including PGC-MDD and AGDS[13,14].

Multi-ancestry fine mapping increased fine-mapping resolution substantially as compared with fine mapping solely based on the data from European ancestry participants. The median size of the 99% credible sets was reduced from 65.5 to 30 variants. Among the 145 loci for which we conducted fine mapping on both sample sets, 113 (77.9%) loci had a smaller 99% credible set from the multi-ancestry fine mapping, while four loci (0 from the European fine-mapping) were resolved to single putatively causal SNPs (Fig. 4 and Supplementary Table 10). For example, rs12699323, annotated as an intronic variant, is linked to expression of *TMEM106B* (transmembrane protein 106B). rs1806152 is a splice region variant associated with expression of the nearby gene *PAUPAR* (PAX6 upstream antisense RNA) on chromosome 11. At another locus, rs9564313 has been linked to expression of *PCDH9* (protocadherin-9), a gene that is also highlighted in our TWAS and multi-marker analysis of genomic annotation (MAGMA) results[31,32].

### TWAS and gene prioritization

To better understand the biological mechanisms of our GWAS findings, we performed several in silico analyses to functionally annotate and prioritize the most likely causal genes. We carried out a transcriptome-wide association study (TWAS) based on the results from the multi-ancestry meta-analysis for expression in tissues relevant to MD[33]. We combined the TWAS results with functional mapping and annotation (FUMA), conventional MAGMA and HiC-MAGMA to prioritize target genes.

The TWAS identified 354 significant associations ($P < 1.37 \times 10^{-6}$) with MD, 205 of which had not been previously reported (Fig. 5 and Supplementary Table 11). The two most significant gene associations with MD were *RPL31P12* (GTEx brain cerebellum, $Z = -10.68$, $P = 1.27 \times 10^{-26}$) and *NEGR1* (GTEx brain caudate basal ganglia, $Z = 10.677$, $P = 1.30 \times 10^{-26}$), consistent with previous findings[33].

*PCDH8P1* (GTEx brain anterior cingulate cortex BA24, $Z = -8.3679$, $P = 5.86 \times 10^{-17}$) was the most significant novel TWAS result. *NDUFAF3* was another novel gene association with MD (GTEx brain nucleus accumbens basal ganglia, $Z = -5.0785$, $P = 3.80 \times 10^{-7}$, best GWAS ID rs7617480, best GWAS $P = 0.00001$). These results were also confirmed by HiC-MAGMA. The protein *NDUFAF3* encodes is targeted by metformin, the first-line drug for treating type 2 diabetes.

Forty-three genes displayed evidence of association across all four gene prioritization methods (TWAS, FUMA, MAGMA and HiC-MAGMA) and were classified as high-confidence genes (Table 1 and Supplementary Tables 11–15). These included genes repeatedly highlighted in previous studies due to their strong evidence of association and biological relevance in MD: *NEGR1*, *DRD2*, *CELF4*, *LRFN5*, *TMEM161B*

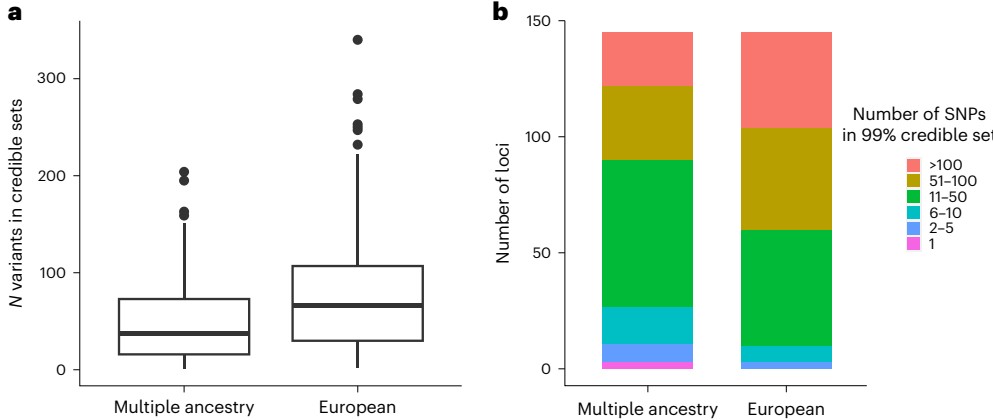

**Fig. 4 | Resolution of the locus fine mapping based on the multi-ancestry and the European ancestry GWAS, showing the size of the credible sets for 155 significant loci. a**, A box plot showing the median (central line) and interquartile range (upper and lower hinges) of the sizes of the credible set for fine-mapped loci. The whiskers extend to 1.5 times the interquartile range. Data points falling outside that range are denoted by individual dots in the figure. **b**, Stacked bar charts showing the number of loci within size categories for credible sets.

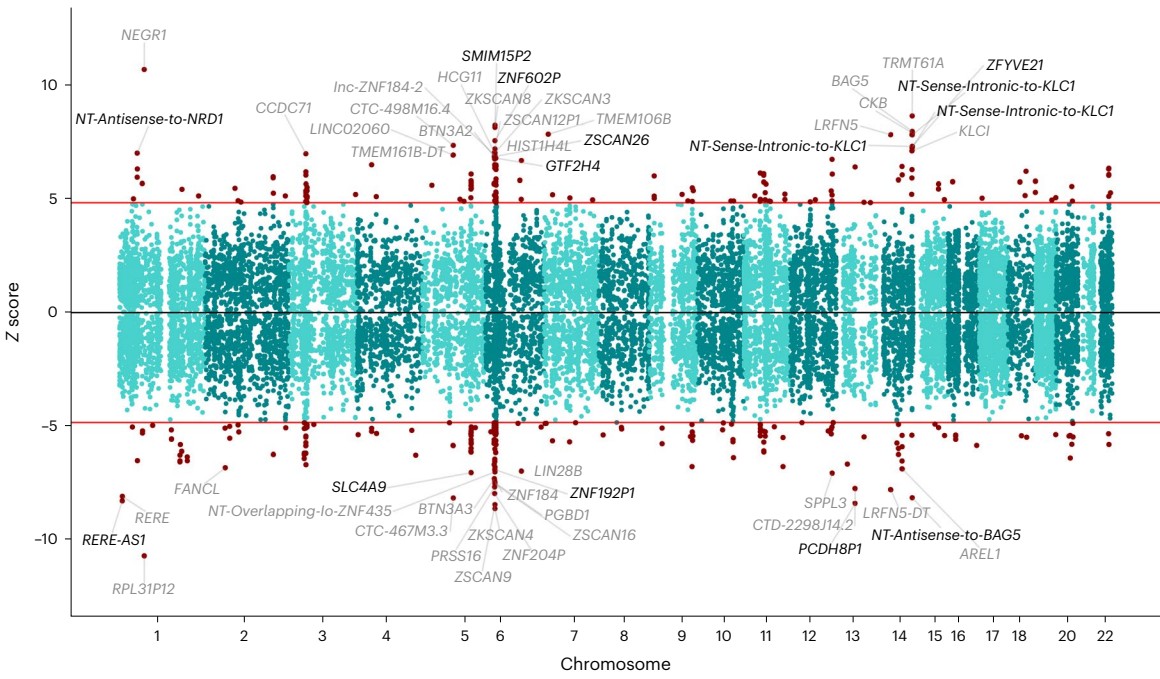

**Fig. 5 | Manhattan-style *Z*-score plot of gene associations with MD in a TWAS based on the GWAS summary statistics for broadly defined MD.** Significant gene associations are shown as red dots (354 significant genes, 205 of them novel), and the 50 most significant gene names are highlighted on both sides of the plot. Novel associations are shown in black, while genes previously associated with MD are shown in gray. The red lines indicate the significance threshold ($P < 1.37 \times 10^{-6}$). For genes on the top part of the graph, increased expression was associated with increased depression risk, while expression of the genes on the bottom part of the plot showed an inverse association. NT, novel transcript.

and *TMEM106B*. Cadherin-9 (*CDH9*) and protocadherins (*PCDHA1*, *PCDHA2* and *PCDHA3*) were also among the high-confidence genes (Supplementary Table 12). Finally, 25 of the high-confidence genes encode targets of established drugs, such as simvastatin (*RHOA*). These may indicate opportunities for drug repurposing.

## MR

We assessed bi-directional causal relationships between MD genetic liability and cardiometabolic traits using ancestry-specific two-sample MR analyses. Our results indicated a positive, bi-directional relationship between MD genetic liability and BMI (MD−>BMI: $\beta = 0.092$, 95% CI 0.024 to 0.161, $P = 8.12 \times 10^{-3}$, BMI−>MD: $\beta = 0.138$, 95% CI 0.097 to 0.180, $P = 6.88 \times 10^{-11}$) (Fig. 6 and Supplementary Table 16). This

bi-directional relationship was exclusively observed in samples of European ancestry ($P > 0.1$ in all other groups). MD genetic liability was also causal for other indicators of unfavorable metabolic profiles in samples of European ancestry: triglycerides (TGs, positive effect; $\beta = 0.116$, 95% CI 0.070 to 0.162, $P = 7.93 \times 10^{-7}$), high-density lipoproteins (HDLs, negative effect; $\beta = -0.058$, 95% CI −0.111 to −0.006, $P = 0.029$) and low-density lipoproteins (LDLs, positive effect; $\beta = 0.054$, 95% CI 0.012 to 0.096, $P = 0.011$). The effects remained significant after removing the variants contributing to the possible heterogeneity bias observed through the MR–pleiotropy residual sum and outlier global test. Additionally, no pleiotropy was observed (Supplementary Table 16). In samples of East Asian ancestry, on the other hand, we found a negative causal association between TG and MD ($\beta = -0.127$, 95% CI −0.223 to

## Table 1 | Genes associated with MD

| Gene[a] | Drug[b] | FUMA[c] | MAGMA$_j$[d] | Hi-C MAGMA[d] | TWAS $P$ | Novel[e] | Credible set[f] |
|---|---|---|---|---|---|---|---|
| **Genes associated in TWAS and Hi-C MAGMA** | | | | | | | |
| NDUFAF3 | Metformin, NADH | No | 1.00 | 0.004 | $3.80 \times 10^{-7}$ | Yes | 9 |
| PBRM1 | Alprazolam, durvalumab, everolimus | No | 0.10 | 0.017 | $3.20 \times 10^{-7}$ | Yes | – |
| TBCA | – | No | 0.16 | 0.042 | $1.29 \times 10^{-6}$ | Yes | – |
| BTN2A3P | – | No | 1.00 | $6.07 \times 10^{-8}$ | $2.33 \times 10^{-8}$ | Yes | – |
| ZNF204P | – | No | 1.00 | 0.014 | $2.13 \times 10^{-15}$ | No | – |
| HLA-B | Thalidomide, ticlopidine, phenobarbital, carbamazepine, clozapine, lamotrigine | No | 0.46 | $3.7 \times 10^{-4}$ | $1.13 \times 10^{-7}$ | No | – |
| RABGAP1 | – | No | 1.00 | 0.001 | $1.91 \times 10^{-8}$ | Yes | – |
| GOLGA1 | – | No | 1.00 | 0.020 | $1.56 \times 10^{-7}$ | Yes | – |
| FRAT2 | – | No | 0.78 | 0.017 | $9.39 \times 10^{-7}$ | Yes | – |
| ENSG00000278376 | – | No | 0.06 | 0.004 | $6.55 \times 10^{-7}$ | – | 62 |
| TRHDE-AS1 | – | No | 0.14 | 0.048 | $6.92 \times 10^{-7}$ | Yes | – |
| INSYN1-AS1 | – | No | 0.25 | 0.014 | $5.53 \times 10^{-8}$ | Yes | 25 |
| **Genes associated across all four methods** | | | | | | | |
| RERE | – | Yes | $3.48 \times 10^{-8}$ | $1.29 \times 10^{-9}$ | $7.35 \times 10^{-16}$ | No | 45 |
| NEGR1 | – | Yes | $2.31 \times 10^{-6}$ | $1.53 \times 10^{-7}$ | $1.30 \times 10^{-26}$ | No | – |
| ZNF638 | Cytidine | Yes | 0.003 | 0.004 | $5.64 \times 10^{-7}$ | No | 61 |
| RFTN2 | Lipopolysaccharide | Yes | 0.003 | $4.28 \times 10^{-4}$ | $1.52 \times 10^{-7}$ | No | 204 |
| ZNF445 | – | Yes | $3.35 \times 10^{-4}$ | 0.001 | $1.52 \times 10^{-10}$ | No | 138 |
| ZNF197 | – | Yes | $2.35 \times 10^{-4}$ | $8.04 \times 10^{-5}$ | $4.61 \times 10^{-10}$ | No | 138 |
| CCDC71 | – | Yes | $5.56 \times 10^{-5}$ | 0.039 | $3.12 \times 10^{-12}$ | Yes | 9 |
| ENSG00000225399 | – | Yes | 0.010 | 0.003 | $1.07 \times 10^{-8}$ | – | 9 |
| RHOA | Simvastatin, pravastatin, atorvastatin, magnesium, CCG-1423 | Yes | 0.031 | 0.019 | $1.45 \times 10^{-7}$ | No | 9 |
| CDH9 | Calcium | Yes | 0.003 | 0.002 | $2.17 \times 10^{-8}$ | No | 95 |
| TMEM161B | Crofelemer | Yes | $2.79 \times 10^{-5}$ | $6.2 \times 10^{-8}$ | $5.26 \times 10^{-9}$ | No | – |
| PFDN1 | – | Yes | 0.025 | $2.94 \times 10^{-4}$ | $5.60 \times 10^{-8}$ | Yes | 67 |
| SLC4A9 | Sodium bicarbonate | Yes | 0.029 | 0.002 | $2.25 \times 10^{-12}$ | No | 67 |
| HARS1 | Adenosine phosphate, pyrophosphate, phosphate, histidine | Yes | 0.024 | 0.017 | $5.29 \times 10^{-8}$ | Yes | 141 |
| HARS2 | Adenosine phosphate, pyrophosphate, phosphate, histidine | Yes | 0.019 | 0.044 | $2.32 \times 10^{-8}$ | No | 141 |
| ZMAT2 | – | Yes | 0.014 | 0.005 | $1.11 \times 10^{-9}$ | No | 141 |
| PCDHA1 | Calcium | Yes | 0.015 | 0.005 | $1.15 \times 10^{-8}$ | No | 141 |
| PCDHA2 | Calcium | Yes | 0.031 | 0.010 | $1.55 \times 10^{-8}$ | No | 141 |
| PCDHA3 | Calcium | Yes | 0.043 | 0.004 | $1.06 \times 10^{-8}$ | No | 141 |
| TMEM106B | Crofelemer | Yes | $2.79 \times 10^{-5}$ | $1.57 \times 10^{-8}$ | $4.87 \times 10^{-15}$ | No | 1 |
| ZDHHC21 | Coenzyme A, palmityl-CoA | Yes | 0.002 | 0.036 | $5.13 \times 10^{-7}$ | No | 42 |
| SORCS3 | – | Yes | $2.23 \times 10^{-13}$ | $1.28 \times 10^{-8}$ | $1.98 \times 10^{-10}$ | No | 16 |
| MYBPC3 | – | Yes | 0.004 | 0.012 | $9.23 \times 10^{-10}$ | No | 48 |
| SLC39A13 | Zinc chloride, zinc sulfate | Yes | 0.007 | 0.003 | $9.14 \times 10^{-7}$ | Yes | 48 |
| CTNND1 | – | Yes | 0.002 | 0.010 | $1.84 \times 10^{-7}$ | No | 60 |
| ANKK1 | Methadone, naltrexone, fostamatinib | Yes | 0.011 | $2.44 \times 10^{-6}$ | $1.41 \times 10^{-11}$ | No | – |
| DRD2 | Cabergoline, ropinirole, sulpiride | Yes | $9.11 \times 10^{-10}$ | $7.81 \times 10^{-10}$ | $3.95 \times 10^{-8}$ | No | – |
| MLEC | – | Yes | 0.013 | $1.32 \times 10^{-6}$ | $8.90 \times 10^{-7}$ | Yes | – |
| SPPL3 | – | Yes | $3.6 \times 10^{-7}$ | $1.47 \times 10^{-7}$ | $1.89 \times 10^{-12}$ | No | – |
| LRFN5 | – | Yes | $4.28 \times 10^{-9}$ | $2.7 \times 10^{-4}$ | $5.79 \times 10^{-15}$ | No | 10 |
| AREL1 | – | Yes | 0.007 | $5.64 \times 10^{-5}$ | $7.24 \times 10^{-12}$ | No | 143 |

**Table (continued) 1 | Genes associated with MD**

| Gene[a] | Drug[b] | FUMA[c] | MAGMA[d] | Hi-C MAGMA[d] | TWAS P | Novel[e] | Credible set[f] |
|---|---|---|---|---|---|---|---|
| DLST | Lipoic acid succinyl-CoA, coenzyme A, dihydrolipoamide (S)–succinyldihydrolipoamide | Yes | $2.2×10^{-4}$ | 0.001 | $1.51×10^{-9}$ | No | 143 |
| MARK3 | Fostamatinib, alsterpaullone | Yes | $1.43×10^{-4}$ | $2.34×10^{-5}$ | $3.50×10^{-9}$ | No | 11 |
| KLC1 | Fluorouracil, irinotecan, leucovorin | Yes | $4.91×10^{-9}$ | $9.99×10^{-8}$ | $1.26×10^{-12}$ | No | 11 |
| XRCC3 | Fluorouracil, irinotecan, leucovorin | Yes | 0.004 | $7.73×10^{-6}$ | $3.49×10^{-10}$ | No | 11 |
| ZFYVE21 | Inositol 1,3-bisphosphate | Yes | $3.0×10^{-5}$ | $1.21×10^{-7}$ | $2.83×10^{-13}$ | Yes | 11 |
| CELF4 | Iloperidone | Yes | $1.52×10^{-8}$ | $3.59×10^{-5}$ | $9.66×10^{-9}$ | No | 8 |
| RAB27B | Guanosine-5′-diphosphate | Yes | $1.23×10^{-6}$ | 0.042 | $5.61×10^{-10}$ | No | – |
| EMILIN3 | – | Yes | 0.039 | 0.001 | $6.66×10^{-8}$ | No | 64 |
| CHD6 | Phosphate, ATP, ADP | Yes | 0.001 | 0.001 | $1.76×10^{-10}$ | No | 64 |
| EP300 | Acetyl-CoA, TGF-β, garcinol, cyclic AMP, curcumin, mocetinostat | Yes | $4.95×10^{-4}$ | $1.87×10^{-6}$ | $1.71×10^{-9}$ | No | 16 |
| RANGAP1 | – | Yes | $1.82×10^{-4}$ | 0.003 | $6.73×10^{-9}$ | No | 16 |
| ZC3H7B | – | Yes | 0.001 | $2.03×10^{-6}$ | $1.37×10^{-9}$ | No | 16 |

This table includes 12 genes significantly associated in the TWAS and Hi-C MAGMA, that is, not in physical proximity to a GWAS hit, and 43 genes significantly associated across all four methods (TWAS, FUMA, MAGMA and Hi-C MAGMA). [a]Ensembl IDs are shown for genes without symbol names. [b]Drugs targeting the prioritized genes or genes of the same family from GeneCards, DrugBank and ChEmbl. [c]Gene mapped by FUMA positional mapping or eQTL mapping. [d]Bonferroni adjusted two-sided $P$ value for MAGMA or Hi-C MAGMA of $z$ statistics ($P < 0.05$ implies statistical significance). [e]Novel report as compared with previous MAGMA and TWAS on MD. [f]Number of variants in the 99% credible set, only available for mapped loci from multi-ancestry fine-mapping.

$-0.032$, $P = 9.22 × 10^{-3}$). Moreover, MD genetic liability showed a positive causal association with systolic blood pressure (SBP, $\beta = 0.034$, 95% CI 0.009 to 0.059, $P = 7.66 × 10^{-3}$). In samples of African ancestry, SBP had a positive causal association with MD ($\beta = 0.080$, 95% CI 0.026 to 0.133, $P = 3.43 × 10^{-3}$).

## Discussion

We present the first large-scale GWAS of MD in an ancestrally diverse sample, including data from almost 1 million participants of African, East Asian and South Asian ancestry, and Hispanic/Latin American samples. The largest previous report included 26,000 cases of African ancestry[15].

By aggregating data in ancestry-specific meta-analyses, we identified two novel loci, 2q24.2 and 6p15. In the Hispanic/Latin American group, variants at 2q24.2 were associated with MD. Most of the cases in this group were defined using symptoms questionnaires. Future studies will be required to assess whether the association of this loci with clinical MD is consistent with our estimate. While the additional association at 6q16 in the GWAS in samples of African ancestry requires further confirmation in future studies, the link with MD is biologically plausible. The lead variant was significantly associated with the expression of *MCHR2* specifically in brain cortex tissue. Melanin-concentrating hormone (MCH) is a neuropeptide that is expressed in the central and peripheral nervous systems. It acts as a neurotransmitter and neuromodulator in a broad array of neuronal functions directed toward the regulation of goal-directed behavior, such as food intake, and general arousal[34].

The diversity, in combination with the large sample size, enabled a comparison of the causal genetic architecture across ancestry groups. We assessed to what extent the 206 previously identified loci from large European ancestry discovery GWAS were transferable to other ancestry groups. Differences in allele frequencies, linkage disequilibrium and variable sample sizes impact on power to observe associations for each group. We recently developed PAT ratios, an approach to account for all these factors by comparing observed transferable loci with what is expected for a study of a given ancestry and sample size[28]. The PAT ratios were about 30% for African, South Asian and East Asian ancestry, remarkably similar and consistently low. We previously computed PAT

ratios for several other traits and found variation between traits, but the estimates for MD were at the bottom[19,28]. With a PAT ratio of 64%, the transferability of MD loci discovered in European ancestry samples was much higher for the Hispanic/Latin American group. This finding may reflect that the Hispanic/Latin American group contained many participants with a high proportion of European ancestry[35,36]. The majority of cases within this group were defined via symptom questionnaires rather than clinical MD. Hence, it may be possible that the transferability for clinical MD is even higher in this group. For African, South Asian and East Asian ancestry, the PAT ratios for clinical MD were all below 0.5 and consistent with the estimates from the main analysis, demonstrating that heterogeneity in outcome definitions does not explain the limited transferability of MD loci across ancestry groups.

To better understand mechanisms underlying individual differences in vulnerability to development of MD, we need to bridge the gap from locus discovery to the identification of target genes. Our study achieved substantial progress in this respect. Fine mapping benefitted from the additional diverse samples, with median credible sets reduced from 65.6 to 35 in size and with 32 loci resolved to ≤10 putatively causal SNPs (11 loci from the European ancestry fine mapping).

On the TWAS, the expression of 354 genes was significantly associated with MD. Out of these, 205 gene associations were novel, and 89 were overlapping with results of the largest previously published MD TWAS[15]. Furthermore, 80 genes were overlapping with associations from another, previously published, large MD TWAS with largely overlapping samples of European ancestry[33]. A number of these TWAS features, including *NEGR1*, *ESR2* and *TMEM106B*, were previously also fine mapped and highlighted as putatively causal in previous post-TWAS analyses, strengthening the role of TWAS as an important tool to better understand the relationship between gene expression and MD.

Through TWAS and three other tools that incorporate the growing body of knowledge about functional annotations of the genome, we classified 43 genes as 'high confidence'. The definition admittedly remains arbitrary until the field establishes clear guidelines. Nevertheless, the high-confidence list represents an evidence-based starting point for further follow-up. It provides confirmation for several genes that have repeatedly been highlighted as being near a GWAS-associated

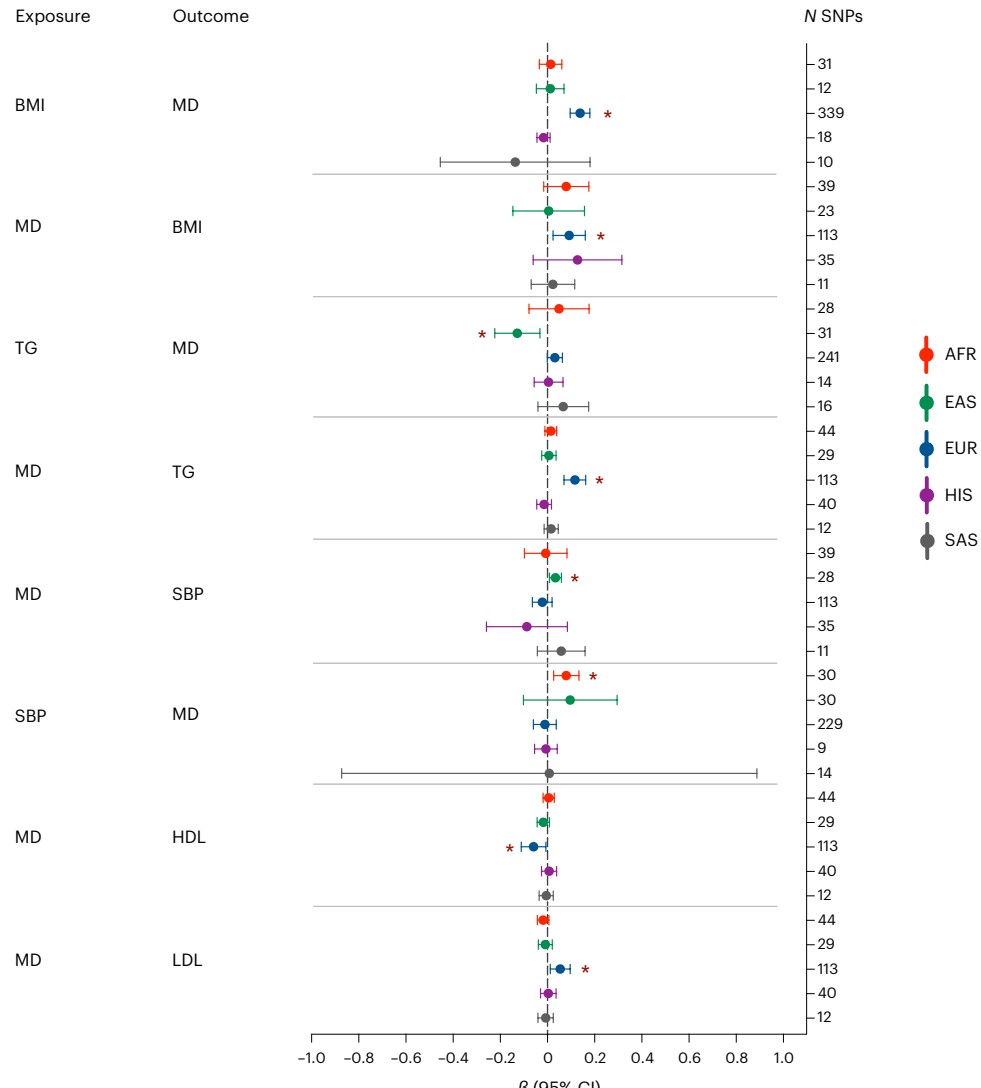

**Fig. 6 | Bi-directional MR tests between MD and cardiometabolic outcomes.** The data are presented with a $\beta$ and a 95% CI. Nominally significant associations are marked with a red asterisk. Statistics have been derived using the $\beta$ and standard errors for the number of variants used as IVs in each analysis, shown as $N$ SNPs. Results are not shown for diastolic blood pressure for which there were no significant associations. *$P < 0.05$ ($P$ values in order from top to bottom: $6.88 \times 10^{-11}$, $8.22 \times 10^{-3}$, $9.22 \times 10^{-3}$, $7.93 \times 10^{-7}$, $7.67 \times 10^{-3}$, $3.43 \times 10^{-3}$, $0.03$ and $0.01$). More details can be found in Supplementary Table 16. AFR, African ancestry; EAS, East Asian ancestry; EUR, European ancestry; HIS, Hispanic/Latin American group; SAS, South Asian ancestry.

variant and having high biological plausibility[2,13–15,33]: *NEGR1*, *DRD2*, *CELF4*, *LRFN5*, *TMEM161B* and *TMEM106B*.

Furthermore, cadherin-9 (*CDH9*) and protocadherins (*PCDHA1*, *PCDHA2* and *PCDHA3*) were classified as high-confidence genes. Cadherins are transmembrane proteins, mediating adhesion between cells and tissues in organisms[37]. In previous studies, cadherins have been linked with MD and with other disorders involving the brain, including late-onset Alzheimer's disease, which often manifests as neuropsychiatric symptoms coupled with depression and anxiety[13,38–40]. The results of our study strengthen the evidence for the involvement of cadherins and protocadherins in the etiology of MD.

Genes newly implicated in MD development in our study highlight novel pathways, pinpoint potential new drug targets and suggest opportunities for drug repurposing. *NDUFAF3* encodes mitochondrial complex I assembly protein, which is the main target of the drug metformin[41], the first-line drug for treating type 2 diabetes. Research in model organisms has provided a tentative link between metformin and a reduction in depression and anxiety[42]. Furthermore, a recent study using more than 360,000 samples from the United Kingdom Biobank

(UKB) found associations between *NDUFAF3* and mood instability, suggesting that energy dysregulation may play an important role in the physiology of mood instability[43].

Previous MR studies conducted in populations of European ancestry suggested a causal relationship of higher BMI increasing the odds of depression[44–46]. To our knowledge, evidence of a reverse causal association (that is, MD genetic liability increases the odds of higher BMI) has not been previously reported[2]. We also observed that the genetic liability to MD was associated with higher TG levels, lower HDL cholesterol and higher LDL cholesterol levels in individuals of European ancestry, which were not significant in the only previous MR study of smaller statistical power[47]. Individuals with depression present higher levels of inflammation and are at increased risk of cardiometabolic disorders, irrespective of the age of onset[48]. The phenotypic associations between MD and cardiometabolic traits may partly reflect the genetic overlap between them[49]. However, in other ancestry groups, no significant relationship between BMI and MD was observed. Our MR analyses showed an effect of reduced TGs on increasing odds of MD in participants of East Asian ancestry.

Therefore, we provide further evidence for an opposite direction of effect for the relationship between MD and metabolic traits in European and East Asian ancestry groups[19,21]. Instead of generalizing findings about depression risk factors across populations, further studies are needed to understand how genetic and environmental factors contribute to the complex relationships across diverse ancestry groups.

Our study has limitations. In this study, we assigned individuals into ancestry and ethnic groups. While this enabled important insights (for example, about transferability of MD loci), such categorical assignments are imprecise and some participants with admixed ancestry may still get excluded. In future research, we aim to implement different analytic strategies that are fully inclusive.

The sample size varied greatly across ancestry groups. The smallest group were individuals of South Asian ancestry. Most of the individuals included in our study live in the United States or in the United Kingdom. To characterize MD in global populations, future studies prioritizing primary data collection are needed. To contribute to this, we are currently recruiting MD patients and controls from Pakistan into the DIVERGE study[50]. However, a concerted global effort to increase diversity in genetics will be necessary to fully address the issue[22]. This also applies to the lack of other omics data and other functional databases to support downstream analyses for ancestrally diverse GWAS, such as large resources for transcriptomics or proteomics in relevant tissues[51,52]. This may have impacted on our TWAS results because the RNA sequencing data was predominantly from participants of European ancestry.

Furthermore, statistical power for discovery of genetic associations may be impacted by reduced coverage of genetic variation present in diverse ancestral groups, as well as other factors such as the reliability of outcome assessment across different groups.

Additionally, our bi-directional MR analysis tested the relationships between MD and cardiometabolic traits. When testing MD as the exposure, the results should be interpreted as the effect of MD genetic liability and not as the effect of MD itself.

This study utilized data from several existing cohorts and bioresources to achieve large sample sizes. This necessitated using different outcome definitions, covering self-administered symptom questionnaires, electronic healthcare records and structured clinical interviews. The potential advantages and disadvantages of these approaches have been extensively discussed in previous studies[2,6]. It is possible that some of the 190 genome-wide significant loci we identified are linked to a more general susceptibility to mental illness instead of being specific to MD. However, given the overlap between different psychiatric disorders[53], such findings are nevertheless of value for our understanding of the biology and for the development of new treatments for MD.

In conclusion, in this first large-scale, multi-ancestry GWAS of MD, we demonstrated through transferability analyses that a notable proportion of MD loci are specific to samples of European ancestry. We identified novel, biologically plausible associations that were missed in European ancestry analyses and demonstrated that large, diverse samples can be important for identifying target genes and putative mechanisms. These findings suggest that for MD, a heterogeneous condition with highly complex etiology, increasing ancestral as well as global diversity in genetic studies may be particularly important to ensure discovery of core genes and to inform about transferability of findings across ancestry groups.

## Online content

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

Xiangrui Meng [1,80], Georgina Navoly [2,80], Olga Giannakopoulou [1,80], Daniel F. Levey [3,4], Dora Koller [3,4,5], Gita A. Pathak [3,4], Nastassja Koen [6], Kuang Lin [7], Mark J. Adams [8], Miguel E. Rentería [9], Yanzhe Feng [1], J. Michael Gaziano [10,11,12], Dan J. Stein [6], Heather J. Zar [13], Megan L. Campbell [14], David A. van Heel [15], Bhavi Trivedi [15], Sarah Finer [16], Andrew McQuillin [1], Nick Bass [1], V. Kartik Chundru [17], Hilary C. Martin [17], Qin Qin Huang [17], Maria Valkovskaya [1], Chia-Yi Chu [1], Susan Kanjira [8], Po-Hsiu Kuo [18,19], Hsi-Chung Chen [19,20], Shih-Jen Tsai [21,22], Yu-Li Liu [23], Kenneth S. Kendler [24], Roseann E. Peterson [24,25], Na Cai [26,27,28], Yu Fang [29], Srijan Sen [29,30], Laura J. Scott [31,32], Margit Burmeister [29,30,33,34], Ruth J. F. Loos [35,36], Michael H. Preuss [35], Ky'Era V. Actkins [37], Lea K. Davis [37,38,39], Monica Uddin [40], Agaz H. Wani [40], Derek E. Wildman [41], Allison E. Aiello [42], Robert J. Ursano [43], Ronald C. Kessler [44], Masahiro Kanai [45,46,47], Yukinori Okada [45,48,49], Saori Sakaue [45,47,50], Jill A. Rabinowitz [51], Brion S. Maher [51], George Uhl [52], William Eaton [51], Carlos S. Cruz-Fuentes [53], Gabriela A. Martinez-Levy [53], Adrian I. Campos [9,54], Iona Y. Millwood [7,55], Zhengming Chen [7,55], Liming Li [56,57,58], Sylvia Wassertheil-Smoller [59], Yunxuan Jiang [60,61], Chao Tian [61], Nicholas G. Martin [62], Brittany L. Mitchell [62], Enda M. Byrne [63], Swapnil Awasthi [64,65], Jonathan R. I. Coleman [66], Stephan Ripke [64,65,67], PGC-MDD Working Group[*,**], China Kadoorie Biobank Collaborative Group[*,**], the 23andMe Research Team[*,**], Genes and Health Research Team[*,**], BioBank Japan Project[*,**], Tamar Sofer [68,69], Robin G. Walters [7,55], Andrew M. McIntosh [8,70], Renato Polimanti [3,4,71], Erin C. Dunn [72,73,74], Murray B. Stein [75,76,77], Joel Gelernter [3,4,78], Cathryn M. Lewis [66,79] & Karoline Kuchenbaecker [1,2] ✉

[1]Division of Psychiatry, UCL, London, UK. [2]UCL Genetics Institute, UCL, London, UK. [3]Department of Psychiatry, VA CT Healthcare Center, West Haven, CT, USA. [4]Department of Psychiatry, Yale University School of Medicine, New Haven, CT, USA. [5]Department of Genetics, Microbiology and Statistics, University of Barcelona, Barcelona, Spain. [6]SAMRC Unit on Risk and Resilience in Mental Disorders, Department of Psychiatry and Neuroscience Institute, University of Cape Town, Cape Town, South Africa. [7]Nuffield Department of Population Health, University of Oxford, Oxford, UK. [8]Division of Psychiatry, University of Edinburgh, Edinburgh, UK. [9]Department of Genetics and Computational Biology, QIMR Berghofer Medical Research Institute, Brisbane, Queensland, Australia. [10]Department of Medicine, VA Boston Healthcare System, Boston, MA, USA. [11]Department of Medicine, Brigham and Women's Hospital, Boston, MA, USA. [12]Department of Medicine, Harvard Medical School, Boston, MA, USA. [13]SAMRC Unit on Child and Adolescent Health, Department of Paediatrics and Child Health, University of Cape Town, Cape Town, South Africa. [14]Department of Psychiatry and Neuroscience Institute, University of Cape Town, Cape Town, South Africa. [15]Blizard Institute, Queen Mary University of London, London, UK. [16]Wolfson Institute of Population Health, Queen Mary University of London, London, UK. [17]Wellcome Sanger Institute, Saffron Walden, UK. [18]Department of Public Health and Institute of Epidemiology and Preventive Medicine, National Taiwan University, Taipei, Taiwan. [19]Department of Psychiatry, National Taiwan University Hospital, Taipei, Taiwan. [20]Center of Sleep Disorders, National Taiwan University Hospital, Taipei, Taiwan. [21]Institute of Brain Science and Division of Psychiatry, National Yang-Ming Chiao Tung University, Taipei, Taiwan. [22]Department of Psychiatry, Taipei Veterans General Hospital, Taipei, Taiwan. [23]Center for Neuropsychiatric Research, National Health Research Institutes, Miaoli County, Taiwan. [24]Department of Psychiatry, VCU, Richmond, VA, USA. [25]Department of Psychiatry, SUNY Downstate Health Sciences University, Brooklyn, NY, USA. [26]Helmholtz Pioneer Campus, Helmholtz Munich, Neuherberg, Germany. [27]Computational Health Centre, Helmholtz Munich, Neuherberg, Germany. [28]Department of Medicine, Technical University of Munich, Munich, Germany. [29]Michigan Neuroscience Institute, University of Michigan, Ann Arbor, MI, USA. [30]Department of Psychiatry, University of Michigan, Ann Arbor, MI, USA. [31]Department of Biostatistics, University of Michigan, Ann Arbor, MI, USA. [32]Center for Statistical Genetics, University of Michigan, Ann Arbor, MI, USA. [33]Department of Computational Medicine and Bioinformatics, University of Michigan, Ann Arbor, MI, USA. [34]Department of Human Genetics, University of Michigan, Ann Arbor, MI, USA. [35]Charles Bronfman Institute for Personalized Medicine, Icahn School of Medicine at Mount Sinai, New York, NY, USA. [36]Novo Nordisk Foundation Center for Basic Metabolic Research, Faculty of Health and Medical Sciences, University of Copenhagen, Copenhagen, Denmark. [37]Department of Medicine, Division of Genetic Medicine, Vanderbilt University Medical Center, Nashville, TN, USA. [38]Department of Biomedical Informatics, Vanderbilt University Medical Center, Nashville, TN, USA. [39]Department of Psychiatry and Behavioral Sciences, Vanderbilt University Medical Center, Nashville, TN, USA. [40]College of Public Health, University of South Florida, Tampa, FL, USA. [41]Genomics Program, College of Public Health, University of South Florida, Tampa, FL, USA. [42]Robert N. Butler Columbia Aging Center, Department of Epidemiology, Mailman School of Public Health, Columbia University, New York, NY, USA. [43]Department of Psychiatry, Uniformed Services University of the Health Sciences, Bethesda, MD, USA. [44]Department of Health Care Policy, Harvard Medical School, Boston, MA, USA. [45]Department of Statistical Genetics, Osaka University Graduate School of Medicine, Osaka, Japan. [46]Analytic and Translational Genetics Unit, Massachusetts General Hospital, Boston, MA, USA. [47]Program in Medical and Population Genetics, Broad Institute of Harvard and MIT, Cambridge, MA, USA. [48]Department of Genome Informatics, Graduate School of Medicine, University of Tokyo, Tokyo, Japan. [49]Laboratory for Systems Genetics, RIKEN Center for Integrative Medical Sciences, Yokohama, Japan. [50]Divisions of Genetics and Rheumatology, Department of Medicine, Brigham and Women's Hospital, Harvard Medical School, Boston, MA, USA. [51]Department of Mental Health, Johns Hopkins Bloomberg School of Public Health, Baltimore, MD, USA. [52]Neurology and Pharmacology, University of Maryland, Maryland VA Healthcare System, Baltimore, MD, USA. [53]Departamento de Genética, Instituto Nacional de Psiquiatría 'Ramón de la Fuente Muñíz', Mexico City, Mexico. [54]Institute for Molecular Bioscience, The University of Queensland, Brisbane, Queensland, Australia. [55]MRC Population Health Research Unit, University of Oxford, Oxford, UK. [56]Department of Epidemiology and Biostatistics, School of Public Health, Peking University, Beijing, China. [57]Peking University Center for Public Health and Epidemic Preparedness and Response, Peking University, Beijing, China. [58]Key Laboratory of Epidemiology of Major Diseases (Peking University), Ministry of Education, Beijing, China. [59]Department of Epidemiology and Population Health, Albert Einstein College of Medicine, Bronx, NY, USA. [60]Department of Biostatistics, Emory University, Atlanta, GA, USA. [61]23andMe, Inc., Mountain View, CA, USA. [62]Mental Health and Neuroscience Research Program, QIMR Berghofer Medical Research Institute, Brisbane, Queensland, Australia. [63]Child Health Research Centre, The University of Queensland, Brisbane, Queensland, Australia. [64]Stanley Center for Psychiatric Research, Broad Institute of Harvard and MIT, Cambridge, MA, USA. [65]Department of Psychiatry and Psychotherapy, Charité – Universitätsmedizin, Berlin, Germany. [66]Social Genetic and Developmental Psychiatry Centre, Institute of Psychiatry, Psychology and Neuroscience, King's College London, London, UK. [67]Analytic and Translational

Genetics Unit, Massachusetts General Hospital, Cambridge, MA, USA. [68]Division of Sleep and Circadian Disorders, Department of Medicine, Brigham and Women's Hospital, Boston, MA, USA. [69]Department of Biostatistics, Harvard T.H. Chan School of Public Health, Boston, MA, USA. [70]Institute for Genomics and Cancer, University of Edinburgh, Edinburgh, UK. [71]VA Connecticut Healthcare Center, West Haven, CT, USA. [72]Department of Psychiatry, Massachusetts General Hospital, Boston, MA, USA. [73]Psychiatric and Neurodevelopmental Genetics Unit (PNGU), Massachusetts General Hospital, Boston, MA, USA. [74]Stanley Center for Psychiatric Research, Broad Institute, Cambridge, MA, USA. [75]Department of Psychiatry, UC San Diego School of Medicine, La Jolla, CA, USA. [76]Herbert Wertheim School of Public Health and Human Longevity, University of California San Diego, La Jolla, CA, USA. [77]Psychiatry Service, VA San Diego Healthcare System, San Diego, CA, USA. [78]Department of Genetics, Yale University School of Medicine, New Haven, CT, USA. [79]Department of Medical and Molecular Genetics, King's College London, London, UK. [80]These authors contributed equally: Xiangrui Meng, Georgina Navoly, Olga Giannakopoulou. *Lists of authors and their affiliations appear at the end of the paper. ✉e-mail: k.kuchenbaecker@ucl.ac.uk

## PGC-MDD Working Group

Xiangrui Meng[1,80], Mark J. Adams[8], Kenneth S. Kendler[24], Roseann E. Peterson[24,25], Na Cai[26,27,28], Nicholas G. Martin[62], Enda M. Byrne[63], Jonathan R. I. Coleman[66], Stephan Ripke[64,65,67], Andrew M. McIntosh[8,70], Erin C. Dunn[72,73,74], Cathryn M. Lewis[66,79] & Karoline Kuchenbaecker[1,2]

A full lists of members and their affiliations appear in the Supplementary Information.

## China Kadoorie Biobank Collaborative Group

Kuang Lin[7], Iona Y. Millwood[7,55], Zhengming Chen[7,55], Liming Li[56,57,58] & Robin G. Walters[7,55]

A full lists of members and their affiliations appear in the Supplementary Information.

## the 23andMe Research Team

Yunxuan Jiang[60,61] & Chao Tian[61]

A full lists of members and their affiliations appear in the Supplementary Information.

## Genes and Health Research Team

David A. van Heel[15], Bhavi Trivedi[15], Sarah Finer[16], Nick Bass[1] & Qin Qin Huang[17]

A full lists of members and their affiliations appear in the Supplementary Information.

## BioBank Japan Project

Masahiro Kanai[45,46,47], Yukinori Okada[45,48,49] & Saori Sakaue[45,47,50]

A full lists of members and their affiliations appear in the Supplementary Information.

## Methods

### Participating cohorts

For the analyses of the African, East Asian, South Asian and Hispanic/Latin American group, we included data from 21 cohorts (Supplementary Table 1) with ancestrally diverse participants, where measurements were taken from distinct samples. Details including study design, genotyping and imputation methods and quality control for these studies had been described by previous publications (Supplementary Note). All participants provided informed consent. All studies obtained ethical approvals from local ethics review boards. Measures were taken from distinct samples rather than repeat measures from the same individual.

For each study, a principal component analysis was carried out based on the genetic similarity of individuals. Individuals who clustered around a reference group with confirmed ancestry were assigned to that specific group and included in the association analysis, except for the Hispanic/Latin American group, which was based on self-reported ethnicity. Individuals with admixture between the predefined ancestry reference groups were excluded.

We also included two previously published studies of MD, using data from ancestrally European participants, including the PGC-MDD2 ($n_{cases}$ = 246,241 and $n_{controls}$ = 558,568)[13] and the AGDS ($n_{cases}$ = 12,123 and $n_{controls}$ = 12,684) (ref. 14) to conduct a multi-ancestry meta-analysis of MD (Supplementary Table 1). The total sample size of the multi-ancestry meta-analysis was 1,815,091 ($n_{cases}$ = 345,389 and $n_{effhalf}$ = 559,332). Of the participants, 70.1% (effective sample size) were of European ancestry, 8.2% East Asian, 11.8% African and 1.5% were of South Asian ancestry, and 7.9% were Hispanic/Latin American.

We used a range of measures to define depression, including structured clinical interviews, medical care records, symptom questionnaires and self-reported surveys (Supplementary Table 1). The meta-analyses were primarily conducted combining GWASs of all phenotype definitions (that is, broad MD). In addition, meta-analyses for clinical depression and relevant downstream analyses were also conducted. We considered depression ascertained by structured clinical interviews (directly assessing diagnostic criteria based on DSM-IV, DSM-5 or ICD9/10 through interviews or self-report) or medical care records (ICD9 or ICD10 from primary or secondary care units) as clinical depression. Among the GWASs, the Genes and Health study, MVP, the Genetic Epidemiology Research on Aging Study (GERA), BioVU, the Prevention Intervention Research Center First Generation Trial (PIRC), the Mexican Adolescent Mental Health Survey (MAMHS), CONVERGE, the UKB, the Army Study to Assess Risk and Resilience in Service Members (Army-STARRS) and BioMe fulfilled the clinical definition of depression. On the basis of European ancestry data from previous published work of the PGC-MDD group[13], all studies fulfilled the clinical definition, except for the UKB and the 23andMe, which were excluded in the analysis of clinical MD.

### Study-level genetic association analyses

Throughout the manuscript, all statistical tests were two sided, unless explicitly indicated. We had access to individual-level data for Army STARRS, UKB, Women's Health Initiative (WHI), Intern Health Study (IHS), GERA, Jackson Heart Study (JHS), Drakenstein Child Health Study and the Detroit Neighborhood Health Study. Data access was granted via our collaborators, the UKB under application ID 51119 and the dbGaP under project ID 18933.

SNP-level associations with depression were assessed through logistic regressions using PLINK version 1.90. The additive per-allele model was employed. Age, sex, principal components and other relevant study-level covariates were included as covariates. Where available, genotypes on chromosome X were coded 0 or 2 in male participants and 0, 1 and 2 in female participants. Data for variants on X were only available for some of the studies (Supplementary Table 1). The effective sample size was 1,763 for African, 58,833 East Asian, 13,099 South Asian ancestry and 79,720 for Hispanic/Latin American. Summary statistics were received from our collaborators for all other studies. Additive-effect logistic regressions were conducted by the 23andMe Inc, Taiwan-MDD study, MVP, BBJ, Rabinowitz, MAMHS, PrOMIS and BioVU. Age, sex, principal components and other relevant study-level covariates were included as covariates.

Mixed-effect models were used in the association analysis for CKB, BioME and Genes and Health with SAIGE (version 0.36.1) (ref. 54). The CONVERGE study initially conducted mixed-effect model GWA tests with Bayesian and logistic regression toolkit–linear mixed model (BOLT-LMM), followed by PLINK logistic regressions to retrieve log odds ratios (ORs). For the CONVERGE study, the logORs and s.e.m. from PLINK were used in our meta-analysis. The HCHS/SOL implemented mixed-effect model GWA tests to adjust for population structure and relatedness with depression as binary outcome[16] and was conducted using GENESIS[55]. The summary statistics from GENESIS were converted into logOR and s.e.m. before meta-analysis. First, the score and its variance were transformed into $\beta$ and s.e.m. by $\beta$ = score/variance and s.e.m. = sqrt(variance)/variance. Afterwards, $\beta$ and s.e.m. were converted into approximate logOR and s.e.m. using $\beta = \beta/(pi \times (1 - pi))$ and s.e.m. = s.e.m./(pi × (1 − pi)), where pi is the proportion of cases in analysis[56].

We restricted the downstream analysis to variants with imputation accuracy info score of 0.7 or higher and effective allele count ($2 \times MAF \times (1 - MAF) \times N \times R^2$) of 50 or higher. For study of small sample size, we required a minor allele frequency of no less than 0.05. The alleles for indels were re-coded as 'I' for the longer allele and 'D' for the shorter one. Indels of different patterns at the same position were removed.

### Meta-analyses

We first implemented inverse variance-weighted (IVW) fixed-effect meta-analyses for GWAS from each ancestry/ethnic group (that is, African ancestry, East Asian ancestry, South Asian ancestry and the Hispanic/Latin American group) using METAL (version 2011-03-25) (ref. 57). The genomic inflation factor $\lambda$ was calculated for each study and meta-analysis with R package GenABEL version 1.8.0 (ref. 58). Given the dependence of this estimate on sample size, we also calculated $\lambda_{1,000}$ (ref. 59) as $\lambda_{1,000} = 1 + (\lambda - 1) \times (1/n_{case} + 1/n_{control}) \times 500$ (ref. 60). The LDSC intercept was also calculated with an ancestry-matched LD reference panel from the Pan UKB reference panel[61] for each meta-analysis with LDSC (version 1.0.1) (ref. 62). For meta-analyses with residual inflation ($\lambda > 1.1$), test statistics for variants were adjusted by LDSC intercept. Following the meta-analyses by METAL, variants present in less than two studies were filtered out. Statistical tests were generally two sided unless otherwise stated. We also performed a heterogeneity analysis with METAL to assess whether observed effect sizes (or test statistics) are homogeneous across samples.

We combined data from 71 cohorts with diverse ancestry using an IVW fixed-effects meta-analysis in METAL[57]. $\lambda$ and $\lambda_{1,000}$ were calculated, and were 1.687 and 1.001, respectively. The LDSC intercept was also calculated with the multi-ancestry LD reference panel (Supplementary Note), which was 1.019 (s.e.m. 0.011). We adjusted the test statistics from the multi-ancestry meta-analysis using the LDSC intercept of 1.019. Only variants present in at least two studies were retained for further analysis, yielding a total of 22,941,580 variants. We also calculated the number of cases and the total number of samples for each variant based on the crude sample size and availability of each study.

We used a significance threshold of $5 \times 10^{-8}$. To identify independent association signals, the GCTA forward selection and backward elimination process (command 'cojo-slt') were applied using the summary statistics from the multi-ancestry meta-analysis, with the aforementioned multi ancestry LD reference panel (GCTA version 1.92.0 beta2)[63,64]. It is possible that the algorithm identifies false positive secondary signals if the LD in the reference set does not match the actual LD in the GWAS data well; therefore, for each independent signal

defined by the GCTA algorithm, locus zoom plots were generated for the 250 kb upstream and downstream region. We then inspected each of these plots manually and removed any secondary signals from our list where there was unclear LD separation, that is, some of the variants close to the secondary hit were in LD with the lead variant.

Loci were defined by the flanking genomic interval mapping 250 kb upstream and downstream of each lead SNP. Where lead SNPs were separated by less than 500 kb, the corresponding loci were aggregated as a single locus with multiple independent signals. The lead SNP for each locus was then selected as the SNP with minimum association *P* value. The analysis for loci identification, along with all other R-related tasks unless otherwise stated, was conducted using R (version 3.4.3) (ref. [65]) and figures were produced using the packages ggplot2 (version 3.2.1) (ref. [66]), qqman (version 0.1.4) (ref. [67]) and ggpubr (version 0.6.0) (ref. [68]).

We conducted sensitivity analyses for outcome definitions, case–control ratio and using a different multi-ancestry meta-analysis approach (Supplementary Note).

## Fine mapping
We fine mapped all loci with statistically significant associations from the multi-ancestry GWAS using a statistical fine-mapping method for multi-ancestry samples[30]. Briefly, this method is an extension of a Bayesian fine-mapping approach[30,69] that utilizes estimates of the heterogeneity across ancestry groups, such that variants with different effect estimates across populations have a smaller prior probability to be the causal variant.

For each lead variant, we first extracted all nearby variants with $r^2 > 0.1$ as determined by the multi-ancestry LD reference. The multi-ancestry prior for each variant to be causal was calculated from a fixed-effects meta-analysis combining the summary statistics from ancestry-specific meta-analysis for each of the five major ancestry groups. $I^2$ statistics were calculated to estimate the heterogeneity of the effect estimates across ancestry groups. The posterior probability for a variant to be included in the credible set was proportional to its chi-square test statistic and the prior. The 99% credible set for each lead variant was determined by ranking all SNPs (within $r^2 > 0.1$ of the lead variant) according to their posterior probabilities and then including ranked SNPs until their cumulative posterior probabilities reached or exceeded 0.99.

As a comparison, we also conducted a Bayesian fine-mapping analysis based on the summary statistics of the European-ancestry meta-analysis. The same list of independent lead SNPs from the multi-ancestry meta-analysis were used for this fine mapping in the European ancestry data. All nearby SNPs with $r^2 > 0.1$ as determined by the 1,000 Genomes European LD reference panel were included in the fine mapping. The posterior probability was calculated in a similar way, but without the multi-ancestry prior. Similar to the multi-ancestry fine mapping, all SNPs were ranked, and 99% of the credible sets were derived accordingly.

Since our fine mapping was based on meta-analysis summary statistics, heterogeneity of individual studies (for example, due to differences in genotyping array) can influence the fine-mapping calibration and recall. We used a novel summary statistics-based quality control method proposed by Kanai and colleagues (SLALOM) to dissect outliers in association statistics for each fine-mapped locus[70]. This method calculates test statistics (DENTIST-S) from *Z*-scores of test variants and the lead variant (the variant of the lowest *P* value in each locus), and the LD *r* between test variants and the lead variant in the locus[71]. Among the 155 fine-mapped loci in our study, there were 134 loci with the largest variant posterior inclusion probability of greater than 0.1. For these 134 loci, *r* values were calculated for all variants within the 1 Mb region of the lead variant for each locus based on our multi-ancestry LD reference from the UKB data. In line with the criteria used by Kanai and colleagues, variants with DENTIST-S *P* value smaller than $1 \times 10^{-4}$ and $r^2$ with the lead variant greater than 0.6 were defined as outliers. Fine-mapped loci were classified as robust if there were no outlying variants.

## Colocalization analysis
We performed colocalization between genetic associations with MD and gene expression in brain and blood tissues from samples of European and African ancestry and Hispanic/Latin American participants using coloc R package[72]. To select genes for testing, we mapped SNPs within a 3 Mb window at 2q24.2 and 6q16.2 using Variant Effect Predictor[31], resulting in eight and four genes, respectively. Loci with posterior probability >90% either for both traits are associated and share two different but linked variants (H3 hypothesis) or a single causal variant (H4 hypothesis) were considered as colocalized. The European and African ancestry summary statistics for MD were tested against multi-ancestry brain eQTLs from European and African American samples[27]. For the Hispanic/Latin American group, we tested gene and protein expression of blood tissue from Multi-Ethnic Study of Atherosclerosis and Trans-omics for Precision Medicine[73]. For African ancestry, we tested gene expression of blood from GENOA study[74] and proteome expression of blood[75]. For European ancestry, we tested gene expression of blood from eQTLgen[76], and proteome expression from blood[75]. We also carried out ancestry-specific eQTL and pQTL colocalization analyses for previously reported loci that were or were not transferable.

## Assessment of transferability of MD-associated loci
We assessed whether published MD-associated loci display evidence of association in the East Asian, South Asian and African ancestry and Hispanic/Latin American samples. Pooling the independent genome-wide significant SNPs from two large GWAS of MD in samples of European ancestry yielded 195 loci[13–15]. The ancestrally diverse groups included in this study had smaller numbers of participants than the European ancestry discovery studies. Also, a given variant may be less frequent in another ancestry group. Therefore, individual lead variants may not display evidence of association because of lack of power. Moreover, in the discovery study, the lead variant is either the causal variant or is strongly correlated with it. However, differences in LD mean that the lead variant may not be correlated in another ancestry group and may therefore not display evidence of association. Our assessment of transferability was therefore based on PAT ratios that aggregate information across loci and account for all three factors, sample size, MAF and differences in LD[28].

First, credible sets for each locus were generated. They consisted of lead variant plus all correlated SNPs ($r^2 \geq 0.8$) within a 50 kb window of the lead variant (based on ancestry matched LD reference panels from the 1,000 Genomes data) and with $P < 100 \times P_{lead}$. A signal was defined as being 'transferable' to another ancestry group if at least one variant from the credible set was associated at two-sided $P < 10^{(\log_{10} 0.05) - P_f \times (N-1)}$ with MD and had consistent direction of effect between the discovery and test study. *N* is the number of SNPs in the credible set for each locus, and $P_f$ is a penalization factor we derived from empirical estimations. The effective number of independent SNPs was often higher in other ancestry groups due to differences in LD, leading to higher multiple testing burden and higher likelihood of identifying SNPs with a low *P* value, by chance alone. This inflates the test statistics and was adjusted for by the penalization factor ($P_f$). To derive the $P_f$ for each ancestry group, we used the summary statistics from a previous GWAS on breast cancer[77], in which phenotypes were believed to be uncorrelated with MD. A total of 441 breast cancer significant SNPs were taken from their paper, and linear regressions were conducted for the *P* values of these SNPs in each of our ancestrally diverse summary statistics for MD on the number of SNPs in credible sets. The coefficient estimates (slope from regressions) were treated as $P_f$ for each ancestry. As a result, $P_f$ were 0.008341, 0.007378, 0.006847 and 0.003147 for samples of African, East Asian, South Asian ancestry and for the Hispanic/Latin American group, respectively.

In the next step, the statistical power to detect an association of a given locus was calculated assuming an additive effect at a type I error rate of 0.05, with effect estimates from the discovery study, and allele frequency and sample size from each of the target datasets from diverse ancestry/ethnic groups. The power estimates were summed up across published loci to give an estimate of the total number of loci expected to be significantly associated. This is the expected number if all loci are transferable and accounts for the statistical power for replication. We calculated the PAT ratio by dividing the observed number of loci by the expected number. In addition, loci were defined as 'nontransferable' if they had sufficient power for identifying an association but did not display evidence of association, that is, if they contained at least one variant in the credible set with >80% power, while none of the variants in the credible set had $P < 0.05$ and no variant within 50 kb of locus had $P < 1 \times 10^{-3}$ in the target dataset.

For comparison, we also conducted a transferability assessment for a European ancestry look-up study. The 102 significant loci reported by Howard and colleagues[13] were evaluated for their transferability in the AGDS study using the aforementioned method.

To assess whether low transferability may be due to heterogeneous outcome definitions, we carried out a sensitivity analysis, where we estimated PAT ratios based only on studies fulfilling the clinical MD definition.

### Trans-ancestry genetic correlations
We estimated trans-ancestry genetic correlations using POPCORN version 1.0 (refs. [29],[54],[78]). Pairwise correlations were calculated between each combination of the five ancestry/ethnic groups (that is, African, European, East Asian, South Asian and Hispanic/Latin American) for broad depression and clinical depression separately.

### Gene annotation
The summary statistic from the multi-ancestry meta-analysis was first annotated with FUMA[79]. Both positional mapping and eQTL mapping results were extracted from FUMA. The 1,000 Genomes European samples were employed as the LD reference panel for FUMA gene annotation. Datasets for brain tissue available in FUMA were employed for eQTL gene annotation.

Gene-based association analyses were implemented using Multi-marker Analysis of GenoMic Annotation (MAGMA, version 1.08) (ref. [80]) and Hi-C coupled MAGMA (H-MAGMA)[81]. The aforementioned multiple ancestry LD reference panel from the UKB was used as the LD reference panel. H-MAGMA assigns noncoding SNPs to their cognate genes based on long-range interactions in disease-relevant tissues measured by Hi-C[81]. We used the adult brain Hi-C annotation file.

### Transcriptome-wide association analysis and drug mapping
To perform a TWAS, the FUSION software was used[82]. SNP weights were downloaded from the FUSION website[83] and were derived from multiple external studies, including (1) SNP weights from all available brain tissues, adrenal gland, pituitary gland, thyroid gland and whole blood[33] from GTEx v8 (ref. [84]) (based on significantly heritable genes and 'All Samples' in GTEx v8, which also includes African American and Asian individuals); (2) SNP weights from the CommonMind Consortium, which includes samples from the brain dorsolateral prefrontal cortex; (3) SNP weights from the Young Finns study; and (4) from the Netherlands Twin Register, which provides SNP weights from blood tissues (whole blood and peripheral blood, respectively).

We used the multi-ancestry LD reference panel described above. Variants present in the 1,000 Genomes European population reference panel were retained. A separate TWAS was also performed using a LD reference panel based on the 1,000 Genomes Project's samples of European ancestry, as a sensitivity analysis.

The transcriptome-wide significance threshold for the TWAS associations in this study was $P < 1.37 \times 10^{-6}$. This threshold was previously derived using a permutation-based procedure, which estimates a significance threshold based on the number of features tested[33].

The results were compared with previous TWAS in MD, including the two largest MD TWAS so far[2,15,33,85,86]. These studies generally used smaller sets of SNP weights (except the study by Dall'Aglio and colleagues, which used similar SNP weights as the current study, but with SNP weights derived from the previous GTEx release, v7). The TWAS Z-score plot was generated using a TWAS-plotter function[87].

To assess the relevance of novel genes to drug discovery, genes were searched in three large drug databases: GeneCards[88], DrugBank and ChEMBL[89,90]. In Table 1, a selection of drugs (the ones reported in multiple publications) probably targeting our high-confidence prioritized gene sets are shown for each gene.

### MR
We performed a bi-directional two-sample MR analysis using the TwoSampleMR R package (version 0.5.6)[91,92] to test possible causal effects between MD and six cardiometabolic traits. We followed the STROBE-MR (strengthening the reporting of observational studies in epidemiology using Mendelian randomisation) guidelines (Supplementary Note). For individuals of European ancestry, the UKB was used to select instruments for BMI, fasting glucose, HDL, LDL, SBP and TGs. SBP summary data were obtained from the UKB for individuals of African and South Asian ancestry and Hispanic/Latin American participants. For samples of African, East Asian and South Asian ancestry and the Hispanic/Latin American group, a meta-analysis was performed using METAL[57] with inverse variance weighting using the UKB and the following consortia: GIANT[93] for BMI; MAGIC[94] for fasting glucose; Global Lipids Genetics Consortium[95] for HDL, LDL and TG; and Biobank Japan[95,96] for SBP in samples of East Asian ancestry. The genetic associations with quantitative variables were estimated with respect to the scale, units and models defined in the original studies. Heterogeneity analyses were also performed. To avoid sample overlap, the datasets used to define instrumental variables (IV) for the cardiometabolic traits were excluded from the MD genome-wide association statistics used for the MR analyses conducted with respect to each ancestry group.

Genome-wide significance ($P = 5 \times 10^{-8}$) was used as the threshold to select IVs for the exposures. However, if less than ten variants were available, a suggestive threshold ($P = 5 \times 10^{-6}$) was used to select IVs (Supplementary Table 16). We only included IVs that were present in both datasets (exposure and the outcome). We followed the three main IV assumptions for the analysis: (1) relevance: the IV is associated with the risk factor of interest; (2) independence: the IV is not associated with confounders; and (3) exclusion: the IV is only associated with the outcome through the exposure. We used the following criteria for clumping: $r^2 = 0.001$ and a 10,000 kb window. The following information was used in both the exposure and outcome data: SNP ID, effect size, effect allele, other allele, EAF and P value. We used five different MR methods: IVW, MR–Egger, weighted median, simple mode and weighted mode[92]. The IVW estimates were reported as the main results due to their higher statistical power[97] while the other tests were used to assess the consistency of the estimates across different methods. MR–Egger regression intercept and MR heterogeneity tests were conducted as additional sensitivity analyses. In case of significant heterogeneity, the MR–pleiotropy residual sum and outlier global test was used to remove genetic variants based on their contribution to heterogeneity[98]).

### Reporting summary
Further information on research design is available in the Nature Portfolio Reporting Summary linked to this article.

### Data availability
GWAS summary statistics will be made available via the PGC website (https://www.med.unc.edu/pgc/download-results/) under dataset identifier 'mdd2023diverse'. 23andMe, WHI and JHS do not permit

sharing of genome-wide summary statistics. The full GWAS summary statistics for the 23andMe discovery dataset will be made available through 23andMe to qualified researchers under an agreement with 23andMe that protects the privacy of the 23andMe participants. Please visit https://research.23andme.com/collaborate/#dataset-access/ for more information and to apply to access the data. Investigators can apply for access to WHI and JHS via dbGaP (https://www.ncbi.nlm.nih.gov/gap/). The current study utilized data from dbGaP studies under application #18933.

## Code availability

We used publicly available software for the analyses. The software used is listed in the Methods section. Custom analysis scripts are available at https://doi.org/10.5281/zenodo.8335659.

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

## Acknowledgements

We are grateful to all the participants who took part in the studies and acknowledge the investigators involved in the participating studies. We thank all members of the UCL HumGen laboratory (https://www.uclhumgen.com/), who gave critical support and suggestions. We thank A. Henry, University College London, for discussions and suggestions in key analytic techniques. This work was conducted on the University College London Computer Science cluster; we thank the cluster team for the support provided. China Kadoorie Biobank's most important acknowledgement is to the participants in the study. The investigators also acknowledge the invaluable contributions of the members of the survey teams in each of the ten regional centres, and of the project development and management teams based at Beijing, Oxford and the ten regional centres. China's National Health Insurance provides electronic linkage to all hospital treatments. The Mexican Adolescent Mental Health Survey cohort thanks C. Benjet and E. Méndez for their contributions to the Mexican Adolescent Mental Health Survey cohort's data acquisition and curation, respectively. Intern Health Study thanks the training physicians for taking part in the Intern Health Study. This project was funded by the National Institute of Mental Health (grant no. R01MH101459). MVP thanks the veterans who participate in the MVP. From the Yale Department of Psychiatry, Division of Human Genetics, we thank and acknowledge the efforts of A. M. Lacobelle, C. Robinson and C. Tyrell. Funding: this work was supported by funding from the Veterans Affairs Office of Research and Development MVP grant CX001849- 01 (MVP025) and VA Cooperative Studies Program CSP575B. D.F.L. was supported by an NARSAD Young Investigator Grant from the Brain and Behavior Research Foundation. The BioVU study used for the analyses described was obtained from Vanderbilt University Medical Center's BioVU, which is supported by numerous sources: institutional funding, private agencies and federal grants. These include the National Institutes of Health (NIH)-funded Shared Instrumentation grant S10RR025141; and Clinical and Translational Science Awards grants UL1TR002243, UL1TR000445 and UL1RR024975. Genomic data are also supported by investigator-led projects that include U01HG004798, R01NS032830, RC2GM092618, P50GM115305, U01HG006378, U19HL065962 and R01HD074711; and additional funding sources listed at https://victr.vumc.org/biovu-funding/. CONVERGE authors are part of the CONVERGE consortium (China, Oxford and Virginia Commonwealth University Experimental Research on Genetic Epidemiology) and gratefully acknowledge the support of all partners in hospitals across China. Special thanks to all the CONVERGE collaborators and patients who made this work possible. CONVERGE Consortium: N. Cai, T. B. Bigdeli, W. Kretzschmar, Y. Li, J. Liang, L. Song, J. Hu, Q. Li, W. Jin, Z. Hu, G. Wang, L. Wang, P. Qian, Y. Liu, T. Jiang, Y. Lu, X. Zhang, Y. Yin, Y. Li, X. Xu, J. Gao, M. Reimers, T. Webb, B. Riley, S. Bacanu, R.E. Peterson, Y. Chen, H. Zhong, Z. Liu, G. Wang, J. Sun, H. Sang, G. Jiang, X. Zhou, Y. Li, Y. Li, W. Zhang, X. Wang, X. Fang, R. Pan, G. Miao, Q. Zhang, J. Hu, F. Yu, B. Du, W. Sang, K. Li, G. Chen, M. Cai, L. Yang, D. Yang, B. Ha, X. Hong, H. Deng, G. Li, K. Li, Y. Song, S. Gao, J. Zhang, Z. Gan, H. Meng, J. Pan, C. Gao, K. Zhang, N. Sun, Y. Li, Q. Niu, Y. Zhang, T. Liu, C. Hu, Z. Zhang, L. Lv, J. Dong, X. Wang, M. Tao, X. Wang, J. Xia, H. Rong, Q. He, T. Liu, G. Huang, Q. Mei, Z. Shen, Y. Liu, J. Shen, T. Tian, X. Liu, W. Wu, D. Gu, G. Fu, J. Shi, Y. Chen, X. Gan, L. Liu, L. Wang, F. Yang, E. Cong, J. Marchini, H. Yang, J. Wang, S. Shi, R. Mott, Q. Xu, J. Wang, K. S. Kendler and J. Flint. The AGDS is indebted to all of the participants for giving their time to contribute to this study. We thank all the people who helped in the conception, implementation, media campaign and data cleaning. We thank R. Parker, S. Cross and L. Sullivan for their valuable work coordinating all the administrative and operational aspects of the AGDS project. The 23andMe Study thanks the research participants and employees of 23andMe for making this work possible. Genes and Health thanks Social Action for Health, Centre of the Cell, members of our Community Advisory Group, and staff who have recruited and collected data from volunteers. We thank the NIHR National Biosample Centre (UK Biocentre), the Social Genetic and Developmental Psychiatry Centre (King's College London), Wellcome Sanger Institute and the Broad Institute for sample processing, genotyping, sequencing and variant annotation. We thank Barts Health NHS Trust, NHS Clinical Commissioning Groups (City and Hackney, Waltham Forest, Tower Hamlets, Newham, Redbridge, Havering, Barking and Dagenham), East London NHS Foundation Trust, Bradford Teaching Hospitals NHS Foundation Trust, Public Health England (especially D. Wyllie), Discovery Data Service/Endeavour Health Charitable Trust (especially D. Stables) for General Data Protection Regulation compliant data sharing backed by individual written informed consent. Most of all we thank all of the volunteers participating in Genes and Health. Supplementary Tables 17–21 contain the complete list of members of the PGC-MDD Working Group, the 23andMe Research Team, the Genes and Health Research Team, the China Kadoorie Biobank Collaborative Group and the BioBank Japan Project. This study is part of a project that has received funding from the European Research Council under the European Union's Horizon 2020 research and innovation program (grant agreement no. 948561) and from Wellcome (212360/Z/18/Z). Computing was supported by the Biotechnology and Biological Sciences Research Council (BB/R01356X/1). D.K. is supported by the MSCA Individual Fellowship, European Commission (101028810). G.N. is supported by the Biotechnology and Biological Sciences Research Council, grant number BB/M009513/1. R.P. is supported by grants from the NIH (R33 DA047527; R21 DC018098) and One Mind Rising Star Award. G.A.P. is supported by the Yale Biological Sciences Training Program (T32 MH014276). P.-H.K. is supported by The Ministry of Science and Technology Project (MOST 108-2314-B-002-136-MY3), the National Health Research Institutes Project (NHRI-EX106-10627NI) and the National Taiwan University Career Development Project (109L7860). A.M.M. is supported by grants from the Wellcome Trust (226770/Z/22/Z, 223165/Z/21/Z, 220857/Z/20/Z and 216767/Z/19/Z), UK Research and Innovation (MR/S035818/1 and MR/W014386/1), the United States NIH (R01MH124873) and the European Union Horizon 2020 scheme (grant agreement 847776). M.J.A. is supported by grants from the Wellcome Trust (104036/Z/14/Z and 220857/Z/20/Z). The AGDS was primarily funded by the National Health and Medical Research Council (NHMRC) of Australia grant 1086683. This work was further supported by NHMRC grants 1145645, 1078901, 1113400 and 1087889 and NIMH. The QSkin study was funded by the NHMRC (grant numbers 1073898, 1058522 and 1123248). N.G.M. is supported through NHMRC investigator grant 1172990. W.E. is supported by National Institute of Drug Abuse grant R01DA009897. CONVERGE was funded by the Wellcome Trust (WT090532/Z/09/Z, WT083573/Z/07/Z and WT089269/Z/09/Z) and by NIH grant MH100549. K.S.K. was supported by NIMH R01MH125938 and R21MH126358. R.E.P. was supported by NIMH R01MH125938, R21MH126358 and The Brain & Behavior Research Foundation NARSAD grant 28632P&S Fund. L.K.D. is supported by funding R01 MH118223. R.J.U., R.C.K. and M.B.S. are supported by the US Department of Defense. J.G. and the MVP study were supported by funding from the Veterans Affairs Office of Research and Development MVP grant CX001849-01 (MVP025) and

VA Cooperative Studies Program CSP575B. S.S. is supported by R01MH101459. M.U., D.E.W., and A.E.A. are supported by 2R01MD011728. C.S.C.-F. is supported by Cohen Veteran Bioscience, Mexico's National Institute of Psychiatry, Mexico's National Council of Science and Technology. The Hispanic Community Health Study/ Study of Latinos is a collaborative study supported by contracts from the National Heart, Lung, and Blood Institute (NHLBI) to the University of North Carolina (HHSN268201300001I/N01-HC-65233), University of Miami (HHSN268201300004I/N01-HC-65234), Albert Einstein College of Medicine (HHSN268201300002I/N01-HC-65235), University of Illinois at Chicago (HHSN268201300003I/ N01-HC-65236 Northwestern University) and San Diego State University (HHSN268201300005I/N01-HC-65237). The following institutes/centers/offices have contributed to the HCHS/SOL through a transfer of funds to the NHLBI: National Institute on Minority Health and Health Disparities, National Institute on Deafness and Other Communication Disorders, National Institute of Dental and Craniofacial Research, National Institute of Diabetes and Digestive and Kidney Diseases, National Institute of Neurological Disorders and Stroke, NIH Institution–Office of Dietary Supplements. The Genetic Analysis Center at the University of Washington was supported by NHLBI and NIDCR contracts (HHSN268201300005C AM03 and MOD03). The Hispanic Community Health Study/Study of Latinos also received support from the National Institutes of Health Award (R01MH113930). B.S.M. and W.E. are supported by the NIDA. J.A.R. and G.U. are supported by NIDA, NIA, VA, University of Maryland, Maryland VA Healthcare System, Baltimore Research and Education Foundation. The China Kadoorie Biobank baseline survey and the first re-survey were supported by the Kadoorie Charitable Foundation in Hong Kong. Long-term follow-up was supported by the Wellcome Trust (212946/Z/18/Z, 202922/Z/16/Z, 104085/Z/14/Z and 088158/Z/09/Z), the National Key Research and Development Program of China (2016YFC0900500, 2016YFC0900501, 2016YFC0900504 and 2016YFC1303904) and the National Natural Science Foundation of China (91843302). DNA extraction and genotyping was funded by GlaxoSmithKline, and the UK Medical Research Council (MC-PC-13049 and MC-PC-14135). The China Kadoorie Biobank is supported by core funding from the UK Medical Research Council (MC_UU_00017/1, MC_UU_12026/2 and MC_U137686851), Cancer Research UK (C16077/A29186 and C500/ A16896) and the British Heart Foundation (CH/1996001/9454) to the Clinical Trial Service Unit and Epidemiological Studies Unit and to the MRC Population Health Research Unit at Oxford University. Genes and Health has recently been core-funded by Wellcome (WT102627 and WT210561), the Medical Research Council (UK) (M009017), Higher Education Funding Council for England Catalyst, Barts Charity (845/1796), Health Data Research UK (for London substantive site) and research delivery support from the NHS National Institute for Health Research Clinical Research Network (North Thames). Genes and Health has recently been funded by Alnylam Pharmaceuticals, Genomics PLC; and a Life Sciences Industry Consortium of Bristol– Myers Squibb Company, GlaxoSmithKline Research and Development Limited, Maze Therapeutics Inc, Merck Sharp & Dohme LLC, Novo Nordisk A/S, Pfizer Inc, Takeda Development Centre Americas Inc. C.M.L. and J.R.I.C. are part-funded by the NIHR Maudsley Biomedical Research Centre at South London and Maudsley NHS Foundation Trust and King's College London (NIHR203318). Research reported in this publication was supported by the National Institute of Mental Health of the National Institutes of Health under award number R01MH124873. Statistical analyses were carried out on the Genetic Cluster Computer (http://www.geneticcluster.org) hosted by SURFsara and financially supported by the Netherlands Scientific Organization (NWO 480-05-003 PI: Posthuma) along with a supplement from the Dutch Brain Foundation and the VU University Amsterdam.

## Author contributions

K.K. conceived this project and supervised the work. X.M., G.N. and K.K. wrote and critically revised the manuscript. C.M.L. critically revised the manuscript. X.M. and O.G. carried out the study-level GWAS analysis. Y. Feng carried out meta-analysis for studies of African ancestry. X.M. carried out all meta-analysis, loci and independent SNP identification, fine mapping and annotation including FUMA, MAGMA and Hi-C MAGMA. X.M. and G.N. carried out the transferability analyses. G.N. carried out the TWAS analysis and conducted the drug target look-up. D.K. and R.P. carried out the MR analysis. G.A.P. and R.P. carried out the QTL and colocalization analysis. G.N., X.M. and K.K. wrote and revised the study descriptions (Supplementary Note). A.M. provided support for analyses and result interpretation. M.V. and C.-Y.C. contributed to drafting and editing of the manuscript. For MVP, J.G. and M.B.S. are the principal investigators; J.M.G. and D.F.L. were involved in preparing data. For China Kadoorie Biobank, Z.C. and L.L. are the principal investigators; R.G.W. is the genomics lead; R.G.W. and I.Y.M. were involved in data collection; R.G.W. and K.L. carried out quality control and genome-wide association analysis; K.L. carried out the genotype imputation. For Genes and Health, D.A.v.H. is the principal investigator; B.T., S.F., N.B., H.C.M., V.K.C. and Q.Q.H. were involved in data collection and analysis. For the GREAT study cohort, P.-H.K. is the principal investigator; H.-C.C., S.-J.T. and Y.-L.L. were involved in data collection and analysis. For AGDS, N.G.M. and E.M.B. are the principal investigators; B.L.M. carried out data analysis. For CONVERGE, K.S.K. is the principal investigator; R.E.P. and N.C. were involved in data collection and analysis. For BioVU, L.K.D. is the principal investigator; K.'E.V.A. was involved in data collection and analysis. For Army STARRS, R.J.U. and M.B.S. are the principal investigators, and R.C.K. was involved in data collection and preparation. For Intern Health Study, S. Sen is the principal investigator; Y. Fang, L.J.S. and M.B. were involved in data collection and preparation. For BioMe, R.J.F.L. is the principal investigator and M.H.P. was involved in data collection and analysis. For 23andMe, Y.J. and C.T. were involved in data collection and analysis. For the Drakenstein Child Health Study, D.J.S. and H.J.Z. are the principal investigators; M.L.C. and N.K. were involved in data collection and preparation. For the Detroit Neighborhood Health Study, M.U. is the principal investigator; A.H.W., D.E.W. and A.E.A. were involved in data collection and preparation. For the Mexican Adolescent Mental Health Survey Cohort, C.S.C.-F. is the principal investigator; G.A.M.-L., M.E.R. and A.I.C. were involved in data collection and analysis. For the Hispanic Community Health Study/Study of Latinos, E.C.D. and S.W.-S. are the principal investigators; T.S. was involved in data collection and analysis. For PIRC first generation trial, W.E. is the principal investigator; J.A.R., B.S.M. and G.U. were involved in data collection and analysis. For the BioBank Japan Project, Y.O. is the principal investigator; M.K. and S. Sakaue were involved in data collection and analysis. For PGC-MDD, C.M.L., S.A., J.R.I.C., A.M.M., M.J.A., S.K. and S.R. contributed to the PGC-MDD study of European ancestry. All authors read and critically revised the manuscript and made substantial intellectual contributions to the study. All authors approved the manuscript.

## Competing interests

O.G. is now a full-time employee at Union Chimique Belge. A.I.C. is currently an employee of Regeneron Pharmaceuticals and may own stock or stock options. C.T. reported being an employee of and receiving stock options from 23andMe during the conduct of the study. Y.J. reported being an employee of 23andMe outside the submitted work. All other authors declare no competing interests.

## Additional information

**Extended data** is available for this paper at https://doi.org/10.1038/s41588-023-01596-4.

**Correspondence and requests for materials** should be addressed to Karoline Kuchenbaecker.

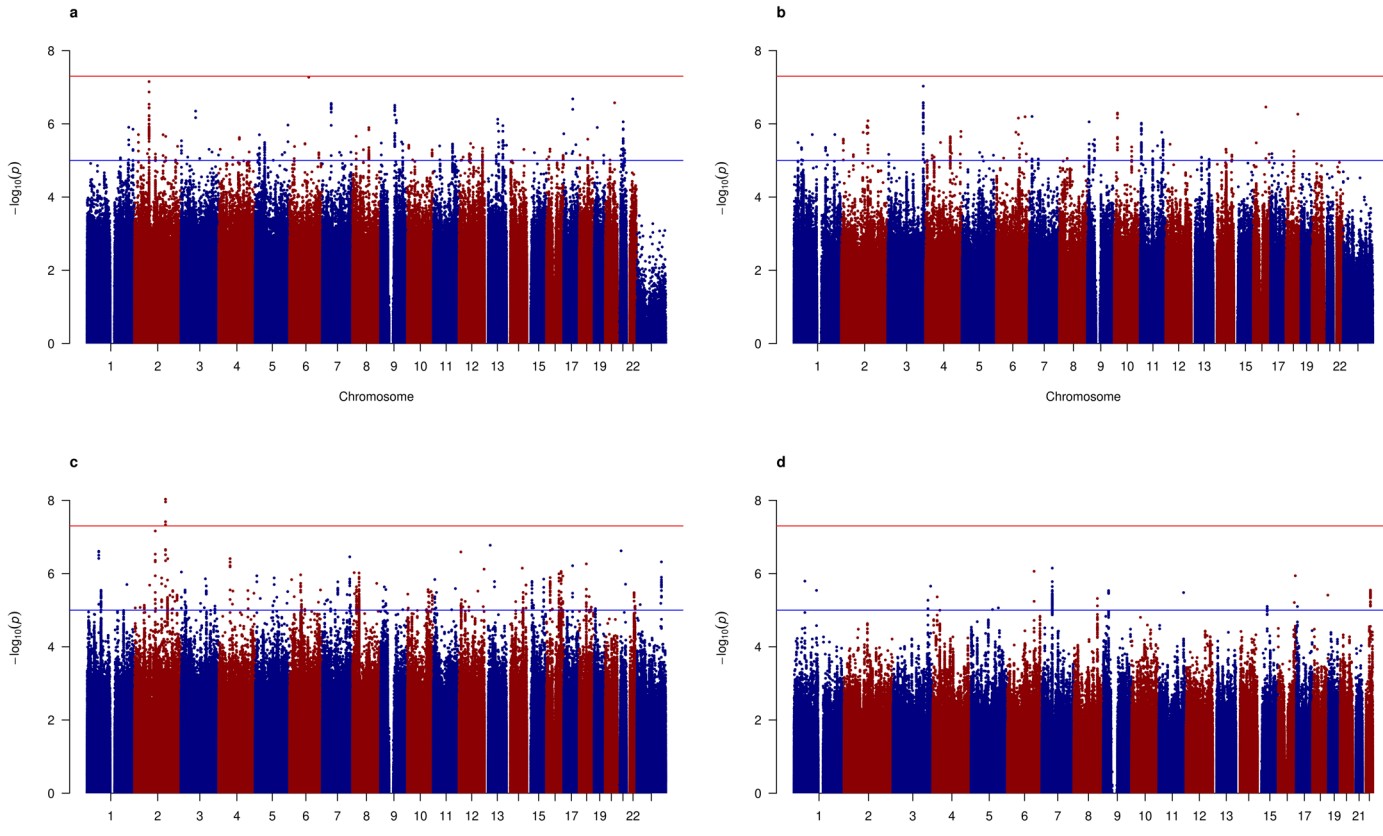

**Extended Data Fig. 1 | Manhattan plots for genetic associations with major depression in non-European ancestries.** The *y*-axes show the −log₁₀*P* values for the associations between each single-nucleotide polymorphism and major depression. The *x*-axes show the chromosomal position (GRCh37). The red line represents the genome-wide significance threshold of $5 \times 10^{-8}$ and the blue line $10^{-5}$. **a**, Manhattan plot for African ancestry. Due to the restriction that SNPs need to be available in at least two studies, only results for 6,051 variants were available on the X chromosome. **b**, Manhattan plot for East Asian ancestry. **c**, Manhattan plot for Latin American ancestry. Association *P* values have been adjusted by the LDSC intercept of 1.0508. **d**, Manhattan plot for South Asian ancestry. Only one cohort provided data for variants on the X chromosome. Those are not included because for the meta-analysis at least two cohorts were required to provide data for each variant.

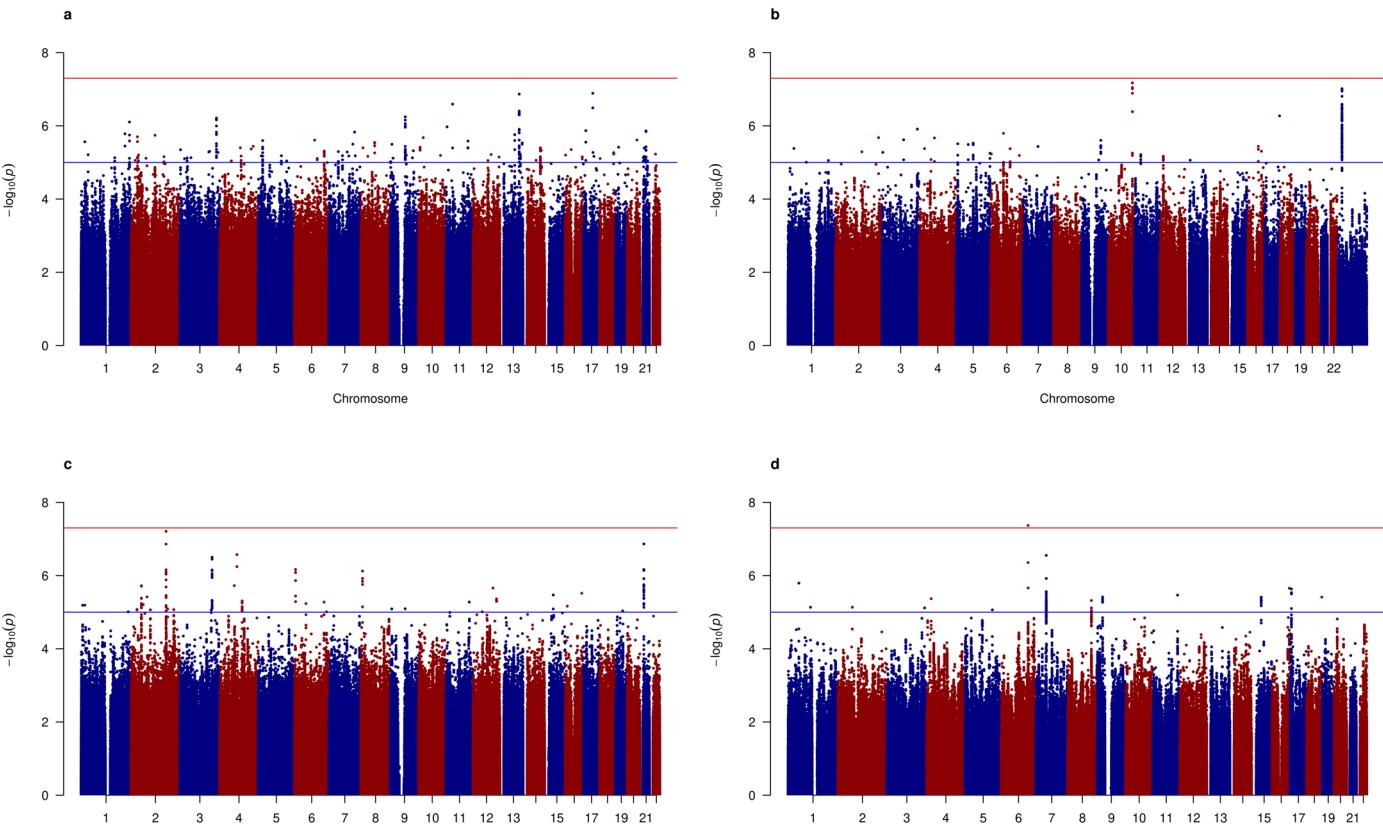

**Extended Data Fig. 2 | Manhattan plots for genetic associations with clinical major depression in individuals of non-European ancestries.** The *y*-axes show the −log$_{10}$*P* values for the associations between each single-nucleotide polymorphism and major depression. The *x*-axes show the chromosomal position (GRCh37). The red line represents the genome-wide significance threshold of $5 \times 10^{-8}$ and the blue line $10^{-5}$. **a**, Manhattan plot for African ancestry. **b**, Manhattan plot for East Asian ancestry. **c**, Manhattan plot for Latin American ancestry. **d**, Manhattan plot for South Asian ancestry.

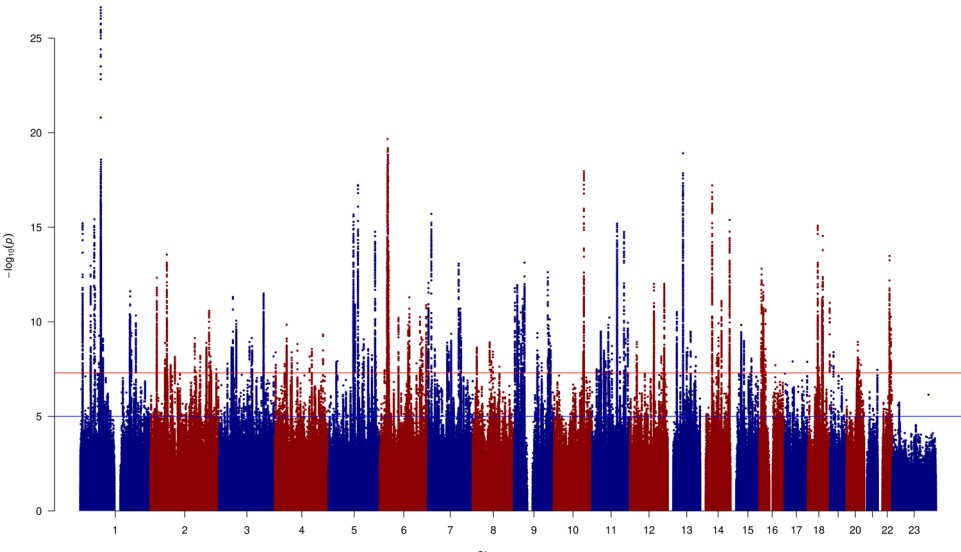

**Extended Data Fig. 3 | Manhattan plot for genetic associations with major depression in the multi-ancestry meta-analysis.** The $y$-axes show the $-\log_{10}P$ values for the associations between each single-nucleotide polymorphism and major depression. The $x$-axes show the chromosomal position (GRCh37). The red line represents the genome-wide significance threshold of $5 \times 10^{-8}$ and the blue line $10^{-5}$. Association $P$ values have been adjusted by the LDSC intercept of 1.0185.

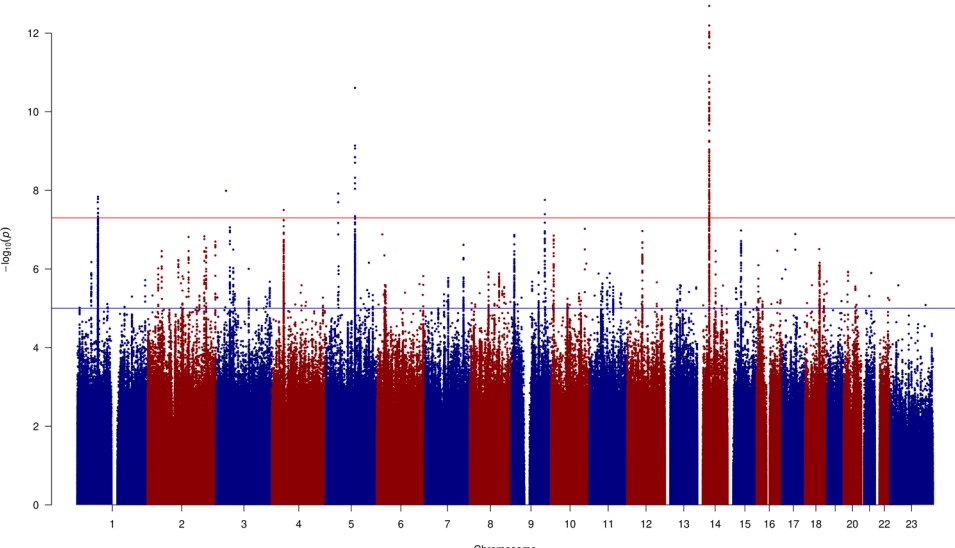

**Extended Data Fig. 4 | Manhattan plot for genetic associations with clinical major depression in the multi-ancestry meta-analysis.** The *y*-axes show the −log₁₀*P* values for the associations between each single-nucleotide polymorphism and major depression. The *x*-axes show the chromosomal position (GRCh37). The red line represents the genome-wide significance threshold of $5 \times 10^{-8}$ and the blue line $10^{-5}$.

# Reporting Summary

## Statistics

For all statistical analyses, confirm that the following items are present in the figure legend, table legend, main text, or Methods section.

| n/a | Confirmed | |
|---|---|---|
| ☐ | ☒ | The exact sample size (*n*) for each experimental group/condition, given as a discrete number and unit of measurement |
| ☐ | ☒ | A statement on whether measurements were taken from distinct samples or whether the same sample was measured repeatedly |
| ☐ | ☒ | The statistical test(s) used AND whether they are one- or two-sided *Only common tests should be described solely by name; describe more complex techniques in the Methods section.* |
| ☐ | ☒ | A description of all covariates tested |
| ☐ | ☒ | A description of any assumptions or corrections, such as tests of normality and adjustment for multiple comparisons |
| ☐ | ☒ | A full description of the statistical parameters including central tendency (e.g. means) or other basic estimates (e.g. regression coefficient) AND variation (e.g. standard deviation) or associated estimates of uncertainty (e.g. confidence intervals) |
| ☐ | ☒ | For null hypothesis testing, the test statistic (e.g. *F*, *t*, *r*) with confidence intervals, effect sizes, degrees of freedom and *P* value noted *Give P values as exact values whenever suitable.* |
| ☒ | ☐ | For Bayesian analysis, information on the choice of priors and Markov chain Monte Carlo settings |
| ☒ | ☐ | For hierarchical and complex designs, identification of the appropriate level for tests and full reporting of outcomes |
| ☐ | ☒ | Estimates of effect sizes (e.g. Cohen's *d*, Pearson's *r*), indicating how they were calculated |

*Our web collection on statistics for biologists contains articles on many of the points above.*

## Software and code

Policy information about availability of computer code

| Data collection | Data collection not part of this study. No software was used for data collection. |
|---|---|
| Data analysis | Mixed-effect models were used in the association analysis for CKB, BioME, Genes & Health with SAIGE (version 0.36.1, version 0.37, or version 0.39). The CONVERGE study initially conducted mixed-effect model GWA tests with FastLMM (version 2.06.20130802), followed by PLINK logistic regressions to retrieve logORs. For the CONVERGE study, the logORs and standard errors from PLINK were used in our meta-analysis. The HCHS/SOL implemented mixed-effect model GWA tests to adjust for population structure and relatedness with depression as binary outcome and was run using GENESIS. We implemented inverse-variance weighted fixed-effect meta-analyses using METAL (version 2011-03-25) and a multi ancestry meta-regression using MR-MEGA (v.0.2). To identify independent association signals, the GCTA (version 1.92.0 beta2) forward selection and backward elimination process (command 'cojo-slt') were applied using the summary statistics from the multi-ancestry meta-analysis, with a multi ancestry LD reference panel. We performed colocalization between genetic associations with MD and gene expression in brain and blood tissues from samples of European and African ancestry and Hispanic/Latinx participants using coloc R package (version). The summary statistic from the multi-ancestry meta-analysis was first annotated with FUMA (v1.3.7). Gene-based association analyses were implemented using Multi-marker Analysis of GenoMic Annotation (MAGMA, v1.08) and Hi-C coupled MAGMA (H-MAGMA). To perform a transcriptome-wide association study (TWAS), the FUSION software was used. We performed a bi-directional two-sample MR analysis using the TwoSampleMR R package (v0.5.6, https://mrcieu.github.io/TwoSampleMR/index.html). We estimated trans-ancestry genetic correlations using POPCORN v1.029,55,64. Pairwise correlations were calculated between each |

combination of the 5 major ancestry/ethnic groups (i.e. African, European, East Asian, Hispanic/Latinx and South Asian) for broad depression and clinical depression separately.

For manuscripts utilizing custom algorithms or software that are central to the research but not yet described in published literature, software must be made available to editors and reviewers. We strongly encourage code deposition in a community repository (e.g. GitHub). See the Nature Portfolio guidelines for submitting code & software for further information.

## Data

Policy information about availability of data

All manuscripts must include a data availability statement. This statement should provide the following information, where applicable:
- Accession codes, unique identifiers, or web links for publicly available datasets
- A description of any restrictions on data availability
- For clinical datasets or third party data, please ensure that the statement adheres to our policy

GWAS summary statistics will be made available via the PGC website https://www.med.unc.edu/pgc/download-results/. Dataset identifier: 'mdd2023diverse'. 23andMe, WHI and JHS do not permit sharing of genome-wide summary statistics. The full GWAS summary statistics for the 23andMe discovery data set will be made available through 23andMe to qualified researchers under an agreement with 23andMe that protects the privacy of the 23andMe participants. Please visit https://research.23andme.com/collaborate/#dataset-access/ for more information and to apply to access the data. Investigators can apply for access to WHI and JHS via dbgap https://www.ncbi.nlm.nih.gov/gap/.

## Human research participants

Policy information about studies involving human research participants and Sex and Gender in Research.

| Reporting on sex and gender | We used biological sex in the study. It was determined based on the participants' genotypes. |
| --- | --- |
| Population characteristics | The population characteristics of participants across multiple studies are as follows:<br><br>CKB Study: Mean age is 52.2 years (SD=10.7), with a 59.5% female cohort.<br>CONVERGE Study: Mean age is 46.1 years, with an entirely female cohort.<br>Taiwan Study: Mean age is 49.2 years (SD=11.3), and 57.5% are female.<br>WHI Study: For different ancestries, the mean ages are as follows: 62.7 years (SD=7.5) for East Asians, 61.5 years (SD=7.1) for Africans, and 60.3 years (SD=6.7) for Hispanic/Latin Americans. The cohort is 100% female.<br>IHS Study: Mean ages by ancestry are 27.4 years (SD=2.4) for East Asians, 27.8 years (SD=2.7) for Africans, and 26.6 years (SD=2.1) for South Asians, with female proportions of 54.8%, 63.3%, and 46.9%, respectively.<br>UKB Study: Mean ages by ancestry are 52.1 years (SD=7.3) for East Asians, 50.7 years (SD=7.4) for Africans, and 53.0 years (SD=8.3) for South Asians. The cohort has 72.1%, 61%, and 43.8% females, respectively.<br>Army-STARRS Study: Mean ages by ancestry are 24.5 years (SD=6.3) for East Asians, 23.5 years (SD=5.7) for Africans, and 22.8 years (SD=5.1) for Hispanic/Latin Americans, with female proportions of 12.5%, 21.4%, and 15.4%, respectively.<br>BioMe Study: Mean age is 58.9 years, with 58.7% females.<br>BBJ Study: Mean age is 63.0 years, with 46.3% females.<br>AGDS Study: Mean age is 44.1 years (SD=15.1), with 75.1% females.<br>IHS Study: Mean age is 55.2 years (SD=12.2), with 63.6% females.<br>DCHS Study: Mean age is 26.4 years (SD=5.6), with an entirely female cohort.<br>HCHS/SOL Study: Mean age is 46 years (SD=14), with 59% females.<br>DNHS Study: Mean age is 53.2 years (SD=16.6), with 58.3% females.<br>PIRC Study: Mean age is 29 years, with 58.5% females.<br>MAMHS Study: Mean age is 14.28 years, with 68.8% females.<br>ProMIS Study: Mean age is 28.2 years (SD=6.3), with an entirely female cohort.<br><br>Further details on population characteristics are provided in Supplementary Table 1, titled "Cohort Summary." |
| Recruitment | We provide detailed descriptions of the 21 cohorts included in this study in the supplementary material. |
| Ethics oversight | Each of the cohorts included was approved by a relevant ethics review board and we have listed the details in the manuscript. |

Note that full information on the approval of the study protocol must also be provided in the manuscript.

# Field-specific reporting

Please select the one below that is the best fit for your research. If you are not sure, read the appropriate sections before making your selection.

☒ Life sciences  ☐ Behavioural & social sciences  ☐ Ecological, evolutionary & environmental sciences

For a reference copy of the document with all sections, see nature.com/documents/nr-reporting-summary-flat.pdf

# Life sciences study design

All studies must disclose on these points even when the disclosure is negative.

| | |
|---|---|
| Sample size | To determine sample size we added up the number of participants of each study that was included in a given analysis. |
| Data exclusions | We restricted the downstream analysis to genetic variants with imputation accuracy info score of 0.7 or higher and effective allele count (2*maf*(1-maf)*N*R2) of 50 or higher. For study of small sample size, we instead required a minor allele frequency of no less than 0.05. The alleles for indels were re-coded as "I" for the longer allele and "D" for the shorter one. Indels of different patterns at the same position were removed. |
| Replication | All available cohorts of major depression cases and controls were included in the primary multi-ancestry meta-analysis and therefore we do not perform replication for significant loci we identified from the multi-ancestry meta-analysis in independent cohorts. We tested replication of previously identified loci linked to depression from European ancestry across non-European ancestry groups and this is described in the manuscript as transferability. Add some more information as requested by editors. |
| Randomization | This was a genetic association study. Allocation by genotype. |
| Blinding | This was a genetic association study, ie observational design. So no blinding was used. |

# Reporting for specific materials, systems and methods

We require information from authors about some types of materials, experimental systems and methods used in many studies. Here, indicate whether each material, system or method listed is relevant to your study. If you are not sure if a list item applies to your research, read the appropriate section before selecting a response.

## Materials & experimental systems

| n/a | Involved in the study |
|---|---|
| ☒ | ☐ Antibodies |
| ☒ | ☐ Eukaryotic cell lines |
| ☒ | ☐ Palaeontology and archaeology |
| ☒ | ☐ Animals and other organisms |
| ☒ | ☐ Clinical data |
| ☒ | ☐ Dual use research of concern |

## Methods

| n/a | Involved in the study |
|---|---|
| ☒ | ☐ ChIP-seq |
| ☒ | ☐ Flow cytometry |
| ☒ | ☐ MRI-based neuroimaging |

