## [Peer Review File · Nature Genetics]

Peer Review Information

Manuscript Title: Multi-ancestry genome-wide association study of major depression aids locus discovery, fine-mapping, gene prioritization and causal inference

Corresponding author name(s): Dr Karoline Kuchenbaecker

Reviewer Comments & Decisions:

Decision Letter, initial version:

27th September 2022

Dear Karoline,

Your Article "Multi-ancestry GWAS of major depression aids locus discovery, fine-mapping, gene prioritisation, and causal inference" has been seen by two referees. You will see from their comments below that, while they find your work of interest, they have raised several relevant points. We are interested in the possibility of publishing your study in Nature Genetics, but we would like to consider your response to these points in the form of a revised manuscript before we make a final decision on publication.

To guide the scope of the revisions, the editors discuss the referee reports in detail within the team, including with the chief editor, with a view to identifying key priorities that should be addressed in revision, and sometimes overruling referee requests that are deemed beyond the scope of the current study. In this case, we ask that you provide further details on how phenotypes were defined within each study population, perform additional analyses to assess the potential impact of study-specific differences in phenotype definition on the results of the analyses, justify the distinct analytical strategy applied to the Hispanic/Latinx subgroup, address all technical queries related to the association analyses and their interpretation, clarify provisions for releasing genome-wide association summary statistics, and revise the presentation for clarity throughout. We hope you will find this prioritized set of referee points to be useful when revising your study. Please do not hesitate to get in touch if you would like to discuss these issues further.

We therefore invite you to revise your manuscript taking into account all reviewer and editor comments. Please highlight all changes in the manuscript text file. At this stage we will need you to upload a copy of the manuscript in MS Word .docx or similar editable format.

We are committed to providing a fair and constructive peer-review process. Do not hesitate to contact

us if there are specific requests from the reviewers that you believe are technically impossible or unlikely to yield a meaningful outcome.

*2) If you have not done so already please begin to revise your manuscript so that it conforms to our Article format instructions, available [here](http://www.nature.com/ng/authors/article_types/index.html). Refer also to any guidelines provided in this letter.

[redacted]

We hope to receive your revised manuscript within 8-12 weeks. If you cannot send it within this time, please let us know.

Sincerely,
Kyle

Kyle Vogan, PhD
Senior Editor
Nature Genetics
<https://orcid.org/0000-0001-9565-9665>

Referee expertise:

Referee #1: Genetics, psychiatric diseases

Referee #2: Genetics, psychiatric diseases

Reviewers' Comments:

Reviewer #1:
Remarks to the Author:

This manuscript is noteworthy in that it reports one of the first efforts in psychiatric genetics to conduct large scale GWAS across multiple diverse ancestral populations, and perhaps the first for depression. Given this context its findings are likely to be widely cited. While there are many features in the work that are truly worthy of such attention – for example the effort to quantify the “transferability” of associations identified in published European studies to the non-European samples that are the main focus of the manuscript – there are some serious shortcomings that must be addressed for the work to be published in a journal with the impact of Nature Genetics.

Specific major issues:

(1) Phenotypes: The lack of detail regarding different definitions of and approaches for assessing “depression” is unacceptable. From the information that is provided (Supplementary material and ST 12) it is, for some studies, impossible to discern how phenotypes were assigned. In other studies, the method indicated describes a screening tool that is not usually considered suitable for the diagnosis of major depression; for example, the CES-D screener asks about depressive symptoms over the past week while the DSM diagnosis of major depressive disorder requires assessment of at least a two-week period. The authors allude to possible advantages and disadvantages of their approach to phenotype inclusion, with citations to papers that have discussed this topic previously (p.16 par 5). I think it is a much more serious point here than in the studies cited, however, for precisely the reason that the authors then note; these differences in phenotype could explain the differences observed across different ethnic populations. There are also potentially major phenotypic differences between samples from different ethnic populations that the authors don’t even mention; for example, one of the Hispanic/Latinx samples consisted of individuals with adolescent depression, which many investigators have considered a phenotype which is distinct from adult depression. This single Discussion paragraph is insufficient as a means to address this topic. The authors must make a

concerted effort to determine the possible effect on their main findings of phenotypic differences between cohorts (and of course must provide more detailed information about the phenotypes themselves).

(2) Populations: I am sympathetic to the authors' statement (Results Par 1) that there is an arbitrary aspect to the divisions of the overall sample into different ancestry groups (and I think because of this point they could have highlighted the fact that, from a "diverse ancestries" standpoint this is a collection of samples of convenience). However, I don't understand why they haven't provided more specific details/results regarding the ancestral composition of the samples, especially given the concerns about phenotypic differences noted above. But given the approach that they have taken with respect to the rest of the sample, I am perplexed as to why they have singled out the Hispanic/Latinx sample set for such a different approach; as they have explicitly not used ancestry (i.e., genetically determined) to delineate this set it is incomprehensible to me that they have not considered (beyond a brief statement in the Discussion) how this decision may have impacted their results. The authors must make an effort to formally investigate this point.

(3) Clarity and interpretation: There are several places where clarification is needed and, overall, I think the authors are overstating their results. Specific comments relate to these points. These comments are in order as they appear in the manuscript, not necessarily in order of importance.

a. Page 10: We tested ancestry specific eQTLs from blood, and observed RBMS1 (H3:PP (Hispanic/Latinx) = 99.12%). Is this even a complete sentence? It's also an example of unexplained jargon – what is H3, what is PP? Supplementary Table 1 also uses this jargon w/o explanation. Perhaps these are known abbreviations/concepts, but they all should be defined them. Note: they define H3 and H4 only in footnote to Supplementary Table 1, we shouldn't have to dig so hard to understand this.

b. In the next sentence, the MS says: For the protein quantitative trait loci (pQTLs) from blood, we either did not find the genes or were not enough to test for colocalisation (Supplementary Table 1). "were not enough" – were not enough genes? What does this mean?

c. The authors highlight two variants, one that is genome-wide significant (GWS) in the Hispanic/Latinx sample, and one that is suggestive in the African ancestry sample. Both of these samples had pretty large sample sizes; why is there such a dearth of GWS results? Might it be a bias with the genotyping arrays, which were mainly developed in European samples? Is it a result of the phenotyping heterogeneity or population definition issues discussed in 1 and 2, above? The authors should make an effort to explore this observation and then devote more of the Discussion to it. Furthermore, with respect to the two putative loci, I am not sure what point the authors are making regarding the observation that in each case there is a non-significant association in European populations in the same general region.

d. Figure 1 presents a Venn Diagram meant to convey how many of the 196 previously identified loci were transferable to each ancestry group. These numbers don't add up to 196, b/c some were not transferable to any – please indicate that number in the caption.

e. Supplementary Table 2 presents transferability info for each of the 196 previously identified loci, by ancestry group. What does NA mean in this context? For each of these, can the authors indicate the number of variants in the credible set around each locus? They say that they expected 65% of loci to

be transferable to the African populations, but observe a much lower percent. Their expectation is based on a power calculation. Can they present the expectation, for each locus, for each ancestry group, or at least the maximum power for each locus? (as there is, I believe, an estimate for each variant in the credible set for each locus). A locus is considered transferable if at least one variant in the credible set had a p-value < their specified threshold: A signal was defined as being 'transferable' to another ancestry group if at least one variant from the credible set was associated at two-sided $P < 10^{-(\log(0.05, \text{base}=10) - P_f \times (N-1))}$ with MD and consistent direction of effect between the discovery and test study. A locus is considered non-transferable if the power was adequate, but it did not meet the p-value threshold indicated in the methods: In addition, loci were defined as 'non-transferable' if they had sufficient power for identifying an association but did not display evidence of association, i.e., if they contained at least one variant in the credible set with > 80% power, while none of the variants in the credible set had $P < 0.05$ and no variant within 50kb of locus had $P < 1 \times 10^{-3}$ in the target data set. For the loci that WERE transferable, did all of them have power >80%? How many loci met the p-value threshold for transferability but had an inconsistent direction of effect?

f. In their multi-ancestry meta-analysis, they discover 190 signals, represented by 169 loci, including 53 SNVs that are novel. Previously, in European GWAS, 196 loci had been discovered. What loci were "lost" – that were previously GWS and now are not? What would be the reason for their becoming non-significant, would the authors consider the previous association to be a false positive?

g. On page 12, the authors describe their TWAS analysis, and say that they use expression in tissues relevant to MD. When I look at Supplementary Table 5, I see results from a very large number of tissues (bone, kidney, etc) – how are they determining what tissue relevant to MD? Is a p-value threshold of 1.37×10^{-6} appropriate for looking at so many tissues?

h. Page 13 "...existing drugs, such as Simvastatin" – I'm not sure what existing means in this context. Please clarify.

i. Supplementary Table 6 – the results of the current study have the header "Meng_Navoly" – but give just Z-scores – can they also provide p-values, and emphasize that this is the result from the current work? The footnote to this table presents a bunch of asterisks indicating important info, but the asterisks are nowhere else in the table.

j. Page 14: The discussion speaks of 2 novel loci: By aggregating data in ancestry-specific meta-analyses, we identified two novel loci. I'm confused here – they go on to talk about the suggestive locus for the African sample, and I assume the 2nd novel locus was the result from the analysis of the Hispanic/Latinx sample. But neither of these are from the "ancestry-specific meta-analysis", are they? At least they are not listed in Supplementary Table 3, which is titled "multi-ancestry meta-analysis". I had thought those 2 loci were from GWAS of these ancestry groups alone. Please clarify. And, if a locus is suggestive, but not significant, please don't highlight it as a novel finding. It's not yet a "finding".

k. Page 14: The authors suggest that the higher PAT for the Hispanic/Latinx sample is because many of these individuals have European ancestry. As noted above (#2), the authors really need to provide us this information so we can judge for ourselves.

l. Page 16, the authors speak of a notable proportion of loci specific to certain ancestries: In conclusion, in this first large-scale multi-ancestry GWAS of MD, we found a notable proportion of loci

that are specific to certain ancestry groups. Where are these results? Other than the one significant locus reported in the Hispanic/Latinx set (which I'm not sure qualifies as an "ancestry group") and the one suggestive locus in the African ancestry group I am not sure which loci the authors are referring to.

Reviewer #2:

Remarks to the Author:

1. Summary of the key results

: The authors gathered MDD patient GWAS data from diverse ancestry and conducted meta-analyses and followed post-GWAS analyses including fine-mapping, prioritisation of target genes. They assessed transferability between EUR and non-EUR MD GWAS. In addition, they carried out MR analysis to verify the causal relationship between MD and cardiometabolic traits.

2. Originality and significance: if not novel, please include reference

: This study is the first large-scale GWAS of MD in non-EUR ancestries. One strength of this study is that the study included a large proportion of African and East Asian samples. However, there are some points that need to be improved.

3. Data & methodology: validity of approach, quality of data, quality of presentation

1) Method - Participating studies, Suppl Text: More detailed information about QC for MVP cohort is needed in Suppl Text.

2) Method - Participating studies: In the case of Hispanic/Latinx, more explanation and references are needed as to the reason why it is not assigned to the ancestry group using PCA like other ancestry groups. Please provide the excluded number of individuals with admixture between the pre-defined ancestry reference groups.

3) Method - Data availability statement: Data availability for all other cohorts and all relevant summary statistics of this study should be mentioned. Unless the researcher does not have permission to disclose, such as 23andMe data, transparently disclosing summary statistics for all relevant cohorts will greatly help further research.

4) Results - Figure 1B: The expected number of transferable loci might differ across the ancestries. Therefore, please do not present the ratio only, provide the exact number of observed/expected transferable loci.

5) (minor point) The author should provide an exact number of Supple figures or text. Please do not just note them as "Supplementary material", specify them.

4. Appropriate use of statistics and treatment of uncertainties

1) Method - Study level genetic association analyses: If the 'case-control ratio' was extreme, have they considered appropriate statistical methods, (including the firth test), to reduce the statistical bias?

2) Results - Mendelian Randomisation (and relevant text): When reporting MR results, especially when they are the main finding, it is recommended to provide the STROBE-MR checklist. This checklist will be helpful to check the appropriateness of the analysis.

5. Conclusions: robustness, validity, reliability

1) Discussion - Mendelian Randomization: I think there is a critical point. MD is a binary trait, thus the authors should be cautious to interpret MR results as a causal relationship when MD was used as an outcome trait.

6. Suggested improvements: experiments, data for possible revision

1) Results - Transferability of MD association across ancestry groups: I think the cross-ancestry genetic correlation (such as 'popcorn') and the cross-ancestry PRS analysis (such as PRS-CSx) can provide additional information on transferability. The cross-ancestry genetic correlation can be analyzed not only between non-EUR and EUR but also between non-EUR cohorts.

2) Results - Multi-ancestry meta-analysis: Did the authors consider using Multi-Ancestry Meta-Analysis (Turley et al.) along with METAL?

7. References: appropriate credit to previous work?

N/A

8. Clarity and context: lucidity of abstract/summary, appropriateness of abstract, introduction and conclusions

: They summarized previous MD GWAS studies from diverse ancestry well. They also appropriately explained the need for GWAS research on MD in diverse ancestry, not only the need to find novel signals, but also the social and medical unmet needs for this study.

Author Rebuttal to Initial comments
--

Response to the Reviewers' Comments

Reviewer #1:

Remarks to the Author:

This manuscript is noteworthy in that it reports one of the first efforts in psychiatric genetics to conduct large scale GWAS across multiple diverse ancestral populations, and perhaps the first for depression. Given this context its findings are likely to be widely cited. While there are many features in the work that are truly worthy of such attention – for example the effort to quantify the “transferability” of associations identified in published European studies to the non-European samples that are the main focus of the manuscript – there are some serious shortcomings that must be addressed for the work to be published in a journal with the impact of Nature Genetics.

We thank the reviewer for this and for the helpful suggestions and comments. We have conducted additional analyses to address the concerns.

Specific major issues:

1) Phenotypes: The lack of detail regarding different definitions of and approaches for assessing “depression” is unacceptable. From the information that is provided (Supplementary material and ST 12) it is, for some studies, impossible to discern how phenotypes were assigned. In other studies, the method indicated describes a screening tool that is not usually considered suitable for the diagnosis of major depression; for example, the CES-D screener asks about depressive symptoms over the past week while the DSM diagnosis of major depressive disorder requires assessment of at least a two-week period. The authors allude to possible advantages and disadvantages of their approach to phenotype inclusion, with citations to papers that have discussed this topic previously (p.16 par 5). I think it is a much more serious point here than in the studies cited, however, for precisely the reason that the authors then note; these differences in phenotype could explain the differences observed across different ethnic populations. There are also potentially major phenotypic differences between samples from different ethnic populations that the authors don’t even mention; for example, one of the Hispanic/Latinx samples consisted of individuals with adolescent depression, which many investigators have considered a phenotype which is distinct from adult depression. This single Discussion paragraph is insufficient as a means to address this topic. The authors must make a concerted effort to determine the possible effect on their main findings of phenotypic differences between cohorts (and of course must provide more detailed information about the phenotypes themselves).

We agree with the reviewer about the importance of considering the case definition of each study. We have added more details about the outcome assessment of each study to **Supplementary Table 1** as well the **Supplementary Note** where each cohort is described.

We have now also evaluated which studies used definitions that are likely to meet the clinical definition of major depressive disorder (MDD) in line with DSM-IV/5 or ICD10 criteria. This included studies which ascertained major depression by structured clinical interviews based on DSM/ICD criteria for MDD and

those using ICD10-based medical healthcare records (p 20). The codes used to define cases were developed with a clinical psychiatrist (co-author Dr Nick Bass). We then conducted sensitivity analyses to test whether our main findings are impacted by the outcome definition: we implemented new ancestry-specific analyses and multi-ancestry meta-analysis restricted to these clinically ascertained studies (100,089 cases and 215,415 controls in the multi ancestry meta-analysis) (p 22). We have also removed the Hispanic study of adolescents (MAMHS) from the meta-analysis for clinical depression due to the reviewer’s concern regarding the phenotype heterogeneity.

The effect estimates of the genome-wide significant SNVs from the original meta-analysis were highly consistent in the subset of studies for clinical depression, with a correlation coefficient of $r = 0.96$ ($P = 5.8 \times 10^{-105}$, **Supplementary Figure 22**) (p 12f).

Using the clinical definition, we also re-evaluated the transferability of previously identified depression loci from European ancestry discovery GWAS (see updated **Supplementary Table 4**). PAT estimates for clinical depression were close to those for broad depression, with overlapping confidence intervals in each case (updated **Figure 1**, also shown below, results described on p 12). Therefore, we conclude that the transferability results are unlikely to be solely driven by the broad outcome definition in our multi-ancestry GWAS. We were unable to carry out this sensitivity analysis for the Hispanic/Latinx group because there were not enough samples ascertained in line with the clinical definition. However, it is worth noting that the PAT ratio in this group based on the broad definition was significantly higher than the PAT ratios for the other groups.

Figure 1 (updated). Transferability of previously reported loci from European ancestry discovery GWAS of major depression to other ancestry groups. Panel (A) shows the numbers of previously identified loci from European ancestry studies with evidence of transferability to the other ancestry/ethnic groups: African, Hispanic/Latinx, South Asian and East Asian (in black) and their intersections (in cyan). Only the 138 loci with evidence of transferability to at least one ancestry group are shown in Figure 1. Panel (B) shows Power Adjusted Transferability ratios, calculated from the observed number of transferable loci divided by the expected number of transferable loci, taking effect estimates from previous European ancestry studies, and allele frequency and sample size information from our African, Hispanic/Latinx, South Asian and East Asian ancestry cohorts. Ratios were calculated and displayed for broadly defined major depression and for clinically ascertained major depression separately.

2) Populations: I am sympathetic to the authors' statement (Results Par 1) that there is an arbitrary aspect to the divisions of the overall sample into different ancestry groups (and I think because of this point they could have highlighted the fact that, from a "diverse ancestries" standpoint this is a collection of samples of convenience). However, I don't understand why they haven't provided more specific details/results regarding the ancestral composition of the samples, especially given the concerns about phenotypic differences noted above. But given the approach that they have taken with respect to the rest of the sample, I am perplexed as to why they have singled out the Hispanic/Latinx sample set for such a different approach; as they have explicitly not used ancestry (i.e., genetically determined) to delineate this set it is incomprehensible to me that they have not considered (beyond a brief statement in the Discussion) how this decision may have impacted their results. The authors must make an effort to formally investigate this point.

We have now added the PCA plots describing the ancestry composition of individual studies to **Supplementary Figures 1-9**. Studies of the same ancestry clustered well, suggesting that our allocation of major ancestry groups is robust. For each of the 21 studies with data from individuals of diverse non-European ancestry in our multi ancestry meta-analysis we now provided PCA plots (**Supplementary Figure 10**) or references to the paper describing the ancestry assignment of the study.

It has been criticised that genetic studies do not clearly describe how samples are assigned to groups and we tried to address this in our work so we were very clear that the Hispanic/Latinx group was based on self-declared ethnicity or country of recruitment while the other groups were defined based on genetic ancestry relative to a reference group. We do not argue that this is consistent. It is reflecting the policy and practice of categorisation in the country with the largest data contribution, the United States of America. This is reflected in many genetic studies that assign individuals to these same groups and recent papers following this approach¹⁻⁵. The fact that these categorisations are arbitrary and reflect local customs (of the USA) may be uncomfortable, but it is appropriate in that all ancestry groupings are

arbitrary and socially informed. We already had a discussion point about this and have now modified it to better reflect this argument.

“In this study, we assigned individuals into ancestry and ethnic groups. Whilst this enabled important insights, e.g. about transferability of MD loci, such categorical assignments are imprecise and some participants with admixed ancestry may still get excluded. In future research, we aim to implement different analytic strategies that are fully inclusive.” (p 18)

Practically, the alternative would be to re-assign the samples from the individuals in the Hispanic/Latinx group based on their genetic ancestry. However, this group is characterised by recent admixture of Indigenous American, African and European ancestry and therefore many individuals do not fit neatly into one of the genetically defined groups. Moreover, some of the studies, such as 23andMe, followed their own ancestry assignment pipelines that we cannot influence.

3) Clarity and interpretation: There are several places where clarification is needed and, overall, I think the authors are overstating their results. Specific comments relate to these points. These comments are in order as they appear in the manuscript, not necessarily in order of importance.

We have addressed these and worked through the manuscript to improve clarity throughout.

a. Page 10: We tested ancestry specific eQTLs from blood, and observed RBMS1 (H3:PP (Hispanic/Latinx) = 99.12%). Is this even a complete sentence? It’s also an example of unexplained jargon – what is H3, what is PP? Supplementary Table 1 also uses this jargon w/o explanation. Perhaps these are known abbreviations/concepts, but they all should be defined them. Note: they define H3 and H4 only in footnote to Supplementary Table 1, we shouldn’t have to dig so hard to understand this.

We thank the reviewer for this comment. We originally had these listed in the methods section; we have now revised the text to introduce the abbreviations, and their interpretation where they are first described (p 11)

b. In the next sentence, the MS says: For the protein quantitative trait loci (pQTLs) from blood, we either did not find the genes or were not enough to test for colocalisation (Supplementary Table 1). “were not enough” – were not enough genes? What does this mean?

The text reflects that either genes were not present in the QTL-dataset, or the number of SNPs within the gene were too low to explain colocalizing effects, i.e. <20 SNPs. We have revised the text to explain this analysis (p 11).

c. The authors highlight two variants, one that is genome-wide significant (GWS) in the Hispanic/Latinx sample, and one that is suggestive in the African ancestry sample. Both of these samples had pretty large sample sizes; why is there such a dearth of GWS results? Might it be a bias with the genotyping arrays, which were mainly developed in European samples? Is it a result of the phenotyping heterogeneity or population definition issues discussed in 1 and 2, above? The authors should make an effort to explore this observation and then devote more of the Discussion to it.

The reviewer raises an interesting point. Despite large sample sizes, the ancestry-specific GWAS we carried out discovered only a small number of genome-wide significant associations. A similar pattern occurred in previous studies based on data from samples of European ancestry. The subsequent research and discussions⁶ demonstrated that major depression has a highly polygenic genetic architecture and therefore requires larger sample sizes than other diseases for successful discovery⁷. The reviewer is also right in that there may be additional factors impacting on discovery power for ancestrally diverse groups, including coverage of population specific genetic variation. We have added these to the limitations in the discussion section (p 18).

We have investigated the potential impact of heterogeneity due to phenotype definition (please see our response to comment 1).

Furthermore, with respect to the two putative loci, I am not sure what point the authors are making regarding the observation that in each case there is a non-significant association in European populations in the same general region.

In the manuscript we described that the lead variants at the newly identified associations from the GWAS in participants of African ancestry and the Hispanic/Latinx group did not display evidence of association in samples of European ancestry. However, there were variants in the vicinity of the lead SNV that displayed suggestive evidence of association in the European ancestry data. So although there is evidence against a shared causal variant, we cannot rule out that there is an association of the same locus but possibly with a different causal variant in European ancestry participants. We have now clarified this (p 11).

d. Figure 1 presents a Venn Diagram meant to convey how many of the 196 previously identified loci were transferable to each ancestry group. These numbers don't add up to 196, b/c some were not transferable to any – please indicate that number in the caption.

We have addressed this and clarified that the diagram only shows loci with evidence of transferability in at least one of the groups in the figure legend.

e. Supplementary Table 2 presents transferability info for each of the 196 previously identified loci, by ancestry group. What does NA mean in this context? For each of these, can the authors indicate the

number of variants in the credible set around each locus? They say that they expected 65% of loci to be transferable to the African populations, but observe a much lower percent. Their expectation is based on a power calculation. Can they present the expectation, for each locus, for each ancestry group, or at least the maximum power for each locus? (as there is, I believe, an estimate for each variant in the credible set for each locus). A locus is considered transferable if at least one variant in the credible set had a p-value < their specified threshold: A signal was defined as being ‘transferable’ to another ancestry group if at least one variant from the credible set was associated at two-sided $P < 10^{-(\log(0.05, \text{base}=10) - P_f \times (N-1))}$ with MD and consistent direction of effect between the discovery and test study. A locus is considered non-transferable if the power was adequate, but it did not meet the p-value threshold indicated in the methods: In addition, loci were defined as ‘non-transferable’ if they had sufficient power for identifying an association but did not display evidence of association, i.e., if they contained at least one variant in the credible set with > 80% power, while none of the variants in the credible set had $P < 0.05$ and no variant within 50kb of locus had $P < 1 \times 10^{-3}$ in the target data set. For the loci that WERE transferable, did all of them have power >80%? How many loci met the p-value threshold for transferability but had an inconsistent direction of effect?

We thank the reviewer for this comment. We have addressed this and added more information about the transferability results to **Supplementary Table 4**, including the number of variants in the credible sets for each locus and the maximum power for each locus by ancestry. The “NA” indicated that a reliable assessment of transferability is not possible: Whilst there was no evidence for transferability, the power to find an association was low, meaning that we cannot tell with any confidence whether the association is transferable or not. We have replaced all ‘NA’ with ‘underpowered’ and explained in table legend as well.

It was not the case that all transferable loci had power > 0.8. Amongst the 34 loci that were transferable in the African ancestry meta-analysis, 11 contained variants with power greater than 0.8. There were 9/71, 0/8 and 3/25 such loci in Hispanic/Latinx group, South Asian group and East Asian group, respectively.

There were 4 loci in the South Asian group which met the P-value threshold for transferability but had an inconsistent direction of effect. There were 8 such loci in the African group, 1 such locus in the Hispanic/Latinx group and 0 such locus in the East Asian group.

f. In their multi-ancestry meta-analysis, they discover 190 signals, represented by 169 loci, including 53 SNVs that are novel. Previously, in European GWAS, 196 loci had been discovered. What loci were “lost” – that were previously GWS and now are not? What would be the reason for their becoming non-significant, would the authors consider the previous association to be a false positive?

We have added a new **Supplementary Table 3**, showing the results for each of the 196 previously reported variants in our multi ancestry meta-analysis for both MD definitions. Amongst the 196 associations, 121

were genome-wide significant in our multi ancestry meta-analysis for broad depression and 180 SNPs reached the 5×10^{-6} level of suggestive significance. 16 variants had p-values larger than this.

We do not consider this evidence that those associations are false positives. There are several factors that may lead to larger p-values, including winner's curse. Furthermore, if the lead variant is not the causal variant, differences in LD structure across ancestry groups may result in larger p-values for this variant. Finally, our overall findings suggest that the causal genetic architecture of major depression is not entirely shared across ancestry groups. Therefore, it is also possible that some of the loci with weaker evidence of association in the multi-ancestry GWAS are specific to individuals of European ancestry.

g. On page 12, the authors describe their TWAS analysis, and say that they use expression in tissues relevant to MD. When I look at Supplementary Table 5, I see results from a very large number of tissues (bone, kidney, etc) – how are they determining what tissue relevant to MD? Is a p-value threshold of 1.37×10^{-6} appropriate for looking at so many tissues?

For the TWAS analysis we used tissues that have previously been associated with MD⁸, including the following tissues: multiple brain tissues, adrenal gland, pituitary gland, thyroid gland and whole blood and peripheral blood. Additionally, we would like to note that the TWAS results can be found in Supplementary Table 9. The original Supplementary Table 5 that Reviewer #1 referred to showed the results of the HIC-Magma analysis, which is based on a larger selection of tissues (bone, kidney etc).

To determine the p-value threshold to use in our TWAS analysis we have leveraged a permutation procedure-based significance threshold, adopted from a previous study⁸, which performed TWAS using a very similar set of tissues. We are confident that the permutation-based threshold of 1.37×10^{-6} is appropriate for the current study, as the permutation-based significance threshold is adjusted for the number of features (genes) tested while accounting for correlation and also co-expression between genes, within and across SNP weights.

h. Page 13 "...existing drugs, such as Simvastatin" – I'm not sure what existing means in this context. Please clarify.

Some of the genes we discovered encode proteins that are targeted by established drugs, often approved for conditions other than MD. This is relevant because it may provide opportunities for repurposing drugs. We have now clarified this in the manuscript (p 14).

i. Supplementary Table 6 – the results of the current study have the header "Meng_Navoly" – but give just Z-scores – can they also provide p-values, and emphasize that this is the result from the current work?

The footnote to this table presents a bunch of asterisks indicating important info, but the asterisks are nowhere else in the table.

We thank the reviewer for this helpful comment. We have now added p-values to the results of the current study (Meng_Navoly, in green). We have also made changes to the appearance of the table: we have highlighted the results of the current study in green, and marked the previous TWASs of MD in grey. We have also improved and clarified the headers and labels in the table and addressed the issues with the asterisks. Please note that the TWAS results can now be found at **Supplementary Table 9**.

j. Page 14: The discussion speaks of 2 novel loci: By aggregating data in ancestry-specific meta-analyses, we identified two novel loci. I'm confused here – they go on to talk about the suggestive locus for the African sample, and I assume the 2nd novel locus was the result from the analysis of the Hispanic/Latinx sample. But neither of these are from the “ancestry-specific meta-analysis”, are they? At least they are not listed in Supplementary Table 3, which is titled “multi-ancestry meta-analysis”. I had thought those 2 loci were from GWAS of these ancestry groups alone. Please clarify. And, if a locus is suggestive, but not significant, please don't highlight it as a novel finding. It's not yet a “finding”.

We first combined GWAS data by ancestry group and we labelled those “ancestry-specific meta-analyses”. We subsequently combined the results from each of these ancestry-specific GWAS and called that “multi-ancestry meta-analysis”.

The reviewer is right that suggestive results should not be highlighted as “novel findings”. Therefore, we have removed that section.

k. Page 14: The authors suggest that the higher PAT for the Hispanic/Latinx sample is because many of these individuals have European ancestry. As noted above (#2), the authors really need to provide us this information so we can judge for ourselves.

We have added **Supplementary Figure 1-9**, which showed the PCAs for all participants from included studies. In addition, we provide a PCA plot showing ancestry composition across studies computed using MR-MEGA (**Supplementary Figure 10**) which also shows that Hispanic/Latinx studies ancestrally are located between European, African and East Asian ancestry groups.

l. Page 16, the authors speak of a notable proportion of loci specific to certain ancestries: In conclusion, in this first large-scale multi-ancestry GWAS of MD, we found a notable proportion of loci that are specific to certain ancestry groups. Where are these results? Other than the one significant locus reported in the Hispanic/Latinx set (which I'm not sure qualifies as an “ancestry group”) and the one suggestive locus in the African ancestry group I am not sure which loci the authors are referring to.

We apologise that this sentence was unclear. We agree with the reviewer that the number of loci associated at genome-wide significance in each of the diverse ancestry groups is small, with the exception of the European ancestry group which discovered about 200 loci in previous studies. Therefore, conclusions can be drawn based on the transferability analysis where PAT ratios ranged from ~0.3-0.6. There were multiple loci that were well powered to display an association in the diverse ancestral groups but had p-values larger than 5% (e.g., 35 well-powered loci not associated in the African ancestry GWAS). We agree that the way it was phrased was unclear and have now clarified what evidence this conclusion is based on. "In conclusion, in this first large-scale multi-ancestry GWAS of MD, we demonstrated through transferability analyses that a notable proportion of loci is specific to certain ancestry groups." (p18)

Reviewer #2:

Remarks to the Author:

1. Summary of the key results

: The authors gathered MDD patient GWAS data from diverse ancestry and conducted meta-analyses and followed post-GWAS analyses including fine-mapping, prioritisation of target genes. They assessed transferability between EUR and non-EUR MD GWAS. In addition, they carried out MR analysis to verify the causal relationship between MD and cardiometabolic traits.

2. Originality and significance: if not novel, please include reference

: This study is the first large-scale GWAS of MD in non-EUR ancestries. One strength of this study is that the study included a large proportion of African and East Asian samples. However, there are some points that need to be improved.

3. Data & methodology: validity of approach, quality of data, quality of presentation

1) Method - Participating studies, Suppl Text: More detailed information about QC for MVP cohort is needed in Suppl Text.

We thank the reviewer for this comment. We have added more details about the QC of the MVP data to the **Supplementary Note (p 5f)**.

2) Method - Participating studies: In the case of Hispanic/Latinx, more explanation and references are needed as to the reason why it is not assigned to the ancestry group using PCA like other ancestry groups. Please provide the excluded number of individuals with admixture between the pre-defined ancestry reference groups.

Please see our response to Reviewer 1, comment 2 for a discussion about the categorisation of samples. In the UK Biobank study, 6,686 participants were excluded due to admixture. In the GERA study, 1,200 participants were excluded due to admixture. These numbers of excluded individuals have also been added to their relevant section in the **Supplementary Note**.

3) Method - Data availability statement: Data availability for all other cohorts and all relevant summary statistics of this study should be mentioned. Unless the researcher does not have permission to disclose, such as 23andMe data, transparently disclosing summary statistics for all relevant cohorts will greatly help further research.

We have added a data availability statement on **p 28**. GWAS summary statistics will be made available via the PGC website <https://www.med.unc.edu/pgc/download-results/>. 23andMe, WHI and JHS do not permit sharing of genome-wide summary statistics. The full GWAS summary statistics for the 23andMe discovery data set will be made available through 23andMe to qualified researchers under an agreement with 23andMe that protects the privacy of the 23andMe participants. Datasets will be made available at no cost for academic use. Please visit <https://research.23andme.com/collaborate/#dataset-access/> for more information and to apply to access the data. Investigators can apply for access to WHI and JHS via dbgap <https://www.ncbi.nlm.nih.gov/gap/>.

4) Results - Figure 1B: The expected number of transferable loci might differ across the ancestries. Therefore, please do not present the ratio only, provide the exact number of observed/expected transferable loci.

We agree that this is important and have added this information to **Figure 1B** as well as confidence intervals to the PAT ratios.

5) (minor point) The author should provide an exact number of Supple figures or text. Please do not just note them as "Supplementary material", specify them.

We now refer to the Supplementary Note which provides detailed cohort descriptions and otherwise refer to each Supplementary Table or Figure by number, following the journal's formatting requirements.

4. Appropriate use of statistics and treatment of uncertainties

1) Method - Study level genetic association analyses: If the 'case-control ratio' was extreme, have they considered appropriate statistical methods, (including the firth test), to reduce the statistical bias?

We thank the reviewer for this comment. There were indeed some cohorts with extreme case-control ratios. Whilst some adjusted analytically, others did not. We have added a sensitivity analysis to test whether this may have impacted on the results. We implemented ancestry-specific fixed effect meta-analysis after excluding those studies with extreme case-control ratios (case/control < 0.25) that did not use analytical methods (e.g., SAIGE) to address this. In total, 18 studies were excluded, leaving 304,288 cases (88.1% of total cases) and 862,757 controls (58.7% of total controls) in the sensitivity meta-analysis. We computed Pearson's correlation coefficient for the effect estimates between the primary meta-analysis and this sensitivity analysis, which yielded high correlations ($R = 1.0$, $P = 2.0 \times 10^{-231}$). We describe the methods and results in the manuscript on **p 23** and **13**, respectively, as well as **Supplementary Figure 23**.

2) Results - Mendelian Randomisation (and relevant text): When reporting MR results, especially when they are the main finding, it is recommended to provide the STROBE-MR checklist. This checklist will be helpful to check the appropriateness of the analysis.

We have updated the manuscript with additional information required by the STROBE-MR guidelines and added the STROBE-MR checklist to the **Supplementary Note**.

5. Conclusions: robustness, validity, reliability

1) Discussion - Mendelian Randomization: I think there is a critical point. MD is a binary trait, thus the authors should be cautious to interpret MR results as a causal relationship when MD was used as an outcome trait.

We thank the reviewer for this comment. MR was developed to test the causal effect of a continuous exposure on a binary outcome⁹. Hence, a two-sample MR analysis testing continuous cardiometabolic traits as exposure and MD as outcome is in line with the classic MR design. Instead, as highlighted by Reviewer #2, testing a binary exposure with respect to a continuous trait requires a more cautious interpretation. Specifically, our MR analysis is testing the effect of MD genetic liability (not MD itself) on cardiometabolic traits. Following Reviewer #2's recommendation, we revised the text through the manuscript to clarify this point (**p 27**, **p 15**) and added this information also in the limitation section: "Additionally, our bidirectional MR analysis tested the relationships between MD and cardiometabolic traits. When testing MD as the exposure, the results should be interpreted as the effect of MD genetic liability and not as the effect of MD itself." (**p 18**).

6. Suggested improvements: experiments, data for possible revision

1) Results - Transferability of MD association across ancestry groups: I think the cross-ancestry genetic correlation (such as 'popcorn') and the cross-ancestry PRS analysis (such as PRS-CSx) can provide additional information on transferability. The cross-ancestry genetic correlation can be analyzed not only between non-EUR and EUR but also between non-EUR cohorts.

We thank the reviewer for this helpful suggestion. Since we do not have access to individual level data for some studies, we focussed on the trans-ancestry genetic correlations which can be computed using summary statistics. We computed this for all combinations of ancestry/ethnic groups using POPCORN. Methods are described on p 26 and results are presented on p 12 and in the new **Figure 2**, also shown below. The sample size for the South Asian ancestry group was too small for this analysis. We excluded estimates where the standard error was larger than 0.3 as the estimates are not reliable. The genetic correlations for MD between the European and the Hispanic/Latinx, African and East Asian ancestry groups were ≥ 0.75 . The lowest estimate was observed between East Asian ancestry and the Hispanic/Latinx group ($r_g=0.52$).

Figure 2. Trans-ancestry genetic correlations for depression. Plot showing the genome-wide trans-ancestry genetic correlations between the European, East Asian and Hispanic/Latinx groups. The intensity of the colouring reflects the strength of the correlation. The estimated coefficients and standard errors are also shown in each cell. We only present estimates where the standard error was smaller than 0.3, otherwise the field is coloured in grey.

2) Results - Multi-ancestry meta-analysis: Did the authors consider using Multi-Ancestry Meta-Analysis (Turley et al.) along with METAL?

We appreciate this helpful suggestion and have now implemented a dedicated multi-ancestry meta-analysis method alongside our fixed effects meta-analysis. However, multi-ancestry meta-analysis (MAMA) requires genome-wide LD estimates¹⁰. We do not have access to raw data for some studies and therefore would have to use LD estimates from reference populations for these studies. LD from reference

is unlikely to be a reliable match, in particular for the Hispanic/Latinx studies, given their complex ancestry composition. Moreover, MAMA is a new method that has not yet been extensively evaluated by the research community. Therefore, we applied another multi-ancestry meta-analysis method, MR-MEGA¹¹ which has recently been used in another large GWAS published in *Nature Genetics*².

A total of 106 independent regions were associated with MD at genome-wide significance in the MR-MEGA analysis, 44 of which remained significant after GC correction. We compared the findings between the two methods. The figures below show the P value comparison between METAL and MR-MEGA. Log-transformed P values from METAL are shown on the X axis, and those from MR-MEGA on the Y axis. The figure on the left shows results for all variants, colored by existence of ancestral heterogeneity as calculated by the MR-MEGA (using a threshold of P for the heterogeneity of < 0.1). The figure on the right shows results for the 190 genome-wide significant SNVs we reported in the manuscript, colored by whether there was between-ancestry heterogeneity and shaped by their significance in MR-MEGA. MR-MEGA yielded similar or smaller P values for variants with significant ancestry heterogeneity. However, for variants without significant heterogeneity the P values from MR-MEGA were larger, i.e. less significant, effectively resulting in a penalisation of those variants. Consequently, some of the 190 SNPs were not genome-wide significant in MR-MEGA. Like other large multi-ancestry GWAS (e.g.,¹), we therefore continue to present the fixed effects meta-analysis as the main results. However, we have added the results from MR-MEGA to the supplement (Supplementary Table 6). In addition, we have added PCA plots from MR-MEGA as Supplementary Figure 10 to showcase the ancestral composition of the studies.

Figure: $-\log_{10}(P)$ for the associations of all variants A) and significant SNVs from the multi-ancestry discovery analysis for major depression B) based on fixed effects meta-analysis (x-axis) and MR-MEGA (y-axis).

References

1. Yengo, L. *et al.* A saturated map of common genetic variants associated with human height. *Nature* **610**, 704–712 (2022).
2. Mahajan, A. *et al.* Multi-ancestry genetic study of type 2 diabetes highlights the power of diverse populations for discovery and translation. *Nat. Genet.* **54**, 560–572 (2022).
3. McDonald, M.-L. N. *et al.* Novel genetic loci associated with osteoarthritis in multi-ancestry analyses in the Million Veteran Program and UK Biobank. *Nat. Genet.* **54**, 1816–1826 (2022).
4. Sun, D. *et al.* Multi-Ancestry Genome-wide Association Study Accounting for Gene-Psychosocial Factor Interactions Identifies Novel Loci for Blood Pressure Traits. *HGG Adv* **2**, (2021).
5. Kim, J. J. *et al.* Multi-ancestry genome-wide meta-analysis in Parkinson’s disease. *bioRxiv* (2022) doi:10.1101/2022.08.04.22278432.
6. Flint, J. & Kendler, K. S. The genetics of major depression. *Neuron* **81**, 484–503 (2014).
7. Zhang, Y., Qi, G., Park, J.-H. & Chatterjee, N. Estimation of complex effect-size distributions using summary-level statistics from genome-wide association studies across 32 complex traits. *Nat. Genet.* **50**, 1318–1326 (2018).
8. Dall’Aglio, L., Lewis, C. M. & Pain, O. Delineating the Genetic Component of Gene Expression in Major Depression. *Biol. Psychiatry* **89**, 627–636 (2021).

9. Sanderson, E. *et al.* Mendelian randomization. *Nat Rev Methods Primers* **2**, (2022).
10. Turley, P. *et al.* Multi-Ancestry Meta-Analysis yields novel genetic discoveries and ancestry-specific associations. (2021) doi:10.1101/2021.04.23.441003.
11. Mägi, R. *et al.* Trans-ethnic meta-regression of genome-wide association studies accounting for ancestry increases power for discovery and improves fine-mapping resolution. *Hum. Mol. Genet.* **26**, 3639–3650 (2017).

Decision Letter, first revision:

31st January 2023

Dear Karoline,

Your revised Article "Multi-ancestry GWAS of major depression aids locus discovery, fine-mapping, gene prioritisation, and causal inference" has been seen by the original referees. You will see from their comments below that, while Reviewer #2 is satisfied with the revision, Reviewer #1 has substantial ongoing concerns. We remain interested in the possibility of publishing your study in Nature Genetics, but we would like to consider your response to these ongoing concerns in the form of a further revision before we make a final decision on publication.

As before, to guide the scope of the revisions, the editors discuss the referee reports in detail within the team, including with the chief editor, with a view to identifying key priorities that should be addressed in revision, and sometimes overruling referee requests that are deemed beyond the scope of the current study. In this case, we agree with Reviewer #1 that heterogeneity in the fraction of clinically vs. broadly defined cases within each ancestry/ethnicity group complicates assessments of transferability and the interpretation of the association results in ways that should be further highlighted and discussed. We also ask that you again carefully revise the text and display items for accuracy and clarity throughout. We hope you will find this prioritized set of referee points to be useful when revising your study. Please do not hesitate to get in touch if you would like to discuss these issues further.

We therefore invite you to revise your manuscript again taking into account all reviewer and editor comments. Please highlight all changes in the manuscript text file. At this stage, we will need you to upload a copy of the manuscript in MS Word .docx or similar editable format.

We are committed to providing a fair and constructive peer-review process. Do not hesitate to contact

us if there are specific requests from the reviewer that you believe are technically impossible or unlikely to yield a meaningful outcome.

*2) If you have not done so already please begin to revise your manuscript so that it conforms to our Article format instructions, available [here](http://www.nature.com/ng/authors/article_types/index.html). Refer also to any guidelines provided in this letter.

[redacted]

We again hope to receive your revised manuscript within 8 weeks. If you cannot send it within this time, please let us know.

Sincerely,
Kyle

Kyle Vogan, PhD
Senior Editor
Nature Genetics
<https://orcid.org/0000-0001-9565-9665>

Referee expertise:

Referee #1: Genetics, psychiatric diseases

Referee #2: Genetics, psychiatric diseases

Reviewers' Comments:

Reviewer #1:
Remarks to the Author:

The authors have responded to some but not all of the critiques in the previous review (see below for details); in doing so they have clarified some questions and raised others. Overall, it remains difficult to follow all of the steps/analyses that they have undertaken, and I expect many readers will be confused by the fact that a great many results are reported, without there being much guidance regarding which ones the authors consider most noteworthy. For example, the first section of Results (on the GWAS in the different samples) focuses on the identification of three specific loci from two different types of analyses, with presentation of results of multiple follow-up analyses (both to better characterize biological meaning and to better localize the signal); yet as far as I could tell, none of these results is even mentioned in the Discussion. Similarly confusing, several genes are highlighted in other parts of the Results but not in the Discussion, and vice versa (and I find it particularly difficult to follow the story when results are given in the Discussion that are not evident in the Results). Thus, overall, I cannot say that the MS is substantially improved compared to the previous version. I continue to have major and minor comments.

Major critiques

1. Phenotype definition for depression: One of my major critiques of the previous version was the lack of distinction made between major depression (MD), i.e. a formally assessed condition defined by operational criteria (what they term "clinical depression"), and depression broadly defined, without specific criteria. I appreciate the effort that the authors have now made to assign one of these two phenotypic categories to all cases across all of the cohorts included in this study. The outcome of this effort is striking as indicated in the table below; overall, the number of cases is very substantially reduced, but what is most notable in the table is that it accentuates the extreme heterogeneity between samples of different ancestries/ethnic identifications in what was actually being assessed. So for example, in the South Asian group almost all cases have clinical depression, while in the Hispanic

group almost none do.

Ancestry	N cases total (broad)	N cases clinical
African	35,818	29,389
East Asian	21,980	7,886
South Asian	4,505	4,252
Hispanic	25,013	848

These differences substantially complicate interpretation of results in ways that the authors do not mention. For example, the only GW significant association in the stratified meta-analyses is the one at 2q24.2 in the Hispanic group. It matters that this result derives from a sample in which only 3% of the cases have clinical MD, as indicated by Cai et al. (cited as Ref#10).

The impact of this heterogeneity in phenotype definition is also a major issue in other analyses, in ways that the authors do not emphasize. For example, in the Transferability of MD Associations analyses, one of their points is that "PAT estimates for clinical MD were close to those for broad MD with overlapping CIs in each case". They do not mention here that they could not perform this comparison in the Hispanic sample given the lack of clinically defined cases, while I would say that for the South Asian sample the estimates are not all that close considering that the clinically defined samples represent 94% of the total. Then in the multi-ancestry meta-analyses (in which ~75% of the cases were European-ancestry samples from previously published studies), it appears from visual comparison of the Manhattan plots in Supp. Figs 20 and 21 that most of the GW significant findings are in the broadly defined sample. In Supplementary Table 3, it is evident that of the 196 associations identified previously, 121 are associated at GWS in the broadly defined sample but only two are GW significant in the clinical sample. They note in the text the 53 novel associations seen in the broadly defined sample, but I could not find information to indicate the number of novel findings in the clinically defined sample (is it two?). The phenotypic heterogeneity between the different population samples could matter here also: most of the signal in these analyses derives from the much larger, previously published European-ancestry sample data, and presumably the Hispanic case samples (which are of predominantly European ancestry and very few of which have clinically defined MD) add to the signal of the analyses shown in Supp Fig. 20 but not those shown in Supp. Fig. 21.

2. Designation of "population": The authors' main response to my previous critique was to include PCA plots of the individual studies, which I appreciated. But I do not think this response addressed my main critique, which was about the difference between the ethnic samples defined by ancestry and the Hispanic sample, defined by self-identification only. They justify their approach primarily by amplifying the point that their category conforms to "custom" in the USA, which I do not find a scientifically compelling argument; as noted in my comments in Point 1 above, my concern has a clear implication for the interpretation of the paper, given the extreme phenotypic heterogeneity between this sample and all of the others. They refer to a plan to implement a better approach in future studies (p18), which I will look forward to seeing; but I don't think that their future plan is relevant to considering the results reported now.

Additional Comments

1. Ancestry-specific meta-analysis: As noted above, these analyses are mostly uninformative; they only find one significant association here in their main analysis, and given that it is in the Hispanic

sample it cannot be considered a locus associated with MD as clinically defined. The colocalization work for the 2q24.2 locus looks at a number of genes that aren't in the Locus Zoom plot – why are they included? It is unclear why they do extensive analyses of the 6q locus (e.g. colocalization) when the locus wasn't significantly associated to begin with.

In the text, the authors refer to (e.g.) H3:PP. They have defined H3, but have not yet defined PP, I assume it's posterior probability but please define this, I asked for it in the first review. Supplementary Table 2 does not define H0, H1, or H2.

2. Gene prioritization: On page 13 the authors state: "To better understand the biological mechanisms of our GWAS findings, we performed several in silico analyses to functionally annotate and prioritise the most likely causal genes. We carried out a transcriptome-wide association study (TWAS) for expression in tissues relevant to MD33. We combined the TWAS results with Functional Mapping and Annotation (FUMA), conventional and HiC-MAGMA, to prioritise target genes. We also carried out ancestry-specific eQTL and pQTL colocalisation analyses (Supplementary Table 8)." As this section comes after the multi-ancestry meta-analysis results, and that was the analysis to produce multiple genome-wide significant findings, I assume that this section will pertain to results from the multi-ancestry meta-analysis. Yet the first table cited is Supplementary Table 8, which refers, I think, to the transferability analysis of the 196 previously found loci. At least the title of the table says: Supplementary Table 8. QTL mapping for the transferability SNPs for all tissues – there are 102 SNPs here – are these SNPs that transferred AND are associated in the multi-ancestry meta-analysis? I believe Supplementary Table 8 in the current draft was Supplementary Table 5 in the previous draft. The rebuttal letter states: The original Supplementary Table 5 that Reviewer #1 referred to showed the results of the HiC-Magma analysis, which is based on a larger selection of tissues (bone, kidney, etc.). If Supplementary Table 8 is displaying HiC-MAGMA results, why isn't it cited after reference to HiC-MAGMA in the text? If it is HiC-MAGMA results, why does the table title refer to QTL? This is quite confusing.

3. Mendelian Randomization: The bidirectional results are difficult to explain, biologically, especially when results differ in direction for the same phenotypes in different ancestries, and the p-values in many cases are pretty weak. It's hard to take this as good evidence for causality or reverse causality.

4. Conclusion: The final paragraph of the discussion includes a sentence I commented on in the first review: "In conclusion, in this first large-scale multi-ancestry GWAS of MD, we demonstrated through transferability analyses that a notable proportion of loci is specific to certain ancestry groups." In the rebuttal letter the authors say: Therefore, conclusions can be drawn based on the transferability analysis where PAT ratios ranged from ~0.3-0.6. There were multiple loci that were well powered to display an association in the diverse ancestral groups but had p-values larger than 5% (e.g., 35 well-powered loci not associated in the African ancestry GWAS). I would interpret this to mean that "a notable proportion of loci is specific to European populations" based on the evidence you've shown us, as there are loci associated in Europeans, that were powered to be detected in diverse ancestry groups, that were not associated.

Other more minor comments (not in order of importance)

1. Supplementary Table 1 is a nice addition to better understand the composition of the studies. The table is rather sloppy. Under "depression measure" what is the difference between "Medically

diagnosed", "Medically" and "Medical diagnosis" – are these the same? What is the difference between "Lifetime depression", "MDD" and "Lifetime diagnosis"? Is "Depression diagnosis" equivalent to "self-report"? For definitions that were symptom-based, the authors did not always give the threshold for the CES-D and PHQ-9 to declare a participant to be a case – can they please update this? What does "computer assisted" mean with respect to "depression measure"? Could the authors add to this table columns indicating whether the study provided raw genotype data or summary statistics? The text says that there are 21 studies, but in counting the number of studies in this table it appears to be more (first column), can they clarify?

2. The summary tables in general have a lot of typos and often do not define the column headers. Frequently, for association results, a column is titled "beta" and I think this is the log-odds ratio – can the authors please be more precise regarding what "beta" means (this is true in the text as well, please specify logOR rather than beta).

3. Supplementary Table 3 lists results for 196 loci but in several places in the text (e.g. page 11) they say 206 loci – which is correct?

4. Supplementary Table 4 has been expanded quite a bit with more information, but it is a little confusing. For example, take rs2089358, the p-value for association in the African sample is 0.47, yet it is indicated that this locus transferred. It must be that some other variant among the 23 in the credible set had a p-value < transfer threshold (?). It would be helpful to include the result for the SNV that did transfer (or at least the most significant SNV in the credible set), as opposed to the results for what I guess are the lead SNPs.

5. Supplementary Tables 12 and 13 are missing a column for gene symbol, it's hard to interpret findings without this information.

6. The supplementary Manhattan plots don't always have the y-axis extended to the point where the red line for genome-wide significance is displayed – can the authors correct this so that all plots have the y-axis extend to at least this level?

7. The text states that 43 genes had consistent results across prioritization methods (page 14) but Supplementary Table 5 says there are 44 – which is correct?

8. On page 16, the authors state: "Gene NDUFAF3 encodes mitochondrial complex I assembly protein which is the main target of the drug metformin⁴³, the first-line drug for treatment of type 2 diabetes. Research in model organisms has provided a tentative link to a reduction in depression and anxiety⁴⁴." In that last sentence, I think the authors mean that use of metformin is linked to a reduction in depression and anxiety.

Reviewer #2:
Remarks to the Author:

The authors have sufficiently answered my questions.

Author Rebuttal, first revision:

Response to the reviewers' comments

We thank the reviewers for their work. We provide a point-by-point response to the comments of reviewer 1 below.

Reviewer #1:

The authors have responded to some but not all of the critiques in the previous review (see below for details); in doing so they have clarified some questions and raised others. Overall, it remains difficult to follow all of the steps/analyses that they have undertaken, and I expect many readers will be confused by the fact that a great many results are reported, without there being much guidance regarding which ones the authors consider most noteworthy. For example, the first section of Results (on the GWAS in the different samples) focuses on the identification of three specific loci from two different types of analyses, with presentation of results of multiple follow-up analyses (both to better characterize biological meaning and to better localize the signal); yet as far as I could tell, none of these results is even mentioned in the Discussion. Similarly confusing, several genes are highlighted in other parts of the Results but not in the Discussion, and vice versa (and I find it particularly difficult to follow the story when results are given in the Discussion that are not evident in the Results). Thus, overall, I cannot say that the MS is substantially improved compared to the previous version. I continue to have major and minor comments.

We appreciate that this is a complex study with several data sets and different analyses. We have now developed a schematic that gives an overview of the different analytical steps to provide better guidance to the reader (new Figure 1, also shown below).

We hope that both the abstract and the conclusions clearly highlight which ones we consider to be the most important results:

“In conclusion, in this first large-scale multi-ancestry GWAS of MD, we demonstrated through transferability analyses that a notable proportion of loci is specific to samples of European ancestry. We identified novel, biologically plausible associations that were missed in European ancestry analyses and demonstrated that large diverse samples can be important for the identification of target genes and putative mechanisms. These findings suggest that for MD, a heterogeneous condition with highly complex aetiology, increasing ancestral as well as global diversity in genetic studies may be particularly important to ensure discovery of core genes and to inform about transferability of findings across ancestry groups.”
(p 19)

Figure 1. We included data from 21 cohorts with diverse ancestry. We first assigned individuals from each study into ancestry/ethnic groups and carried out association analyses with major depression (MD) for each one. Subsequently, we meta-analysed the results by ancestry/ethnic group. These meta-analysis results were the basis to test whether previously reported MD loci from European ancestry studies are transferable to other groups. We also used those results for discovery of novel depression associations, and Mendelian Randomisation to assess causal effects of cardiometabolic traits by ancestry. We subsequently merged all ancestry/ethnicity-specific results in a multi-ancestry meta-analysis that also included samples with European ancestry. The multi-ancestry meta-analysis results formed the basis for locus discovery, fine-mapping to identify causal variants and several gene prioritisation approaches to identify genes linked to MD risk. ST.(n) refers to the corresponding Supplementary Table. ST.2* (in green) - refers to Supplementary Table 2, showing genomic inflation estimates of multiple analyses.

We have carefully checked consistency between the results and discussion section. The discussion includes sections on ancestry-specific hits, transferability, fine-mapping, TWAS/gene prioritisation and Mendelian Randomisation. This is consistent with the sections in the results. After the last revision the discussion section on ancestry-specific hits had been shortened to one sentence. We have now expanded this to be more in line with the results section.

We also checked the genes that are mentioned. There is a sentence in the discussion where we mention additional cadherin genes. We had only mentioned one of the related genes in the results (*PCDH8P*) in the results section.. To ensure consistency we have now mentioned the other cadherin genes in the results as well (p 14).

Major critiques

1. Phenotype definition for depression: One of my major critiques of the previous version was the lack of distinction made between major depression (MD), i.e. a formally assessed condition defined by operational criteria (what they term “clinical depression”), and depression broadly defined, without specific criteria. I appreciate the effort that the authors have now made to assign one of these two phenotypic categories to all cases across all of the cohorts included in this study. The outcome of this effort is striking as indicated in the table below; overall, the number of cases is very substantially reduced, but what is most notable in the table is that it accentuates the extreme heterogeneity between samples of different ancestries/ethnic identifications in what was actually being assessed. So for example, in the South Asian group almost all cases have clinical depression, while in the Hispanic group almost none do.

Ancestry N cases total (broad) N cases clinical

African 35,818 29,389

East Asian 21,980 7,886

South Asian 4,505 4,252

Hispanic 25,013 848

These differences substantially complicate interpretation of results in ways that the authors do not mention. For example, the only GW significant association in the stratified meta-analyses is the one at 2q24.2 in the Hispanic group. It matters that this result derives from a sample in which only 3% of the cases have clinical MD, as indicated by Cai et al. (cited as Ref#10).

The impact of this heterogeneity in phenotype definition is also a major issue in other analyses, in ways that the authors do not emphasize. For example, in the Transferability of MD Associations analyses, one of their points is that “PAT estimates for clinical MD were close to those for broad MD with overlapping CIs in each case”. They do not mention here that they could not perform this comparison in the Hispanic sample given the lack of clinically defined cases, while I would say that for the South Asian sample the estimates are not all that close considering that the clinically defined samples represent 94% of the total.

Then in the multi-ancestry meta-analyses (in which ~75% of the cases were European-ancestry samples from previously published studies), it appears from visual comparison of the Manhattan plots in Supp. Figs 20 and 21 that most of the GW significant findings are in the broadly defined sample. In Supplementary Table 3, it is evident that of the 196 associations identified previously, 121 are associated at GWS in the broadly defined sample but only two are GW significant in the clinical sample. They note in the text the 53 novel associations seen in the broadly defined sample, but I could not find information to indicate the number of novel findings in the clinically defined sample (is it two?). The phenotypic heterogeneity between the different population samples could matter here also: most of the signal in these analyses derives from the much larger, previously published European-ancestry sample data, and presumably the Hispanic case samples (which are of predominantly European ancestry and very few of which have clinically defined MD) add to the signal of the analyses shown in Supp Fig. 20 but not those shown in Supp. Fig. 21.

Recent analyses done by Naomi Wray’s group as part of the Psychiatric Genomics Consortium provide a comprehensive comparison between clinical MD, electronic health record, questionnaire and self-report of diagnosis. The genetic correlations between all outcome definitions were high (0.78-0.88), in line with previous research (Wray et al, 2018, NG). Genomic structural equation modelling demonstrated that a common factor model with the loading on the clinical MD fixed to 1 was consistent with the data and all outcome definitions had high positive loadings on the common factor (≥ 0.85). There was no evidence of

heterogeneity at depression loci except for one. This strongly supports the use of multiple outcome definitions in genetic association studies of MD.

Whilst our work includes substantial numbers of samples with clinical definition for all groups except the Latinx/Hispanic group, statistical power for discovery of novel associations is immensely improved by including studies that used other outcome definitions, as is evident when comparing the Manhattan plots for broadly defined vs clinical MD. Here, it is important to note that the restriction to clinical MD also affects the sample size of the European ancestry sample. Broad: 258,364 cases and 571,252 controls of European ancestry vs clinical: 57,714 cases and 110,358 controls. This leads to a total sample size for clinical MD of 100,089 cases and 214,415 controls in comparison to 345,389 cases and 1,469,702 controls for broad MD. The analysis of clinical MD identified seven genome-wide significant loci, two of which were novel (rs2085224 at 3p22.3, rs78676209 at 5p12) (Supplementary Table 8). We now report this on page 13.

The vast majority of the >200 genetic loci that have been linked to depression in previous studies are based on mixed outcome definitions (Wray et al, 2018, *Nature Genetics*; Howard et al, 2019, *Nature Neuroscience*; Mitchell et al, 2021, *Biological Psychiatry*; Levey et al, 2021, *Nature Neuroscience*). Work restricting to clinically ascertained samples confirmed through structured interviews is equally important but has only identified a handful of loci so far (Cai et al, 2015, *Nature*).

We agree with the reviewer that implications of the outcome definitions need to be discussed. It is possible that some of the loci are linked to susceptibility to mental illness more widely instead of being MD specific (Cai et al, 2020, *Nature Genetics*). However, given the overlap across most psychiatric disorders (Cross-disorder group et al, 2019, *Cell*) and the lack of heterogeneity of the association of previously identified depression loci with a clinical MD factor, such findings may nevertheless be of great value for our understanding of the biology and the development of new treatments for depression. We now also mention in the abstract that different outcome definitions were included and have expanded the relevant section in the discussion to cover this point.

“By aggregating data in ancestry-specific meta-analyses, we identified two novel loci. In the Hispanic/Latinx group, variants at 2q24.2 were associated with MD. Most of the cases in this group were defined using symptoms questionnaires. Future studies will be required to assess whether the association of this loci with clinical MD is consistent with our estimate.” (p 15)

“This study utilised data from several existing cohorts and bioresources to achieve large sample sizes. This necessitated using different outcome definitions, covering self-administered symptoms questionnaires,

electronic healthcare records, as well as structured clinical interviews. The potential advantages and disadvantages of these approaches have been extensively discussed in previous studies^{6,10}. As a consequence, it is possible that some of the 190 genome-wide significant loci we identified are linked to a more general susceptibility to mental illness instead of being specific to MD. However, given the overlap between different psychiatric disorders⁵⁴, such findings are nevertheless of value for our understanding of the biology and the development of new treatments for MD.” (p 18)

The reviewer is concerned that heterogeneity due to ancestry and heterogeneity due to depression definition could confound each other. Here, it is important to distinguish between the discovery analyses and the transferability analyses. For the discovery analyses, we demonstrated during the last revision that the vast majority of genome-wide significant loci show highly consistent results in the analysis of clinical depression. We believe that providing full results for both analyses will be of great value to the research community. Depending on the specific follow-up research question, investigators can select the loci from our manuscript that are most suitable to their interest.

With respect to transferability, the additional analyses we carried out for the last revision addressed potential concerns about confounding by outcome definition. There, we re-estimated the transferability for the clinical MD definition. The results were highly consistent with the estimates for broad MD. The reviewer points out that “for the South Asian sample the estimates are not all that close considering that the clinically defined samples represent 94% of the total”. For samples of South Asian ancestry, 8 vs 6 out of 27 loci are transferable using broad vs clinical MD. The confidence intervals of the respective PAT ratios are widely overlapping (0.12-0.46 vs 0.07-0.40). So these differences are in line with chance. Most importantly, the conclusion holds true: Broad or clinical, our analyses show that there is transferability but it is substantially and statistically significantly less than 1 (PAT ratio broad vs clinical MD for samples of African ancestry: 0.27 vs 0.26, South Asian: 0.29 vs 0.22, East Asian: 0.29 vs 0.45), demonstrating that many of the previously reported loci in European ancestry are not associated in African, East and South Asian ancestry groups.

The reviewer is correct in pointing out that for the Latinx/Hispanic group, there were not enough studies using a clinical definition to carry out the transferability analyses for clinical MD. This is a limitation of our work. Only additional data collection can address this. We had in fact included this in the discussion “We were unable to carry out this sensitivity analysis for the Hispanic/Latinx group because there were not enough samples ascertained in line with the clinical definition. However, it is worth noting that the PAT ratio in this group based on the broad definition was significantly higher than the PAT ratios for the other

groups.” (p 18) but it was in the limitations section. So we have moved this now and instead mention it both in the results and the discussion where the PAT ratios are discussed.

“PAT estimates for clinical MD were close to those for broad MD, with overlapping CIs in each case (Fig. 1). We were unable to estimate PAT ratios for clinical MD in the Latinx/Hispanic group because of insufficient numbers of cases.” (p 12)

“With a PAT ratio of 64%, the transferability of MD loci discovered in European ancestry samples was much higher for the Hispanic/Latinx group. This finding may reflect that the Hispanic/Latinx group contained many participants with a high proportion of European ancestry (https://www.ncbi.nlm.nih.gov/projects/gap/cgi-bin/study.cgi?study_id=phs000810.v1.p1, https://www.ncbi.nlm.nih.gov/projects/gap/cgi-bin/study.cgi?study_id=phs000674.v3.p3). The majority of cases amongst this group were defined via symptom questionnaires rather than clinical MD. Hence, it may be possible that the transferability for clinical MD is even higher in this group. For African, South Asian, and East Asian ancestry, the PAT ratios for clinical MD were all below 0.5 and consistent with the estimates from the main analysis, demonstrating that heterogeneity in outcome definitions does not explain the limited transferability of MD loci across ancestry groups.” (p 16)

The reviewer’s assumption that most of the “signal” comes from the European and Latinx/Hispanic samples is incorrect. We identified 53 loci that were not previously reported and only reached genome-wide significance when adding data from ancestrally diverse participants to the previously published studies based on European ancestry. To test whether these were largely driven by data from the Latinx/Hispanic participants, we excluded the latter and redid the meta-analysis (Manhattan plot shown below). The Latinx/Hispanic group contributes 32% of the new samples. So some reduction in significant associations is expected. The total number of independent associations in the analysis without Latinx/Hispanic samples is 155 (81.6% of the 190 loci from the analysis including all groups) and 34 of these are novel.

Figure. Manhattan plot showing the results for the multi-ancestry meta-analysis excluding samples from the Latinx/Hispanic group. 155 independent variants are associated at genome-wide significance ($p < 5 \times 10^{-8}$).

2. Designation of “population”: The authors’ main response to my previous critique was to include PCA plots of the individual studies, which I appreciated. But I do not think this response addressed my main critique, which was about the difference between the ethnic samples defined by ancestry and the Hispanic sample, defined by self-identification only. They justify their approach primarily by amplifying the point that their category conforms to “custom” in the USA, which I do not find a scientifically compelling argument; as noted in my comments in Point 1 above, my concern has a clear implication for the interpretation of the paper, given the extreme phenotypic heterogeneity between this sample and all of the others. They refer to a plan to implement a better approach in future studies (p18), which I will look forward to seeing; but I don’t think that their future plan is relevant to considering the results reported now.

The United States Census Bureau covers 5 race groups and the ethnicity hispanic or latino origin and it can be a requirement for research to document races/ethnicity using these groups <https://www.census.gov/newsroom/blogs/random-samplings/2021/08/measuring-racial-ethnic-diversity-2020-census.html> .

These groupings are very widely used in genome-wide association studies that includes diverse ancestry/ethnic groups. We searched pubmed for “multi-ancestry genome-wide association study”

published since 2020. The 61 search results included 32 GWAS studies. Of these, 24 included a latin x or hispanic group, alongside ancestry groups, such as African and European. Only two studies with >2 groups did not have latinx/hispanic.

During the last revision, we not only improved our reporting of genetic ancestry and added PCA plots, but also leveraged a dedicated multi-ancestry meta-analysis method, MR-MEGA. The study-level PCA plot shows that studies of the same ancestry group cluster well together and that the on the study level Hispanic/Latinx studies also cluster well together and sit between the European, African and East Asian ancestry groups with respect to their ancestry composition.

The inclusion of this group in our manuscript is vital because of the immense underrepresentation in previous genetic studies, as we have previously demonstrated (see figure below).

Figure reproduced from Fatumo S, Chikowore B, Choudhury A, Ayub M, Martin AR, Kuchenbaecker K. A roadmap to increase diversity in genomic studies. *Nat Med.* 2022 Feb;28(2):243-250. doi: 10.1038/s41591-021-01672-4. Epub 2022 Feb 10. PMID: 35145307.)

We clearly explain the assignment strategy in the manuscript (p 10). We describe that only a small proportion of the cases in this group were defined as clinical MD and that this may affect the interpretation of the novel MD locus identified in this group (p15). We highlight the limitation that we were unable to assess transferability for clinical MD in this group in the discussion (p 16) and, following the reviewer's suggestion, now also in the results section (p 12).

Additional Comments

1. Ancestry-specific meta-analysis: As noted above, these analyses are mostly uninformative; they only find one significant association here in their main analysis, and given that it is in the Hispanic sample it cannot be considered a locus associated with MD as clinically defined. The colocalization work for the 2q24.2 locus looks at a number of genes that aren't in the Locus Zoom plot – why are they included? It is unclear why they do extensive analyses of the 6q locus (e.g. colocalization) when the locus wasn't significantly associated to begin with.

We discuss locus discovery based on the ancestry-specific analyses as well as the multi-ancestry meta-analysis. It is correct that the latter discovered a larger number of loci. However, we feel that it is important to highlight the results from the ancestry-specific analyses. For each of the non-European ancestry groups this is either the first or the largest major depression GWAS ever conducted. Our findings demonstrate that inclusion of ancestrally diverse samples enables new discoveries that were missed in the much larger samples of European ancestry.

As discussed above, a non-clinical case definition may be relevant to the interpretation of a locus but does not reduce the importance of the discovery. The vast majority of previously reported depression loci were identified in mixed outcome GWAS and were nevertheless strongly linked to a clinical MD factor in the SEM in the recent work by Naomi Wray's group.

As the reviewer says, the locus at 6q16 missed genome-wide significance ($p=5.3 \times 10^{-8}$). We are describing this clearly in the manuscript (p 11, 15). Following the reviewer's suggestion we removed most of the text describing the gene link and its biological function. However, we would like to retain the current brief description of its association. Our manuscript is the first to investigate MD genetics in diverse ancestral and ethnic groups. In the literature, no association has so far been described in individuals of African

descent. Moreover, this locus has high biological plausibility. We leave it to the editor to decide and will remove all mention of this locus if preferred, however.

In the text, the authors refer to (e.g.) H3:PP. They have defined H3, but have not yet defined PP, I assume it's posterior probability but please define this, I asked for it in the first review. Supplementary Table 2 does not define H0, H1, or H2.

We thank the reviewer for pointing this out. We have now included definitions for H0, H1, H2, H3, H4 and PP in the manuscript and in Supplementary Table 3 (Previously Supplementary Table (ST) 2).

2. Gene prioritization: On page 13 the authors state: "To better understand the biological mechanisms of our GWAS findings, we performed several in silico analyses to functionally annotate and prioritise the most likely causal genes. We carried out a transcriptome-wide association study (TWAS) for expression in tissues relevant to MD33. We combined the TWAS results with Functional Mapping and Annotation (FUMA), conventional and HiC-MAGMA, to prioritise target genes. We also carried out ancestry-specific eQTL and pQTL colocalisation analyses (Supplementary Table 8)." As this section comes after the multi-ancestry meta-analysis results, and that was the analysis to produce multiple genome-wide significant findings, I assume that this section will pertain to results from the multi-ancestry meta-analysis. Yet the first table cited is Supplementary Table 8, which refers, I think, to the transferability analysis of the 196 previously found loci. At least the title of the table says: Supplementary Table 8. QTL mapping for the transferability SNPs for all tissues – there are 102 SNPs here – are these SNPs that transferred AND are associated in the multi-ancestry meta-analysis? I believe Supplementary Table 8 in the current draft was Supplementary Table 5 in the previous draft. The rebuttal letter states: The original Supplementary Table 5 that Reviewer #1 referred to showed the results of the HiC-Magma analysis, which is based on a larger selection of tissues (bone, kidney, etc.). If Supplementary Table 8 is displaying HiC-MAGMA results, why isn't it cited after reference to HiC-MAGMA in the text? If it is HiC-MAGMA results, why does the table title refer to QTL? This is quite confusing.

We would like to thank the reviewer for highlighting these issues. To improve the clarity of the manuscript and the different analyses within this study, we have now developed a schematic diagram that gives an overview of the different analytical steps to provide better guidance to the reader (Figure 1). We have also included the corresponding Supplementary Table numbers at each analysis for more clarity.

The reviewer is correct that the section above on gene prioritisation pertains to results from the multi-ancestry meta-analysis results, and we have now clarified this in the manuscript (p 14). We agree with the reviewer that the naming and numbering of the Supplementary Tables became difficult to follow during the revision process and we apologise for this. We have now addressed this and carefully revised the Supplementary Tables, improved the table titles and added definitions throughout. For more clarity, we would like to include details of the following Supplementary Tables below:

Supplementary Table 6 (ST5 and ST8 previously) - “Ancestry-specific eQTL/pQTL coloc”: refers to the results of the ancestry-specific eQTL and pQTL colocalisation analyses (QTL mapping for the transferability SNPs for all tissues) where we present genes linked to each previously reported MD SNP by ancestry (Please also see the schematic diagram (Figure 1)). We include this table because differences in e/pQTL associations could explain the lack of transferability for some loci.

Supplementary Tables 11-15.: show the results of the TWAS, FUMA, MAGMA, HiC-MAGMA analyses.

3. Mendelian Randomization: The bidirectional results are difficult to explain, biologically, especially when results differ in direction for the same phenotypes in different ancestries, and the p-values in many cases are pretty weak. It’s hard to take this as good evidence for causality or reverse causality.

The Mendelian Randomisation analyses show that the causal link between MD and unfavourable cardiometabolic profiles (BMI, lipids) is restricted to individuals of European ancestry. This is corroborated by other work we were involved in (O’Loughlin, 2023, *BMC Med*). These differences between ancestry groups are meaningful. The relationship between cardiometabolic profiles and MD appears to be setting-specific and therefore unlikely to be due to shared biological pathways (e.g. inflammation).

4. Conclusion: The final paragraph of the discussion includes a sentence I commented on in the first review: "In conclusion, in this first large-scale multi-ancestry GWAS of MD, we demonstrated through transferability analyses that a notable proportion of loci is specific to certain ancestry groups." In the rebuttal letter the authors say: Therefore, conclusions can be drawn based on the transferability analysis where PAT ratios ranged from ~0.3-0.6. There were multiple loci that were well powered to display an association in the diverse ancestral groups but had p-values larger than 5% (e.g., 35 well-powered loci not associated in the African ancestry GWAS). I would interpret this to mean that “a notable proportion of loci

is specific to European populations” based on the evidence you’ve shown us, as there are loci associated in Europeans, that were powered to be detected in diverse ancestry groups, that were not associated.

We thank the reviewer for pointing this out and have now modified the conclusion section to reflect that we have strong evidence that many of the MD loci discovered in European ancestry are population-specific. Due to the limited number of significant loci from other ancestry groups, it is not yet possible to draw firm conclusions about transferability for loci found in these groups.

Other more minor comments (not in order of importance)

1. Supplementary Table 1 is a nice addition to better understand the composition of the studies. The table is rather sloppy. Under “depression measure” what is the difference between “Medically diagnosed”, “Medically” and “Medical diagnosis” – are these the same? What is the difference between “Lifetime depression”, “MDD” and “Lifetime diagnosis”? Is “Depression diagnosis” equivalent to “self-report”? For definitions that were symptom-based, the authors did not always give the threshold for the CES-D and PHQ-9 to declare a participant to be a case – can they please update this? What does “computer assisted” mean with respect to “depression measure”? Could the authors add to this table columns indicating whether the study provided raw genotype data or summary statistics? The text says that there are 21 studies, but in counting the number of studies in this table it appears to be more (first column), can they clarify?

We thank the reviewer for this comment. We have now amended Supplementary Table 1; added more clarity, improved consistency and included definitions throughout. We now categorised all studies into “Structured interview”, “Medical records”, “Symptom-based” or “Multiple” in case more than one source of information was used by the study to define cases. We have also updated the thresholds used (where any) for the CES-D/PHQ-9 to assign MD cases.

We have also added an extra column specifying the type of data provided for each study and clarified and clearly labelled the number of studies used in our analyses in the manuscript and in Supplementary Table 1. Briefly, for the analyses of the African, East Asian, South Asian and Hispanic/Latinx group, we included data from 21 cohorts, with ancestrally diverse participants (consistent with the text of the main manuscript). We considered the Taiwan Major Depressive Disorder study - Platform 1 and Platform 2 **as one study**. Supplementary Table 1 shows all the datasets that were included in our study (24 cohorts in

total), including two previously published studies of MD, using data from ancestrally European participants: the PGC-MDD2 and the AGDS, which were used in our multi-ancestry meta-analysis of MD.

2. The summary tables in general have a lot of typos and often do not define the column headers. Frequently, for association results, a column is titled “beta” and I think this is the log-odds ratio – can the authors please be more precise regarding what “beta” means (this is true in the text as well, please specify logOR rather than beta).

Thank you for highlighting these issues. We have carefully revised the Supplementary Tables and added column header definitions throughout. We have also defined beta (regression coefficient) for our association results in the main text of the manuscript, as well as in the Supplementary Tables. Interpreting the estimates in terms of how much risk/increased odds each variant confers should be done with caution because of the different designs and outcome definitions of the studies. Therefore, we prefer not to write the regression coefficients as logORs.

3. Supplementary Table 3 lists results for 196 loci but in several places in the text (e.g. page 11) they say 206 loci – which is correct?

We apologise for the confusion. Overall, 206 loci have been previously reported. However, not all of these were available in our results (e.g. due to quality control filtering). Therefore, our assessment of the transferability was done for 196 loci for which we had results in at least one of the diverse ancestry/ethnic groups. We have now clarified that both on page 11 and in the Supplementary Tables. (ST.3 in the previous draft is now Supplementary Table 4).

4. Supplementary Table 4 has been expanded quite a bit with more information, but it is a little confusing. For example, take rs2089358, the p-value for association in the African sample is 0.47, yet it is indicated that this locus transferred. It must be that some other variant among the 23 in the credible set had a p-value < transfer threshold (?). It would be helpful to include the result for the SNV that did transfer (or at least the most significant SNV in the credible set), as opposed to the results for what I guess are the lead SNPs.

The reviewer is correct. As we describe in the methods, the lead variant may not be the causal variant. Therefore, looking up the lead variant only may lead to incorrect conclusions. We have now clarified that by adding a note to the table (now Supplementary Table 5): “A locus was classified as transferable if at least one variant in the credible set was associated below the p-value threshold. The lead variant may not be significant for a transferable locus.

The p-value thresholds were: $p < 0.008341$, 0.007378, 0.006847 and 0.003147 for samples of African, East Asian, South Asian ancestry and Hispanic/Latinx, respectively.” (p24)

Moreover, we have added SNP with the smallest p-value from the credible set for each locus, as suggested by the reviewer.

5. Supplementary Tables 12 and 13 are missing a column for gene symbol, it’s hard to interpret findings without this information.

We thank the reviewer for this comment. We have now included a column for gene symbols in Supplementary Tables 14 and 15 (Previously ST12 and ST13).

6. The supplementary Manhattan plots don’t always have the y-axis extended to the point where the red line for genome-wide significance is displayed – can the authors correct this so that all plots have the y-axis extend to at least this level?

We thank the reviewer for highlighting this issue with the Manhattan plots. We have now corrected this and updated the plots, so that all plots have the y-axis extended to the point where the red line for genome-wide significance is displayed.

7. The text states that 43 genes had consistent results across prioritization methods (page 14) but Supplementary Table 5 says there are 44 – which is correct?

We thank the reviewer for raising this point. We have carefully checked this and can confirm that: 43 genes displayed consistent evidence of association across all the four gene prioritisation methods tested (TWAS, FUMA, MAGMA, HIC-MAGMA). The results for these tests are shown in Table 1 and in Supplementary Tables 11-15.

Furthermore, we would like to clarify that Supplementary table 9. (which might have been Supplementary Table 5 in the previous draft) shows the results of the multi-ancestry meta-analysis approach implemented in MR-MEGA, which resulted in 44 independent regions associated with MD after lambda GC correction (Supplementary Table 9)

8. On page 16, the authors state: "Gene *NDUFAF3* encodes mitochondrial complex I assembly protein which is the main target of the drug metformin⁴³, the first-line drug for treatment of type 2 diabetes. Research in model organisms has provided a tentative link to a reduction in depression and anxiety⁴⁴." In that last sentence, I think the authors mean that use of metformin is linked to a reduction in depression and anxiety.

We thank the reviewer for this comment. That is correct. We have now clarified that there is a tentative link between metformin and a reduction in depression (p17). For the first time, our results showed that expression of the gene *NDUFAF3* (drug target of metformin) is significantly associated ($P = 3.80 \times 10^{-7}$) with MD in our multi-ancestry, multi-tissue TWAS. Our results suggest that the gene *NDUFAF3* might be important for drug repurposing and a promising drug target for MD (not just for metformin).

Decision Letter, second revision:

13th July 2023

Dear Karoline,

Thank you for submitting your revised manuscript "Multi-ancestry GWAS of major depression aids locus discovery, fine-mapping, gene prioritisation, and causal inference" (NG-A60603R1). In light of your revisions and responses to Reviewer #1, we will be happy in principle to publish your study in Nature Genetics as an Article pending final revisions to comply with our editorial and formatting guidelines.

We are now performing detailed checks on your paper, and we will send you a checklist detailing our editorial and formatting requirements soon. Please do not upload the final materials or make any revisions until you receive this additional information from us.

Thank you again for your interest in Nature Genetics. Please do not hesitate to contact me if you have any questions.

Sincerely,
Kyle

Kyle Vogan, PhD
Senior Editor
Nature Genetics
<https://orcid.org/0000-0001-9565-9665>

Final Decision Letter:

26th October 2023

Dear Karoline,

I am delighted to say that your manuscript "Multi-ancestry genome-wide association study of major depression aids locus discovery, fine-mapping, gene prioritization and causal inference" has been accepted for publication in an upcoming issue of Nature Genetics.

Your paper will be published online after we receive your corrections and will appear in print in the next available issue. You can find out your date of online publication by contacting the Nature Press Office (press@nature.com) after sending your e-proof corrections. Now is the time to inform your Public Relations or Press Office about your paper, as they might be interested in promoting its publication. This will allow them time to prepare an accurate and satisfactory press release. Include your manuscript tracking number (NG-A60603R2) and the name of the journal, which they will need when they contact our Press Office.

Before your paper is published online, we will be distributing a press release to news organizations worldwide, which may very well include details of your work. We are happy for your institution or funding agency to prepare its own press release, but it must mention the embargo date and Nature Genetics. Our Press Office may contact you closer to the time of publication, but if you or your Press Office have any enquiries in the meantime, please contact press@nature.com.

Acceptance is conditional on the data in the manuscript not being published elsewhere, or announced in the print or electronic media, until the embargo/publication date. These restrictions are not

intended to deter you from presenting your data at academic meetings and conferences, but any enquiries from the media about papers not yet scheduled for publication should be referred to us.

Please note that Nature Genetics is a Transformative Journal (TJ). Authors may publish their research with us through the traditional subscription access route or make their paper immediately open access through payment of an article-processing charge (APC). Authors will not be required to make a final decision about access to their article until it has been accepted. [Find out more about Transformative Journals](https://www.springernature.com/gp/open-research/transformative-journals)

Authors may need to take specific actions to achieve [compliance](https://www.springernature.com/gp/open-research/funding/policy-compliance-faqs) with funder and institutional open access mandates. If your research is supported by a funder that requires immediate open access (e.g. according to [Plan S principles](https://www.springernature.com/gp/open-research/plan-s-compliance)), then you should select the gold OA route, and we will direct you to the compliant route where possible. For authors selecting the subscription publication route, the journal's standard licensing terms will need to be accepted, including [self-archiving-and-license-to-publish](https://www.nature.com/nature-portfolio/editorial-policies/self-archiving-and-license-to-publish). Those licensing terms will supersede any other terms that the author or any third party may assert apply to any version of the manuscript.

If you have not already done so, we invite you to upload the step-by-step protocols used in this manuscript to the Protocols Exchange, part of our on-line web resource, natureprotocols.com. If you complete the upload by the time you receive your manuscript proofs, we can insert links in your article

that lead directly to the protocol details. Your protocol will be made freely available upon publication of your paper. By participating in natureprotocols.com, you are enabling researchers to more readily reproduce or adapt the methodology you use. [Natureprotocols.com](https://natureprotocols.com) is fully searchable, providing your protocols and paper with increased utility and visibility. Please submit your protocol to <https://protocolexchange.researchsquare.com/>. After entering your nature.com username and password you will need to enter your manuscript number (NG-A60603R2). Further information can be found at <https://www.nature.com/nature-portfolio/editorial-policies/reporting-standards#protocols>

Sincerely,
Kyle

Kyle Vogan, PhD
Senior Editor
Nature Genetics
<https://orcid.org/0000-0001-9565-9665>